# Fair Classification by Direct Intervention on Operating Characteristics

**Kevin Jiang & Edgar Dobriban**
Department of Statistics and Data Science
University of Pennsylvania
Philadelphia, PA 19104, USA
`{kcjiang,dobriban}@wharton.upenn.edu`

## Abstract

We develop new classifiers under group fairness in the attribute-aware setting for binary classification with multiple group fairness constraints (e.g., demographic parity (DP), equalized odds (EO), and predictive parity (PP)). We propose a novel approach, applicable to linear fractional constraints, based on directly intervening on the operating characteristics of a pre-trained base classifier, by (i) identifying optimal operating characteristics using the base classifier's group-wise ROC convex hulls and (ii) post-processing the base classifier to match those targets. As practical post-processors, we consider randomizing a mixture of group-wise thresholding rules subject to minimizing the expected number of interventions. We further extend our approach to handle multiple protected attributes and multiple linear fractional constraints. On standard datasets (COMPAS and ACSIncome), our method simultaneously satisfies approximate DP, EO, and PP with few interventions and a near-oracle drop in accuracy; comparing favorably to previous methods.

## 1 Introduction

Modern machine learning systems inherit and can amplify historical and measurement biases present in data, raising concerns in high-stakes applications such as criminal justice, credit, hiring, and healthcare (see e.g., Barocas & Selbst, 2016; Obermeyer et al., 2019; Raghavan et al., 2020; Chouldechova & Roth, 2020; Fuster et al., 2022, etc). These concerns have galvanized the field of *algorithmic fairness* (see e.g., Barocas et al., 2023), which formalizes fairness criteria and develops methods to enforce them.

One broad family of metrics, termed *group fairness* criteria, requires that a performance metric of a model is equal across protected groups (e.g., race, gender, age). These performance metrics are often motivated by legal rulings, regulatory guidance, and broader normative considerations (e.g., anti-discrimination norms, see Hardt et al. (2016); Verma & Rubin (2018); Barocas et al. (2023)).

Methods designed to ensure group fairness can be roughly categorized into three categories based on where they intervene in the training pipeline: *pre-processing* intervenes on the training data via methods like reweighing/sampling, causal inference, and adversarial debiasing (see e.g., Salazar et al., 2021; Zhang & Sang, 2020; Zeng et al., 2024; Plečko et al., 2024, etc); *in-processing* modifies the learner's objective or model architecture (see e.g., Zhang et al., 2018a; Agarwal et al., 2018; Cho et al., 2020, etc); and *post-processing* alters the predictions of a fixed learned model (see e.g., Hardt et al., 2016; Alghamdi et al., 2022; Xian & Zhao, 2024, etc). These methods have found broad applications (see e.g., Jammalamadaka & Itapu, 2023; Weerts et al., 2023; Mackin et al., 2025, etc).

In parallel, policy and litigation have highlighted the desire to simultaneously satisfy multiple fairness constraints (e.g., analyses surrounding the COMPAS model, see Larson & Angwin (2016); Dieterich et al. (2016)). However, incompatibility results in the literature show that perfect compliance across sufficiently many metrics is generally impossible except in special cases (Chouldechova, 2017; Kleinberg, 2018; Defrance & De Bie, 2025). Consequently, many works adopt *approximate* group fairness, where a model's performance across groups are only required to be approximately equal (see, e.g., Celis et al. (2019); Zeng et al. (2024)).

**Our contributions.** In this work, we develop new classifiers aiming for approximate group fairness for binary classification with multiple group fairness constraints (e.g., demographic parity (DP), equalized odds (EO), and predictive parity (PP)), as long as they have a linear fractional representation related to that considered in Celis et al. (2019). Our novel approach directly intervenes on the operating characteristics of a pre-trained base classifier, by: (i) identifying optimal operating characteristics using the base classifier's group-wise receiver operating characteristic (ROC) convex hulls; and then (ii) post-processing the base classifier to match those targets. We extend our approach to handle multiple protected attributes and multiple linear fractional constraints. On standard datasets (COMPAS and ACSIncome), our method simultaneously satisfies approximate DP, EO, and PP. They lead to few changes to the predictions made by the original classifier and a nearly optimal drop in accuracy; comparing favorably to previous methods.

## 2 RELATED WORK

Both empirically and theoretically, it is known that satisfying multiple fairness constraints can be challenging (Chouldechova, 2017; Kleinberg, 2018; Majumder et al., 2023; Defrance & De Bie, 2025). Bell et al. (2023) show that, in principle, there may exist classifiers that achieve approximate fairness for multiple constraints. However, this assumes the existence of classifiers that can satisfy an arbitrary assignment of labels to the test set, and also requires tuning population-level quantities that are typically out of our control.

In highly relevant work, Celis et al. (2019) propose a meta-algorithm based on an optimization perspective in the space of classifiers. Our approach differs fundamentally, since we rely on the geometry of the realizable regions traced by the convex hull of each group's ROC curve. Hsu et al. (2022) develop post-processing methods based on mixed-integer linear programming for a more granular notion of fairness, equalizing group rates across scores/predicted probabilities as opposed to binary labels. Their definitions reduce to ours when using two bins. With this approach, empirically, our method outperforms those from Celis et al. (2019) and Hsu et al. (2022). Due to space limitations, additional related work is reviewed in Section A.1.

## 3 PROBLEM SETUP

We consider a fair binary classification problem where two types of feature are observed [1]: the usual feature $X \in \mathcal{X}$ and the *protected feature* $A \in \mathcal{A}$ which we take to be discrete, with $\mathcal{A} = \{1, 2, \ldots, m\}$, $m \geq 2$. We would like to achieve non-discrimination with respect to the *protected groups*, defined by the values $a \in \mathcal{A}$ of the protected feature. The features and binary label are distributed according to $(X, A, Y) \sim \mathbb{P}$. Given a pre-trained probabilistic predictor $s : \mathcal{X} \times \mathcal{A} \to [0, 1]$ and an i.i.d.sampled post-processing set [2] $\mathcal{D}_{\text{post}} := \{(x_i, a_i, y_i)\}_{i=1}^{N}$ drawn from $\mathbb{P}$, we seek to develop a (possibly randomized) classifier $f : \mathcal{X} \times \mathcal{A} \to \{0, 1\}$ that minimizes the population risk $\mathcal{L}(f) = \mathbb{E}[\ell(f(X, A), Y)]$, for a loss $\ell : \{0, 1\} \times \mathcal{Y} \to \mathbb{R}$ subject to $K \geq 1$ *fairness constraints*.

To impose fairness constraints, we consider equalizing certain *group performance functions* $G_{k,a}$, which reflect the performance of the classifier over protected groups. We focus on loss functions and group performance rates which can be expressed as ratios of linear combinations of a classifier's true positive rate (TPR) and false positive rate (FPR); as well as the associated false negative rate (FNR) and true negative rate (TNR), for each group [3] $a \in \mathcal{A}$:

$$\text{TPR}_a(f) := \Pr(f{=}1 \mid Y{=}1, A{=}a), \qquad \text{FPR}_a(f) := \Pr(f{=}1 \mid Y{=}0, A{=}a);$$
$$\text{FNR}_a(f) := 1 - \text{TPR}_a(f), \qquad \text{TNR}_a(f) := 1 - \text{FPR}_a(f). \tag{1}$$

We consider the following class of group performance functions, a special case of more general notions from Celis et al. (2019), but encompassing many common fairness metrics, see Table 1).

---

[1] We defer the extension for multiple classes to §4.4.

[2] In the post-processing literature for fairness, this set has been alternatively called a *holdout* or *calibration* set (Pleiss et al., 2017; Hansen et al., 2024).

[3] When it is clear which classifier $f$ the group-wise operating characteristics are tied to, we will sometimes omit the classifier from the argument, writing, for instance $\text{TPR}_a$, etc.

Table 1: Fairness metrics covered by our LF/L group performance metrics

| metric | type | coefficients $(\vec{u}_{k,a}; \vec{v}_{k,a})$ |
|---|---|---|
| Demographic parity (DP) | linear | $\vec{u} = (\pi_a, 1 - \pi_a, 0); \vec{v} = (0, 0, 1)$ |
| Equal opportunity (TPR parity) | linear | $\vec{u} = (1, 0, 0); \vec{v} = (0, 0, 1)$ |
| Predictive equality (FPR parity) | linear | $\vec{u} = (0, 1, 0); \vec{v} = (0, 0, 1)$ |
| Equalized odds (TPR & FPR parity) | linear pair | — |
| Predictive parity (PPV parity) | lin.–frac. | $\vec{u} = (\pi_a, 0, 0); \vec{v} = (\pi_a, 1 - \pi_a, 0)$ |
| False omission rate (FOR) parity | lin.–frac. | $\vec{u} = (-\pi_a, 0, \pi_a); \vec{v} = (-\pi_a, -(1 - \pi_a), 1)$ |
| Accuracy parity | linear | $\vec{u} = (\pi_a, -(1 - \pi_a), 1 - \pi_a); \vec{v} = (0, 0, 1)$ |

*Notes.* Here, $\pi_a = \Pr(Y{=}1 \mid A{=}a)$ denotes the group prevalence/base rate. For linear–fractional (lin.–frac.) metrics, denominator positivity is required: for PPV, $\pi_a \text{TPR}_a + (1 - \pi_a)\text{FPR}_a > 0$ (nonzero selection); for FOR, $\pi_a(1 - \text{TPR}_a) + (1 - \pi_a)(1 - \text{FPR}_a) > 0$ (nonzero non-selection).

**Definition 3.1** (LF group performance functions). For each group $a$ and constraint $k$, a linear fractional (LF) group performance function $G_{k,a}$ is defined as

$$G_{k,a}(f) := \frac{\langle \vec{u}_{k,a}, \vec{\rho}_a \rangle}{\langle \vec{v}_{k,a}, \vec{\rho}_a \rangle} \quad \text{with} \quad \vec{\rho}_a := (\text{TPR}_a(f), \text{FPR}_a(f), 1)^\top, \langle \vec{v}_{k,a}, \vec{\rho}_a \rangle > 0,$$

where $\vec{u}_{k,a}, \vec{v}_{k,a} \in \mathbb{R}^3$ are coefficient vectors that may depend on the underlying population distribution, but not on the classifier $f$. A linear group performance function $G_{k,a}(f) = \langle \vec{u}_{k,a}, \vec{\rho}_a \rangle$ is obtained when $\vec{v}_{k,a} = (0, 0, 1)^\top$ selects the constant component only.

*Linear fractional (linear) fairness constraints* are functions of only linear fractional (linear) group performance functions. For instance, demographic parity (Calders et al., 2009) is a linear fairness constraint, since it requires $G_{\text{DP},a}(f) - G_{\text{DP},a'}(f) = 0$ for all $a \neq a'$, where the group-wise selection rate is, with $\pi_a := \Pr(Y{=}1 \mid A{=}a)$,

$$G_{\text{DP},a}(f) = \Pr(f{=}1 \mid A{=}a) = \pi_a \text{TPR}_a + (1 - \pi_a)\text{FPR}_a = \frac{\langle \vec{u}_{\text{DP},a}, \vec{\rho}_a \rangle}{\langle \vec{v}_{\text{DP},a}, \vec{\rho}_a \rangle},$$

with $\vec{u}_{\text{DP},a} = (\pi_a, 1 - \pi_a, 0)$, $\vec{v}_{\text{DP},a} = (0, 0, 1)$. *Predictive parity* (Chouldechova, 2017) is a linear fractional constraint, since it requires $G_{\text{PP},a}(f) - G_{\text{PP},a'}(f) = 0$ for all $a \neq a'$, where the positive predictive value is

$$G_{\text{PP},a}(f) = \Pr(Y{=}1 \mid f{=}1, A{=}a) = \frac{\pi_a \text{TPR}_a}{\pi_a \text{TPR}_a + (1 - \pi_a)\text{FPR}_a} = \frac{\langle \vec{u}_{\text{PP},a}, \vec{\rho}_a \rangle}{\langle \vec{v}_{\text{PP},a}, \vec{\rho}_a \rangle},$$

where $\vec{u}_{\text{PP},a} = (\pi_a, 0, 0)$, $\vec{v}_{\text{PP},a} = (\pi_a, 1 - \pi_a, 0)$, with denominator $\langle \vec{v}_{\text{PP},a}, \vec{\rho}_a \rangle = \Pr(f{=}1 \mid A{=}a) > 0$ for groups with nonzero selection rate.

Moreover, since satisfying multiple fairness constraints exactly may be impossible (Chouldechova, 2017; Kleinberg, 2018; Defrance & De Bie, 2025), we consider $\vec{\delta}$-*approximate fairness*, as, e.g., (Xian & Zhao, 2024; Zeng et al., 2024). This requires pairwise differences of group performance functions to differ by no more than $\delta_k \geq 0$: $|G_{k,a}(f) - G_{k,a'}(f)| \leq \delta_k, \forall a \neq a', k \in [K]$, where we further collect all the user-specified disparities $\delta_k$ as a single vector $\vec{\delta} \in [0, 1]^k$.

We consider loss functions that are a linear combination of *linear* group performance functions[4] $\mathcal{L}(f) = \sum_{a \in \mathcal{A}} \langle \vec{\gamma}_a, \vec{\rho}_a \rangle, \vec{\rho}_a = (\text{TPR}_a, \text{FPR}_a, 1)^\top, \vec{\gamma}_a \in \mathbb{R}^3$. Altogether, on the population level, our fairness constrained optimization problem is then

---

[4]The usual misclassification rate $\mathcal{L}(f) = \sum_{a \in \mathcal{A}} p_a [\pi_a(1 - \text{TPR}_a) + (1 - \pi_a)\text{FPR}_a] = \Pr\left(f(X, A) \neq Y\right)$, can be recovered by taking coefficients $\vec{\gamma}_a$ to be $\vec{\gamma}_a = (-p_a \pi_a, p_a(1 - \pi_a), p_a \pi_a), p_a = \Pr(A{=}a), \pi_a = \Pr(Y{=}1 \mid A{=}a)$.

---

**Population-level Optimal Classification with Linear Fractional Fairness Constraints**

$$\min_{f:\mathcal{X}\times\mathcal{A}\to\{0,1\}} \sum_{a\in\mathcal{A}} \langle\vec{\gamma}_a, \vec{\rho}_a(f)\rangle \quad \text{s.t.} \max_{a,a'\in\mathcal{A}} |G_{k,a}(f) - G_{k,a'}(f)| \le \delta_k, k\in[K], \quad (2)$$

$$G_{k,a}(f) = \frac{\langle\vec{u}_{k,a}, \vec{\rho}_a(f)\rangle}{\langle\vec{v}_{k,a}, \vec{\rho}_a(f)\rangle}, \langle\vec{v}_{k,a}, \vec{\rho}_a(f)\rangle > 0, \forall a, k; \quad \vec{\rho}_a(f) := (\text{TPR}_a(f), \text{FPR}_a(f), 1)^\top,$$

where $\vec{\gamma}_a \in \mathbb{R}^3$ are fixed loss-coefficient vectors; $\vec{u}_{k,a}, \vec{v}_{k,a} \in \mathbb{R}^3$ are fixed coefficient vectors for constraint $k \in [K]$ and group $a \in \mathcal{A}$; and $\delta_k \ge 0$ are prescribed fairness tolerances for each $k \in [K]$.

The Bayes optimal regression function $\eta_a$ takes values, for all $x$, $\eta_a(x) := \Pr(Y{=}1|X{=}x, A{=}a)$, $a \in \mathcal{A}$. We call a classifier $f : \mathcal{X} \times \mathcal{A} \to \{0,1\}$ a *group-wise thresholding rule (GWTR)* if it thresholds $\eta_a$ at group-specific values $t_a, a \in \mathcal{A}$:

$$f(x,a) = 1\,(\eta_a(x) > t_a) \quad \text{for some} \quad t_a \in [0,1], a \in \mathcal{A}. \quad (3)$$

In what follows, we will be considering the convex hull of ROC curves, in which case points on the hull are obtained by allowing a mixture of thresholding rules.

**Definition 3.2** (Mixed-GWTR)**.** A (possibly randomized) classifier $f : \mathcal{X} \times \mathcal{A} \to \{0,1\}$ is a *mixed-GWTR* if, for each group $a \in \mathcal{A}$, there is a finite index set $J_a$, thresholds $\{t_{a,j} \in [0,1]\}_{j\in J_a}$, and nonnegative weights $\{\lambda_{a,j}\}_{j\in J_a}$ with $\sum_{j\in J_a} \lambda_{a,j} = 1$, such that

$$\mathbb{P}(f(x,a) = 1 \mid x, a) = \sum_{j\in J_a} \lambda_{a,j} \cdot 1\,(\eta_a(x) > t_{a,j}). \quad (4)$$

For a broad family of fairness constraints defined by linear fractional group performance functions, an optimal solution to the population problem equation 2 can be realized by a GWTR with suitable thresholds $\{t_{a^*}\}_{a\in\mathcal{A}}$ (Celis et al., 2019). This result, however, is not directly practicable, as it only describes the optimum at the level of the unknown population from which the data has been sampled, and presumes access to the regression functions $\eta_a$. In practice, one must estimate these functions as well as $\{t_a\}_{a\in\mathcal{A}}$. Due to this estimation step, in challenging situations when the sample size is small, the feasibility constraints can possibly become infeasible. We will see experimentally that this step can indeed be highly noise-sensitive.

## 4 FAIR CLASSIFICATION VIA ROC FEASIBILITY REGIONS

Motivated by the above observations, our approach relies on the following two steps.

**(A) Restricting to post-processors.** Given a pre-trained probabilistic predictor $s : \mathcal{X}\times\mathcal{A} \to [0,1]$—intended to approximate the label probabilities—and a post-processing set $\mathcal{D}_{\text{post}} = \{(x_i, a_i, y_i)\}_{i=1}^N$, we consider *post-processors* obtained from $s$:

$$\mathcal{F}_N \equiv \mathcal{F}_N(s; \mathcal{D}_{\text{post}}) := \{f(\cdot, a) \text{ obtained from } s(\cdot, a) \text{ and } \mathcal{D}_{\text{post}} \text{ (and possibly randomized)}\}. \quad (5)$$

**(B) Moving to operating characteristic space.** Instead of optimizing over functions $f$, we directly work with their *group-wise operating characteristics* (or, *rates*). For each group $a$, define the *realizable (or post-processing) ROC region*

$$\mathcal{R}_a(s) := \left\{(\text{tpr}, \text{fpr}) \middle| \exists f \in \mathcal{F}_N \text{ such that } (\text{TPR}_a(f), \text{FPR}_a(f)) = (\text{tpr}, \text{fpr})\right\}. \quad (6)$$

Allowing randomized thresholding as in Definition 3.2 ensures that the set of achievable rates starting from any given $s$ is convex; so $\mathcal{R}_a(s)$ coincides with the convex hull of the ROC points generated by thresholding $s(\cdot, a)$.[5]

---

[5]Hence, to restrict to a post-processor, we equivalently require the classifier's group rate vectors to lie in its ROC convex hull (see §A.2.1).

**Fair classification via operating characteristic feasibility regions (ROCF).** Our method, ROCF, departs from traditional post-processing methods (and associated direct threshold search) by directly working with operating characteristics. We work with post-processors, constraining each group's rates to lie in the (population) realizable ROC region $\widetilde{\mathcal{R}}_a(s)$ (see §4.1 for the population setting and §A.2.1 for its empirical convex hull analog).

We next provide an overview of the steps. Due to space considerations, some details are presented in the supplementary material. For clarity, we also provide the full end-to-end implementation of our procedure for the setting of controlling for single linear-fractional and linear constraint in §A.3.3.

## 4.1 RATE-SPACE REFORMULATION & POST-PROCESSING CONDITION

Let $\vec{\rho}_a = (\text{TPR}_a, \text{FPR}_a, 1)^\top$ be as before, and define the lifted realizable set $\widetilde{\mathcal{R}}_a(s) := \{\rho = (\text{tpr}, \text{fpr}, 1)^\top : (\text{tpr}, \text{fpr}) \in \mathcal{R}_a(s)\}$. To incorporate the post-processing constraint, and to move to the low-dimensional space of operating characteristics, we reformulate equation 2 to the form:

---

**Population-level Optimal Fair Classification via Post-Processor Operating Characteristics**

$$\min_{\{\vec{\rho}_a\}_{a \in \mathcal{A}}} \sum_{a \in \mathcal{A}} \langle \vec{\gamma}_a, \vec{\rho}_a \rangle \qquad \text{s.t.} \qquad \max_{a, a' \in \mathcal{A}} |G_{k,a}(\vec{\rho}_a) - G_{k,a'}(\vec{\rho}_{a'})| \leq \delta_k, k \in [K];$$

$$G_{k,a}(\vec{\rho}_a) = \frac{\langle \vec{u}_{k,a}, \vec{\rho}_a \rangle}{\langle \vec{v}_{k,a}, \vec{\rho}_a \rangle}, \langle \vec{v}_{k,a}, \vec{\rho}_a \rangle > 0, \forall a, k; \quad \vec{\rho}_a \in \widetilde{\mathcal{R}}_a(s), \forall a \in \mathcal{A}.$$

(7)

---

## 4.2 OPERATING CHARACTERISTIC FEASIBILITY REGIONS AND CENTROID-BASED LINEARIZATION

The objective from equation 7 can be non-convex in $\{\vec{\rho}_a\}_{a \in \mathcal{A}}$ due to the linear fractional group performance constraints. Following Celis et al. (2019) and Xian & Zhao (2024), we note that ensuring bounded pairwise differences $|G_{k,a} - G_{k,a'}| \leq \delta_k$ is equivalent to the existence of a *centroid* $q_k \in [0, 1]$ with $|G_{k,a} - q_k| \leq \delta_k/2$ for all $a \in \mathcal{A}$. Thus, to reduce the number of constraints, for each constraint $k \in [K]$, introduce a centroid $q_k$. Moreover, since pure linear constraints stay linear after the introduction of the centroids, we split the constraints: Let $\mathcal{K}_L$ and $\mathcal{K}_{LF}$ denote the indices of linear and linear fractional fairness constraints, respectively, so that $\mathcal{K}_L \sqcup \mathcal{K}_{LF} = [K]$.

**Linear constraints via centroids.** For $k \in \mathcal{K}_L$, the disparity constraint in equation 7 is equivalent to the existence of a centroid $q_k$ such that $-\frac{\delta_k}{2} \leq \langle \vec{u}_{k,a}, \vec{\rho}_a \rangle - q_k \leq \frac{\delta_k}{2}, \forall a \in \mathcal{A}$. These are linear inequalities in $(\vec{\rho}_a, q_k)$, so we can treat $q_k$ as a decision variable while still having an LP.

**Linear–fractional constraints via fixed centroids.** For $k \in \mathcal{K}_{LF}$, write $U_{k,a}(\rho) := \langle \vec{u}_{k,a}, \rho \rangle$ and $V_{k,a}(\rho) := \langle \vec{v}_{k,a}, \rho \rangle$. Assuming $V_{k,a}(\vec{\rho}_a) > 0$, the centroid form $|U_{k,a}(\vec{\rho}_a)/V_{k,a}(\vec{\rho}_a) - q_k| \leq \frac{\delta_k}{2}$ is equivalent to the pair of linear inequalities

$$U_{k,a}(\vec{\rho}_a) - \left(q_k + \frac{\delta_k}{2}\right) V_{k,a}(\vec{\rho}_a) \leq 0, \quad \left(q_k - \frac{\delta_k}{2}\right) V_{k,a}(\vec{\rho}_a) - U_{k,a}(\vec{\rho}_a) \leq 0, \quad \forall a \in \mathcal{A}. \quad (8)$$

Thus, for any fixed centroid $q_k$, the linear fractional constraint becomes linear in $\vec{\rho}_a$. Given $\vec{q} = (q_k)_{k \in \mathcal{K}_{LF}}$, the set $\mathfrak{F}(\vec{q}; s, \vec{\delta})$ of $\left\{\{\vec{\rho}_a\}_{a \in \mathcal{A}} \text{ satisfying the above constraints}\right\}$ is called an *operating characteristic feasibility region* (see Definition A.6), which we simply refer to as *feasibility region* for convenience.

**Inner linear program for fixed linear fractional centroids.** Given centroids $\{q_k\}_{k \in LF}$, we obtain the following linear optimization problem in the rate variables $\vec{\rho}_a$ and the linear-centroids $q_k$, with $\widetilde{\mathcal{R}}_a$ from the beginning of Section 4.1 and $V_{k,a}(\rho) = \langle \vec{v}_{k,a}, \rho \rangle$ from above equation 8:

---

**Inner Optimization for Fixed Linear Fractional Centroids**

$$\min_{\{\vec{\rho}_a\}_{a\in\mathcal{A}},\{q_k\}_{k\in\mathcal{K}_{\mathrm{L}}}} \quad \sum_{a\in\mathcal{A}} \langle\vec{\gamma}_a,\vec{\rho}_a\rangle, \text{s.t.} -\tfrac{\delta_k}{2} \leq \langle\vec{u}_{k,a},\vec{\rho}_a\rangle - q_k \leq \tfrac{\delta_k}{2}, \forall a\in\mathcal{A}, k\in\mathcal{K}_{\mathrm{L}},$$

$$\text{equation 8 holds } \& \ V_{k,a}(\vec{\rho}_a) > 0, \ \forall k\in\mathcal{K}_{\mathrm{LF}}; \quad \vec{\rho}_a \in \widetilde{\mathcal{R}}_a(s), \forall a\in\mathcal{A}. \tag{9}$$

---

Moreover, at the population level $\widetilde{\mathcal{R}}_a$ is convex. If it is represented polyhedrally (as will be the case empirically via convex hulls in §A.2.1), then equation 9 is a linear program. In practice, for stability purposes, we additionally enforce that the denominators are strictly bounded away from zero via the linear constraints $V_{k,a}(\vec{\rho}_a) \geq \varepsilon_k > 0, \forall a\in\mathcal{A}, k\in\mathcal{K}_{\mathrm{LF}}$, for negligibly small positive constants (e.g., $\varepsilon_k = 1e-7$ and see §A.5.1).

**Outer search over linear fractional centroids.** For each linear fractional constraint $k\in\mathcal{K}_{\mathrm{LF}}$, we restrict the centroid to a compact interval $\mathcal{Q}_k \subset (0,1)$ that is consistent with post-processing and denominator positivity, i.e., choices of $q_k$ for which equation 8 holds. We show later in §A.4 that we can take $\mathcal{Q}_k = [\delta_k/2, 1 - \delta_k/2]$. We call these *admissible* centroids.

We then search over $\vec{q} \in \mathcal{Q} := \prod_{k\in\mathcal{K}_{\mathrm{LF}}} \mathcal{Q}_k$ (e.g., via a coarse grid search) and solve the inner linear program (cf. equation 9) at each $\vec{q}$. The explicit construction of $\mathcal{Q}_k$ and the exact linear bands for common LF metrics (e.g., predictive parity) are deferred to the supplementary material (see §A.5.1).

This outer search for fixed values of linear fractional centroids is justified by the following theorem (proof is deferred to Appendix A.4).

**Theorem 4.1.** *Define the value function $\Phi$ such that $\Phi(\vec{q})$ is the optimal objective value of equation 9 for any $\vec{q} \in \mathbb{R}^{|\mathcal{K}_{\mathrm{LF}}|}$. Then, the value of the optimization problem from equation 7 is equal to $\min_{\vec{q}\in\mathcal{Q}} \Phi(\vec{q})$. Moreover, we can find an optimizer $\{\vec{\rho}_a^\star\}_{a\in\mathcal{A}}$ of the objective in equation 7 by selecting any minimizer $\vec{q}^\star \in \arg\min_{\vec{q}\in\mathcal{Q}} \Phi(\vec{q})$ and then optimizing the objective in equation 9 at $\vec{q}^\star$.*

**Empirical region search over ROC-hull supports.** Given the post-processing set, we search over plug-in ROC-hull supports and solve modest-sized LPs for centroid-specific feasibility. A feasibility guard ensures that, when needed, tolerances are minimally relaxed (see §A.2.1 for details).

The following theorem demonstrates that the minimizer of our empirical region search procedure has a $\tilde{O}(1/\sqrt{n})$ rate of convergence to the optimal fairness-constrained minimizer of equation 7 in risk error and fairness attainment, for any fixed probabilistic predictor. We will suppose that the feasible set of the population-level fairness-constrained problem from equation 7 is nonempty, and let $\varrho^\star = \{\vec{\rho}_a^\star\}_{a\in\mathcal{A}}$ be an optimizer.[6] Denote the objective value of equation 7 for group-wise lifted operating characteristic rates $\varrho = \{\vec{\rho}_a\}_{a\in\mathcal{A}}$ as $J(\varrho) = \sum_{a\in\mathcal{A}} \langle\vec{\gamma}_a,\vec{\rho}_a\rangle$.

**Theorem 4.2.** *Let $s : \mathcal{X} \times \mathcal{A} \to [0,1]$ be a fixed probabilistic predictor and let $\mathcal{D}_{\mathrm{post}}$ be an i.i.d. post-processing sample with group and group–label specific sample sizes $n_a$ and $n_{a,y}$, respectively, with $n_{\min} := \min_{a,y} n_{a,y}$ for $a \in \mathcal{A}$ and $y \in \{0,1\}$. Also, let $\hat{\varrho} = \{\widehat{\vec{\rho}_a}\}_{a\in\mathcal{A}}$ be the lifted operating characteristics returned by our empirical region search procedure (i.e., Algorithm 2 in §A.2.1) with grid steps $h_k$ for each linear-fractional centroid and with $\varepsilon_k$ as the denominator guards. For $\delta > 0$, define for $j \in \{0,1\}$,*

$$\eta_{a,j} = \sqrt{\frac{1}{2n_{a,j}} \log\frac{4m}{\delta}} + C \cdot \sqrt{\frac{1}{2n_a} \log\frac{4mK}{\delta}}, \tag{10}$$

*where $C > 0$ is a universal constant that depends only on $\{u_{k,a}, v_{k,a}\}_{a\in\mathcal{A}, k\in\mathcal{K}}$, but not on $n_{a,j}, m, K$, or $\delta$. Then, with probability at least $1 - \delta$,*

$$\left| J(\hat{\varrho}) - J(\varrho^\star) \right| \lesssim \sum_{a\in\mathcal{A}} (\eta_{a,1} + \eta_{a,0}) + \sum_{k\in K_{\mathrm{LF}}} h_k + \max_{k\in K_{\mathrm{LF}}} \varepsilon_k,$$

---

[6]We neither require the upstream probabilistic predictor $s(x,a)$ to be well-calibrated nor close to the Bayes-optimal regression function $\Pr(Y = 1|X = x, A = a)$.

*and for every fairness metric $k \in K_{\mathrm{LF}} \cup K_{\mathrm{L}}$,*

$$\max_{a,a'} \left| G_{k,a}(\widehat{\rho}_a) - G_{k,a'}(\widehat{\rho}_{a'}) \right| \lesssim \delta_k + \max_a (\eta_{a,1} + \eta_{a,0}).$$

Thus, if $n_{\min} \gtrsim \varepsilon^{-2} \log(mK/\delta)$ and $\sum_k h_k \lesssim \varepsilon$, then both the fairness slack and the excess risk are $O(\varepsilon)$. Furthermore, in the (idealized) special case where the probabilistic predictor coincides with the Bayes-optimal regression function, Theorem 4.2 implies that our region search procedure achieves the canonical $\tilde{O}(1/\sqrt{n})$ parametric rate of convergence to the Bayes-optimal operating characteristics, both in risk minimization and fairness attainment.

In brief, the proof relies on uniform convergence of the empirical ROC hulls via the DKW inequality (see, e.g., Dvoretzky et al., 1956; Massart, 1990, and Lemma A.7) and Lipschitz control of both linear and linear-fractional fairness constraints (Lemma A.8); for space considerations, the full proof is deferred to §A.4.2.

## 4.3 Constructing classifiers

We now seek to construct classifiers that achieve the optimal rates. Since there are many classifiers that can achieve any operating characteristic, the practitioner can impose additional desiderata such as minimizing the expected number of interventions, which we define as label flips from the base classifier $f^{(0)}$ (see §A.3.1). Here, we propose two versions of an approach for this problem.

### 4.3.1 Randomization procedure

Take the base post-processor $f^{(0)}$ to be a mixed-GWTR (see equation 4) whose operating point lies on the boundary of the convex hull of the empirical ROC curve, i.e., for all $x, a$:

$$f^{(0)}(x,a) = (1 - \theta_a) \mathbb{1}\left(s(x,a) \geq t_{h,a}\right) + \theta_a \mathbb{1}\left(s(x,a) \geq t_{h+1,a}\right), \qquad \theta_a \in [0,1], \tag{11}$$

where $t_{h+1,a} \geq t_{h,a}$, $t_{h,a}, t_{h+1,a} \in \widehat{\mathcal{H}}_a$ are two adjacent points on the support of the empirical convex hull[7] (cf. §A.2.1). Let, for all $x, a$, $q_a(x) := \Pr(f^{(0)} = 1 \mid X = x, A = a)$ be the positive prediction rate of the base post-processor, and denote the operating characteristics of $f^{(0)}$ as $\mathrm{TPR}_a^{(0)} := \Pr(f^{(0)} = 1 \mid Y = 1, A = a)$, $\mathrm{FPR}_a^{(0)} := \Pr(f^{(0)} = 1 \mid Y = 0, A = a)$, $\mathrm{FNR}^{(0)} = 1 - \mathrm{TPR}^{(0)}$, $\mathrm{TNR}^{(0)} = 1 - \mathrm{FPR}^{(0)}$. The group-wise optimal operating characteristics are fixed at values $(\widetilde{\mathrm{TPR}}_a, \widetilde{\mathrm{FPR}}_a)$, e.g., outputs from Algorithm 3.

Then the final post-processor $\widetilde{f}$ randomizes $f^{(0)}$ aiming to "regularize" or "shrink" $q_a$ towards a classifier that does not depend on the inputs $x$. This transformation also shifts the operating characteristics of $f^{(0)}$ towards the non-random classifier. There are several possible ways to implement this transformation, and here we discuss two.

`LabelFlipping` flips the mixed-GWTR labels with outcome-dependent probabilities $\widetilde{p}_{a,y} := \Pr(\widetilde{f} = 1 \mid A = a, f^{(0)} = y)$, $y \in \{0,1\}$. The final prediction is distributed as a Bernoulli random variable with success probability

$$\Pr\left(\widetilde{f}(x,a) = 1 \mid X = x, A = a\right) = \widetilde{p}_{a,0}\left(1 - q_a(x)\right) + \widetilde{p}_{a,1} q_a(x). \tag{12}$$

This induces a linear map on the operating characteristics:

$$\widetilde{\mathrm{TPR}}_a = \widetilde{p}_{a,1}\mathrm{TPR}_a^{(0)} + \widetilde{p}_{a,0}\left(1 - \mathrm{TPR}_a^{(0)}\right), \quad \widetilde{\mathrm{FPR}}_a = \widetilde{p}_{a,1}\mathrm{FPR}_a^{(0)} + \widetilde{p}_{a,0}\left(1 - \mathrm{FPR}_a^{(0)}\right). \tag{13}$$

Fixing the baseline post-processor and the target operating points uniquely determines the free parameters of the randomization procedure (i.e., the flipping probabilities $\widetilde{p}_{a,y}$ for `LabelFlipping`).

Geometrically, the set of operating characteristics attainable by equation 13 is the triangle spanned by the baseline hull point $(\mathrm{TPR}^{(0)}, \mathrm{FPR}^{(0)})$ and the trivial classifiers $(\mathrm{tpr}, \mathrm{fpr}) = (1,1)$ and $(0,0)$ (equivalently $(0,1)$ and $(1,0)$ in the $(\mathrm{FNR}, \mathrm{FPR})$-plane).

---

[7]Since $\widehat{\mathcal{H}}_a$ is constructed solely from a probabilistic predictor and a post-processing set, and we add randomization implicit in the mixing parameter $\theta_a$, $f^{(0)}$ is indeed a valid post-processor (cf. equation 5).

Since $f^{(0)}$ is a mixed-GWTR as in equation 11, its operating point along any hull edge can be parameterized by a single mixing parameter $\theta_a \in [0, 1]$ between adjacent supports $(h, h+1)$:

$$\text{FNR}_{a,\theta}^{(0)} = (1 - \theta_a)\text{FNR}_a^{(h)} + \theta_a\text{FNR}_a^{(h+1)}, \text{FPR}_{a,\theta}^{(0)} = (1 - \theta_a)\text{FPR}_a^{(h)} + \theta_a\text{FPR}_a^{(h+1)}. \quad (14)$$

Thus, the linear map from equation 13 gives the $2 \times 2$ system

$$\begin{bmatrix} \widetilde{\text{FPR}}_a \\ 1 - \widetilde{\text{FNR}}_a \end{bmatrix} = \begin{bmatrix} \text{FPR}_{a,\theta}^{(0)} & 1 - \text{FPR}_{a,\theta}^{(0)} \\ 1 - \text{FNR}_{a,\theta}^{(0)} & \text{FNR}_{a,\theta}^{(0)} \end{bmatrix} \begin{bmatrix} p_{a,1}(\theta) \\ p_{a,0}(\theta) \end{bmatrix}, \quad (15)$$

whose determinant is $\det(\theta) = \text{FPR}_{a,\theta}^{(0)} + \text{FNR}_{a,\theta}^{(0)} - 1$, which is zero only when the post-processor has a degenerate ROC curve. Consequently, given a target operating characteristic $\left( \widetilde{\text{FNR}}_a, \widetilde{\text{FPR}}_a \right)$ and fixed mixing parameter $\theta_a$, the parameters of randomization are uniquely determined by the rate-matching equations when $\det(\theta) \neq 0$ with feasibility requiring $0 \leq p_{a,0}(\theta), p_{a,1}(\theta) \leq 1$. [8] We present the details of minimizing interventions with this label flipping scheme in Section A.3.2.

*Remark* 4.3. While Hardt et al. (2016) considered label flipping and threshold search in some special cases, they did these separately; which limits the set of operating points their classifier can attain. More recently, Hsu et al. (2022) used label flipping randomization on top of GWTRs to simultaneously satisfy multiple fairness constraints. However, their procedure implicitly fixes the group-wise thresholds at the medians of the group-specific probabilistic predictor scores. These thresholds are overly restrictive, as the set of operating characteristics attainable from label flipping may not include the optimal operating point (see §5.1).

### 4.4 MULTIPLE PROTECTED ATTRIBUTES & MULTIPLE CLASSES

Our method directly supports multiple protected attributes and can also be generalized to handle multiple classes, following common definitions of multiclass group fairness that generalize the binary class setting (Alghamdi et al., 2022; Xian & Zhao, 2024, see, e.g.,); see §A.3.3 for details.

## 5 EXPERIMENTAL RESULTS

We evaluate our method on standard empirical datasets, aiming to simultaneously satisfy demographic parity (DP); equality of opportunity/TPR parity (EOpp), see Hardt et al. (2016); predictive equality/FPR parity (PEq), see Corbett-Davies et al. (2017); predictive parity (PP), see Chouldechova (2017); and possibly false omission rate parity (FOR-parity), see Barocas et al. (2023).[9] while maximizing accuracy. Enforcing both $\delta_{\text{EOpp}}$ and $\delta_{\text{PEq}}$-approximate fairness is equivalent to enforcing $\max\{\delta_{\text{EOpp}}, \delta_{\text{PEq}}\} = \delta_{\text{EO}}$-approximate fairness for equalized odds (EO), see Hardt et al. (2016).

Due to space constraints, we defer full implementation details, training hyperparameters, and exact baseline settings to the appendix §A.5. We use the COMPAS (Larson & Angwin, 2016), Lawschool (Wightman, 1998; Fabris et al., 2022), BiasBios (De-Arteaga et al., 2019; Ravfogel et al., 2020), and ACSIncome (Ding et al., 2021) datasets, with a TRAIN/POST/TEST= $30/35/35$ split: TRAIN fits $s$ (using a three-layer neural net), and POST fits all post-processors. We report accuracy and five fairness metrics (DP, EOpp, PEq, PP, FOR-parity), aggregating mean results over 50 random seeds, along with standard deviations.

**Baselines** We compare with two post-processing methods that seek to simultaneously control LF fairness constraints (META (Celis et al., 2019) and MFOpt (Hsu et al., 2022)), and one state-of-the-art post-processing method that controls for linear fairness constraints (LPP, (Xian & Zhao, 2024)). [10] In addition, we record the performance of the unconstrained probabilistic classifier

---

[8]Although we focused on fairness constraints expressible as pairwise differences in LF/L functionals, our procedure for constructing classifiers provides recipes to match any given target operating characteristics that lie in the convex hull of the empirical ROC curve of the base probabilistic predictor $s$, subject to minimizing the expected number of interventions (Algorithm 5).

[9]The first three are linear constraints, while the latter two are LF constraints (see §3).

[10]We omit comparisons with popular in-processing algorithms (e.g., Agarwal et al., 2018; Zhang et al., 2018b) since Xian & Zhao (2024) demonstrate their method compares favorably against state-of-the-art in-processing algorithms; and, in-processing methods generally do not support controlling for the linear-fractional constraints we consider here.

Table 2: Performance on the test set for (A) COMPAS ($|\mathcal{A}|$=2) and (B) ACSIncome ($|\mathcal{A}|$=5). The disparities $\delta_{DP}, \delta_{EOpp}, \delta_{PEq}, \delta_{PP}$ are controlled at level 0.05 whenever they are active. **Interv.** is the empirical intervention rate on the test set (see Definition A.3). Cells in green indicate that the fairness constraint is satisfied within two standard deviations at the nominal level, whereas cells in red indicate violation. Entries in the Oracle rows are shaded in lighter colors to denote that they are not practically feasible baselines.

| | | | (A) COMPAS ($|\mathcal{A}|$=2) | | | | |
|---|---|---|---|---|---|---|---|
| **Method** | **Acc** | **DP** | **EOpp** | **PEq** | **PP** | **FOR** | **Interv.** |
| Baseline | $0.68 \pm 0.01$ | $0.28 \pm 0.05$ | $0.27 \pm 0.05$ | $0.19 \pm 0.05$ | $0.07 \pm 0.04$ | $0.03 \pm 0.02$ | $0.00 \pm 0.00$ |
| Oracle | $0.62 \pm 0.02$ | $0.04 \pm 0.01$ | $0.03 \pm 0.02$ | $0.05 \pm 0.01$ | $0.05 \pm 0.00$ | $0.16 \pm 0.02$ | N/A |
| **ROCF-LF** (ours) | $0.61 \pm 0.02$ | $0.05 \pm 0.03$ | $0.03 \pm 0.03$ | $0.05 \pm 0.03$ | $0.07 \pm 0.04$ | $0.15 \pm 0.02$ | $0.06 \pm 0.03$ |
| MFOpt | $0.63 \pm 0.01$ | $0.26 \pm 0.04$ | $0.25 \pm 0.04$ | $0.21 \pm 0.04$ | $0.08 \pm 0.04$ | $0.07 \pm 0.03$ | $0.13 \pm 0.02$ |
| META | $0.50 \pm 0.02$ | $0.05 \pm 0.17$ | $0.05 \pm 0.16$ | $0.04 \pm 0.18$ | $0.06 \pm 0.19$ | $0.15 \pm 0.05$ | $0.00 \pm 0.00$ |
| LPP-DP | $0.67 \pm 0.01$ | $0.06 \pm 0.03$ | $0.04 \pm 0.03$ | $0.03 \pm 0.03$ | $0.15 \pm 0.03$ | $0.09 \pm 0.03$ | $0.00 \pm 0.00$ |
| LPP-EO | $0.67 \pm 0.01$ | $0.10 \pm 0.04$ | $0.09 \pm 0.04$ | $0.04 \pm 0.03$ | $0.14 \pm 0.03$ | $0.08 \pm 0.03$ | $0.00 \pm 0.00$ |

| | | | (B) ACSIncome ($|\mathcal{A}|$=5) | | | | |
|---|---|---|---|---|---|---|---|
| **Method** | **Acc** | **DP** | **EOpp** | **PEq** | **PP** | **FOR** | **Interv.** |
| Baseline | $0.79 \pm 0.00$ | $0.25 \pm 0.01$ | $0.24 \pm 0.02$ | $0.09 \pm 0.02$ | $0.23 \pm 0.02$ | $0.08 \pm 0.01$ | $0.00 \pm 0.00$ |
| Oracle | $0.69 \pm 0.01$ | $0.05 \pm 0.00$ | $0.05 \pm 0.00$ | $0.03 \pm 0.01$ | $0.05 \pm 0.00$ | $0.23 \pm 0.01$ | N/A |
| **ROCF-LF** (ours) | $0.69 \pm 0.01$ | $0.05 \pm 0.01$ | $0.05 \pm 0.01$ | $0.03 \pm 0.01$ | $0.07 \pm 0.03$ | $0.23 \pm 0.01$ | $0.03 \pm 0.01$ |
| LPP-DP | $0.78 \pm 0.00$ | $0.06 \pm 0.01$ | $0.07 \pm 0.01$ | $0.09 \pm 0.01$ | $0.35 \pm 0.01$ | $0.15 \pm 0.01$ | $0.00 \pm 0.00$ |
| LPP-EO | $0.78 \pm 0.00$ | $0.12 \pm 0.01$ | $0.06 \pm 0.02$ | $0.06 \pm 0.01$ | $0.33 \pm 0.01$ | $0.14 \pm 0.01$ | $0.00 \pm 0.00$ |

$s$ (`Baseline`) and an infeasible oracle post-processor with operating characteristics returned by `RegionSearch-FG` (Algorithm 3) using the true labels of the TEST set as an upper bound.

**Results.** We present the primary results here in the main text and leave additional experimental results to the appendix (see Tables 5 and 6 in §A.6.1, and §A.6.3)

## 5.1 APPROXIMATE FAIRNESS FOR DP, EOPP, PEQ, AND PP

We impose *four* constraints (DP, EOpp, PEq, and PP) simultaneously, controlling for $\vec{\delta}$-approximate fairness at either the level $\delta_i$=0.05 (for COMPAS and ACSIncome) or $\delta_i$=0.02 (for Lawschool) across all $i$. Results are shown in Tables 2 and 5, where our method ROCF-LF refers to the LabelFlipping minimimum intervention algorithm (cf. Algorithm 5).

**Results.** First, we observe that our method controls all four disparity metrics. Among the baselines, only META achieves this on the COMPAS dataset, and no other baseline achieves it on the other datasets. However, META exhibits a much larger accuracy drop than our method. The reason for the performance gap might be that they aim to optimize in the space of classifiers, whereas our method achieves an advantage by working directly with the operating characteristics. Moreover, on both examples, we observe that our method has accuracy close to the oracle.

**Inevitable trade-offs.** Since even the oracle method exhibits an accuracy drop, this suggests that fairness/accuracy trade-offs when imposing all four constraints might be unavoidable in our examples. This is in contrast with simpler settings where we only impose fewer constraints Xian & Zhao (2024); Celis et al. (2019); Baumann et al. (2022).

**Interventions.** We also observe that the number of interventions is small, around 6% for COMPAS, 1% for Lawschool, 0.5% for BiasBios, and 3% for ACSIncome for our method.

**Scalability.** Our method scales well to the much larger ACSIncome dataset, demonstrating both practical computational runtimes (see §A.6.2) and attaining the nominal disparity levels across all fairness constraints. The latter is to be expected, since our method can better approximate the population level feasibility region and the realizable ROC region $\mathcal{R}_a(s)$ (cf. equation 6) with a better trained probabilistic predictor and a larger post-processing set.

## 5.2 TWO LINEAR FRACTIONAL CONSTRAINTS: ADDING FALSE OMISSION RATE PARITY

**False omission rate parity.** We consider satisfying an additional linear fractional constraint, the *false omission rate parity* (FOR-parity), along with two linear constraints. For these experiments,

Table 3: Performance on the test set for (A) Lawschool ($|\mathcal{A}|$=2) and (B) ACSIncome ($|\mathcal{A}|$=5). The same protocol as in Table 2 is used, except $\delta_{\text{EOpp}}$, $\delta_{\text{PP}}$ and $\delta_{\text{FOR}}$ are controlled at either level 0.03 (for Lawschool) or 0.10 (for ACSIncome).

| Method | Acc | DP | EOpp | PEq | PP | FOR | Interv. |
|---|---|---|---|---|---|---|---|
| **(A) Lawschool ($|\mathcal{A}|$=2)** | | | | | | | |
| Baseline | $0.79 \pm 0.00$ | $0.07 \pm 0.01$ | $0.11 \pm 0.02$ | $0.03 \pm 0.01$ | $0.04 \pm 0.02$ | $0.04 \pm 0.00$ | $0.00 \pm 0.00$ |
| Oracle | $0.57 \pm 0.03$ | $0.16 \pm 0.02$ | $0.03 \pm 0.00$ | $0.16 \pm 0.03$ | $0.02 \pm 0.01$ | $0.03 \pm 0.00$ | N/A |
| **ROCF-LF** (ours) | $0.58 \pm 0.04$ | $0.15 \pm 0.02$ | $0.03 \pm 0.01$ | $0.16 \pm 0.03$ | $0.02 \pm 0.01$ | $0.03 \pm 0.01$ | $0.01 \pm 0.01$ |
| LPP-EOpp | $0.79 \pm 0.00$ | $0.04 \pm 0.01$ | $0.04 \pm 0.02$ | $0.01 \pm 0.00$ | $0.09 \pm 0.01$ | $0.05 \pm 0.00$ | $0.00 \pm 0.00$ |
| **(B) ACSIncome ($|\mathcal{A}|$=5)** | | | | | | | |
| Baseline | $0.78 \pm 0.01$ | $0.24 \pm 0.01$ | $0.25 \pm 0.03$ | $0.09 \pm 0.02$ | $0.23 \pm 0.03$ | $0.08 \pm 0.01$ | $0.00 \pm 0.00$ |
| Oracle | $0.57 \pm 0.04$ | $0.48 \pm 0.05$ | $0.10 \pm 0.01$ | $0.57 \pm 0.08$ | $0.10 \pm 0.01$ | $0.10 \pm 0.01$ | N/A |
| **ROCF-LF** (ours) | $0.56 \pm 0.02$ | $0.49 \pm 0.02$ | $0.11 \pm 0.01$ | $0.59 \pm 0.04$ | $0.11 \pm 0.01$ | $0.11 \pm 0.03$ | $0.05 \pm 0.01$ |
| LPP-EOpp | $0.79 \pm 0.00$ | $0.15 \pm 0.01$ | $0.11 \pm 0.01$ | $0.06 \pm 0.01$ | $0.31 \pm 0.01$ | $0.12 \pm 0.01$ | $0.00 \pm 0.00$ |

we increase the tolerance level to either $\vec{\delta}$=0.10 (for COMPAS and ACSIncome), $\vec{\delta}$=0.05 (for BiasBios), or $\vec{\delta}$=0.03 (for Lawschool) and exclude PEq and either DP or EOpp, since we often find that the empirical feasibility region $\widehat{\mathfrak{F}}(s, \vec{\delta}; \mathcal{D}_{\text{post}})$ (see Remark A.1) is empty when controlling at a smaller $\vec{\delta}$ level (cf.§ 5.1). Indeed, additionally satisfying a low level of FOR-parity is substantively nontrivial: Majumder et al. (2023) show through extensive empirical studies that fairness metrics cluster differently, with FOR-parity separating from DP/EO and PP.

**Methods compared.** Existing baselines do not explicitly support this configuration out of the box: Hsu et al. (2022) only handles $\vec{\delta}$-approximate DP, EO, and PP, and, while Celis et al. (2019) can, in principle, encode any combination of LF/L fairness constraints, their released code does not implement FOR-parity (Keswani et al., 2019). Consequently, we report results only for our method (ROCF-LF), the optimal operating point (Oracle), and the linear post-processing method of Xian & Zhao (2024) that only controls for demographic parity / equality of opportunity (LPP-DP/EOpp). The latter is included merely as a strong recent baseline from the literature, and we emphasize that it does not aim to ensure predictive parity and false omission rate parity.

**Results.** Table 3 reports results for the Lawschool with binary protected attributes and AC-SINCOME dataset with multiple protected attributes. As we can see, our method controls all three fairness constraints at the desired level, while achieving a near-oracle accuracy. This reinforces the effectiveness of our method. In contrast, the LPP-EOpp method does not control PP and/or FOR-parity; which is reasonable as it does not aim to do so. Moreover, we observe a substantial accuracy-fairness tradeoff of $21-22$ percent across both datasets. This suggests that satisfying fairness across multiple protected attributes for multiple LF constraints is difficult (Table 3).

**Sensitivity Analysis.** We also conduct a sensitivity analysis on the granularity of the grid search used for the linear-fractional constraints (cf.§4.2). We observe that using a grid size of 25 or 50 achieves near-oracle level performance and often attains the nominal disparity levels, while exhibiting a nearly $5 - 20$ fold decrease in computational time compared to the full grid search with 100 points; for space considerations, full results are deferred to §A.6.3.

## 6 DISCUSSION

In this paper, we proposed an approach for fair classification with linear fractional approximate fairness constraints, which relied on reformulating the goal in the space of the operating characteristics of the classifiers and then designing fairness interventions at that level. We observed that our method compares favorably to existing baselines in experiments. An important direction for future work is that the proposed randomization approaches may randomize any individuals, but it could be of interest to design intervention policies that restrict randomization to only a subpopulation.

## REPRODUCIBILITY STATEMENT

All experimental details, including dataset information, preprocessing, and evaluation protocols, are provided in Section 5, Appendix A.5, and Appendix A.6. An anonymous GitHub repository, containing the implementation of our ROCF method, baseline methods, and code to reproduce all experiments, is available at this repository. All theoretical results and assumptions are stated in Sections 3 and 4, with complete proofs provided in Appendix A.4.

## ETHICS STATEMENT

Our work falls in the broad area of ethical machine learning. However, our contribution is purely technical and does not make any specific recommendations that might have ethical implications. Instead, our contribution is to develop powerful algorithms that enable practitioners to achieve algorithmic fairness in contexts that were not possible due to technical constraints in the past; by efficiently incorporating multiple group fairness constraints that were previously hard to simultaneously satisfy.

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

## A    APPENDIX

*Usage of LLMs.* Open-AI's ChatGPT 5-thinking model helped with the code implementation of the algorithm `ROCF-LF`, `MFOpt`, and `META`, and with viewing / summarizing the raw result files; it was also used to help format equations and tables during the writing process.

### A.1 ADDITIONAL RELATED WORK

Recent work by Zehlike et al. (2025) interpolates only between accuracy equality, false negative and false positive rate parity, leveraging optimal transport. Theoretical work by Hamman & Dutta (2024) studies the tradeoff between demographic parity, equalized odds, and predictive parity using partial information decomposition by fixing one of the fairness metrics and exploring the tradeoffs between the other two, without explicitly designing classifiers. Intervening on ROC curves has also been used to ensure forms of fairness different from the ones that we consider in this paper (Vummintala et al., 2025).

Beyond our focus on post-processing methods, there are also pre- and in-processing methods that seek to modify the training process or dataset. Some of these aim to simultaneously control a broad set of disparity measures by removing the dependence of the protected attribute $A$ on $(X, Y)$ (Zhang et al., 2018a; Bothmann et al., 2023; Plečko et al., 2024; Leininger et al., 2025).

Post-processing is attractive when retraining is impractical or constrained by governance, cost, or latency—conditions that have become increasingly common with generative AI (Hu et al., 2022; Dettmers et al., 2023; Wang et al., 2025) and with deployed models in regulated domains (e.g., credit, hiring, healthcare; see Quell et al. (2021); Groves et al. (2024); Dupreez & McDermott (2025)).

### A.2 ADDITIONAL METHODOLOGICAL DETAILS

#### A.2.1 EMPIRICAL REGION SEARCH OVER ROC-HULL SUPPORTS

We instantiate the empirical analog of equation 7 by searching over linear fractional centroids for the best post-processing operating points from each group's empirical ROC convex hull. More formally, let the plug-in estimates of the operating characteristics in equation 1 on the postprocessing set $\mathcal{D}_{\text{post}}$ be

$$n_{a,1} = \sum_{i=1}^{N} 1\left(a_i = a, y_i = 1\right), \qquad n_{a,0} = \sum_{i=1}^{N} 1\left(a_i = a, y_i = 0\right), \qquad n_a = n_{a,1} + n_{a,0},$$

$$\widehat{\text{TPR}}_a(f) = \frac{1}{n_{a,1}} \sum_{i=1}^{N} 1\left(a_i = a, y_i = 1\right) \cdot 1\left(f(x_i, a_i) = 1\right),$$

$$\widehat{\text{FPR}}_a(f) = \frac{1}{n_{a,0}} \sum_{i=1}^{N} 1\left(a_i = a, y_i = 0\right) \cdot 1\left(f(x_i, a_i) = 1\right),$$

$$\widehat{\text{FNR}}_a(f) = 1 - \widehat{\text{TPR}}_a(f), \qquad \widehat{\text{TNR}}_a(f) = 1 - \widehat{\text{FPR}}_a(f). \quad (16)$$

Then, for each $a \in \mathcal{A}$, we form the group-wise empirical ROC $\{(\widehat{\text{TPR}}_a^{(j)}, \widehat{\text{FPR}}_a^{(j)})\}_{j \in [n_a]}$ from the probabilistic predictor scores $s(x_i, a_i)$ and labels $y_i$ on the post-processing set by setting

$$f^{(j)}(x, a) := 1\left(\text{rank}_a\left(s(x, a)\right) \le j\right), \qquad j \in [n_a]$$

in equation 16 where[11] $\text{rank}_a\left(s(x, a)\right) := 1 + \sum_{i=1}^{N} 1\left(a_i = a, s(x_i, a_i) > s(x, a)\right)$. We then retain the upper convex hull vertices $\widehat{\mathcal{H}}_a = \{(\widehat{\text{TPR}}_a^{(j)}, \widehat{\text{FPR}}_a^{(j)})\}_{j \in S_a}$ for some index set $S_a \subseteq [n_a]$ (e.g., using Andrew's monotone chain algorithm (Andrew, 1979; o'Rourke, 1998)), and construct lifted operating characteristic vectors $\widehat{r}_a^{(j)} = (\widehat{\text{TPR}}_a^{(j)}, \widehat{\text{FPR}}_a^{(j)}, 1)^{\top}$ from $\widehat{\mathcal{H}}_a$. The set of all lifted operating characteristics vectors is termed the *empirical ROC (convex) hull* $\widehat{R}_a(s)$.

Since any operating characteristic in $\widehat{R}_a(s)$ is as a linear combination of the convex hull support points, we enforce post-processing by introducing convex weights $\{\lambda_{a,j}\}_{j \in S_a}$ that satisfy

$$\sum_{j \in S_a} \lambda_{a,j} = 1, \quad \lambda_{a,j} \ge 0, \qquad \widehat{\rho}_a = \sum_{j \in S_a} \lambda_{a,j} \cdot \widehat{r}_a^{(j)}, \quad (17)$$

---

[11] If scores are tied, then ties can be broken by a deterministic rule (e.g., via lexicographic ordering) so that exactly $j$ group-$a$ points satisfy $f^{(j)}(x_i, a_i) = 1$.

where $\widehat{\rho}_a$ are the (lifted) operating points $(\widehat{\mathrm{TPR}}_a, \widehat{\mathrm{FPR}}_a, 1)^\top$.

Finally, we form empirical plug-in coefficients for $\widehat{\gamma}_a, \widehat{u}_{k,a}, \widehat{v}_{k,a}$ (e.g., replace $\pi_a$ by $\widehat{\pi}_a$). The resulting optimization problem and full procedure for finding the optimal operating points is presented in Algorithm 2 (`RegionSearch`).

*Remark* A.1. The region of operating points specified by the constraints of the inner LP (i.e., the linear fractional/linear fairness constraints and post-processing constraint) in Algorithm 2 is the empirical analog of the centroid-specific population-level feasibility region $\mathfrak{F}(\vec{q}; s, \vec{\delta})$ discussed in §4. We call the union of this centroid specific region over the set of admissible centroids (cf. §4.2) the *empirical feasibility region* $\widehat{\mathfrak{F}}(s, \vec{\delta}; \mathcal{D}_{\mathrm{post}})$.

*Remark* A.2. Each inner problem is a modest-sized LP with $\sum_{a\in\mathcal{A}} |S_a| + |\mathcal{K}_{\mathrm{L}}|$ decision variables. The number of linear constraints is often smaller: $|\mathcal{A}| + 1$ for the simplex and post-processing per group, $2|\mathcal{A}||\mathcal{K}_{\mathrm{L}}|$ for the linear centroid bands, $2|\mathcal{A}||\mathcal{K}_{\mathrm{LF}}|$ for the LF bands, and $|\mathcal{A}||\mathcal{K}_{\mathrm{LF}}|$ for the denominator checks.

Since the hull support sizes $|S_a|$ are typically small (see e.g., Groeneboom, 2021), each LP solve is fast in practice, so that the overall runtime is dominated by the low-dimensional outer search over linear fractional centroids. We report computational runtimes in §A.6.2.

**A feasibility guard.** To ensure that we produce a classifier even when `RegionSearch` does not yield a feasible operating point, one can wrap expansion policies around `RegionSearch` procedure of Algorithm 2 that increase the nominal disparity levels until a feasible point is found. Here, we introduce two simple expansion policies that are intuitive and that we find have worked well in practice.

$\alpha$-*expansion policy:* On a high level, if Algorithm 2 is infeasible with the initial, user-specified tolerances, we re-solve Algorithm 2 with fairness tolerances that are uniformly relaxed by an expansion factor of $\alpha$. By performing a bisection search on an appropriately designed interval for $\alpha$, our `RegionSearch` is guaranteed to produce a feasible operating point, from which we can construct a classifier (per §4.3).

More formally, given user-specified tolerances $\vec{\delta}=\{\delta_k\}_{k\in\mathcal{K}_{\mathrm{L}}\cup\mathcal{K}_{\mathrm{LF}}}$, define $\vec{\delta}(\alpha) := \{\alpha\delta_k\}_{k\in\mathcal{K}_{\mathrm{L}}\cup\mathcal{K}_{\mathrm{LF}}}$ for $\alpha \geq 1$. Let the unconstrained/baseline per-group operating points be, for $a \in \mathcal{A}$, $j_a^\star \in \arg\min_{j\in S_a} \left\langle \widehat{\gamma}_a, \widehat{r}_a^{(j)} \right\rangle$ and $\widehat{\rho}_a^{\mathrm{base}} := \widehat{r}_a^{(j_a^\star)}$. These are the solution of `RegionSearch` with no fairness constraints.[12] The corresponding baseline group performance functions are then

$$g_{\ell,a}^{\mathrm{base}} := \left\langle \widehat{u}_{\ell,a}, \widehat{\rho}_a^{\mathrm{base}} \right\rangle, \qquad r_{k,a}^{\mathrm{base}} := \frac{\left\langle \widehat{u}_{k,a}, \widehat{\rho}_a^{\mathrm{base}} \right\rangle}{\left\langle \widehat{v}_{k,a}, \widehat{\rho}_a^{\mathrm{base}} \right\rangle} \quad \text{(whenever } \langle \widehat{v}_{k,a}, \widehat{\rho}_a^{\mathrm{base}} \rangle \geq \varepsilon_k\text{)},$$

with baseline disparity measures $\Delta_\ell := \max_a g_{\ell,a}^{\mathrm{base}} - \min_a g_{\ell,a}^{\mathrm{base}}$; $\Delta_k := \max_a r_{k,a}^{\mathrm{base}} - \min_a r_{k,a}^{\mathrm{base}}$.

Now, note that we can recover the baseline operating points $\widehat{\rho}_a^{\mathrm{base}}$ by re-solving `RegionSearch` with $\vec{\delta}(\alpha_{\mathrm{hi}})$, where $\alpha_{\mathrm{hi}}$ is the maximum ratio between the baseline disparity measures and the nominal disparity levels. That is, an upper bound for the expansion parameter is $\alpha_{\mathrm{hi}} := \max\left\{1, \max_{i\in\mathcal{K}_{\mathrm{L}}\cup\mathcal{K}_{\mathrm{LF}}} \frac{\Delta_i}{\delta_i}\right\}$.

By design, $\{\widehat{\rho}_a^{\mathrm{base}}\}_{a\in\mathcal{A}}$ is feasible at $\alpha = \alpha_{\mathrm{hi}}$, and feasibility is monotone in $\alpha$: if the inner LP of `RegionSearch` is feasible at $\alpha$, it remains feasible for any $\alpha' \geq \alpha$. We therefore bisect on $\alpha \in [\alpha_{\mathrm{lo}}, \alpha_{\mathrm{hi}}] = [1, \alpha_{\mathrm{hi}}]$ by setting, at each step, $\alpha \leftarrow (\alpha_{\mathrm{lo}} + \alpha_{\mathrm{hi}})/2$, re-solving Algorithm 2 with $\vec{\delta}(\alpha)$, and updating the search interval appropriately. For completeness, the full wrapper algorithm is provided in Algorithm 3 (`RegionSearch-FG`).

This feasibility guard produces a classifier while minimally relaxing the user-specified tolerances. In our experiments, it rarely activates; when it does, the expansion parameter is relatively small (e.g., $\alpha \approx 1.04$ and see §A.6.1).

$\Delta$-*incremental policy.* While the $\alpha$-expansion policy uniformly scales all fairness tolerances, it can be useful in practice to selectively relax only a subset of constraints. To support this use case, we

---

[12]The objective is linear in the operating characteristics, so an optimum $\widehat{\rho}_a^{\mathrm{base}}$ occurs at a hull vertex.

introduce an alternative expansion policy that enlarges the nominal disparities for a chosen pair of constraints until feasibility is restored.

For example, suppose the user wishes to preserve the nominal disparity levels for all constraints except two: a linear constraint $k \in \mathcal{K}_{\mathrm{L}}$ (e.g., equality of opportunity) and a linear–fractional constraint $j \in \mathcal{K}_{\mathrm{LF}}$ (e.g., predictive Parity). Let their user-specified tolerances be $\delta_k$ and $\delta_j$. If `RegionSearch` is infeasible under these nominal tolerances, we generate a sequence of *incrementally relaxed* tolerances

$$\delta_k^{(m)} := \delta_k + m \cdot \eta_k, \qquad \delta_j^{(m)} := \delta_j + m \cdot \eta_j, \qquad m = 0, 1, 2, \ldots,$$

where $\eta_k > 0$ and $\eta_j > 0$ are user-selected step sizes that govern how quickly each constraint is relaxed. We allow $\eta_k$ and $\eta_j$ to differ (e.g., $\eta_k = 1e{-}2$ and $\eta_j = 5e{-}3$), thereby encoding *asymmetric/non-uniform preferences*—such as relaxing equality of opportunity twice as quickly as predictive parity.

At iteration $m$, we re-solve Algorithm 2 using the updated tolerances $(\delta_k^{(m)}, \delta_j^{(m)})$, while keeping all other fairness constraints fixed at their original user-specified levels. Feasibility is monotone in $m$: once Algorithm 2 becomes feasible for some $m^\star$, it remains feasible for all $m > m^\star$ because the constraints are only being relaxed. The procedure therefore terminates at the smallest $m^\star$ for which a feasible operating point is found.

We present preliminary results of this expansion policy in §A.6.4.

## A.3 ALGORITHMS

In this section, we provide the pseudo-code for several of our proposed algorithms. We summarize each of the algorithms here for clarity.

- `ROCF-Pipeline` (Algorithm 1): Top level procedure that first finds target operating characteristics using a pre-trained classifier $s$, post-processing set $\mathcal{D}_{\mathrm{post}}$, and user-specified tolerances $\vec{\delta}$ for LF/F fairness constraints (Algorithm 3). Then, it constructs a classifier to achieve those operating characteristics with the option of imposing additional desiderata (e.g., minimizing expected number of interventions; see Algorithms 4 and 5).

- `RegionSearch` (Algorithm 2): Empirical instantiation of the population level optimal fair classification problem (cf. equation 7); searches over each group's empirical ROC-hull support to select target operating characteristics. Handles LF constraints via a small outer search over admissible centroids (cf. §A.5.1).

- `RegionSearch-FG` (Algorithm 3): Wraps a feasibility guard around `RegionSearch` that guarantees feasibility by loosening the tolerance levels $\vec{\delta}$ when the initial search is infeasible.

- `ConstructClassifier` (Algorithm 4): Constructs a randomized post-processor with target operating characteristics, subject to additional desiderata.

- `MinIntervention` (Algorithm 5): Specific instantiation of Algorithm 4. Given target operating characteristics, this algorithm produces a mixed-GWTR (cf. Definition 3.2) with added randomization, to achieve the target characteristics while minimizing the expected intervention rate (cf. Definition A.3). Implements the LABELFLIPPING) randomization procedure discussed in the text.

### A.3.1 INTERVENTIONS

Using the output of Algorithm 3, we seek to construct classifiers that achieve target operating characteristics. Since there are many classifiers that can achieve any operating characteristic, the practitioner can impose additional desiderata such as minimizing the expected number of interventions (see Definition A.3 below). We provide one particular construction in this work, though we leave the full question of optimally designing these particular classifiers as future work, since this requires answering broader questions on a sociotechnical level.

Concretely, we provide a modular meta-algorithm that accepts any constructed classifier (Algorithm 4). For the additional desideratum of minimizing the expected number of interventions, we provide a

---

**Algorithm 1** (End-to-end ROCF pipeline)

---

**Require:** Training set $\mathcal{D}_{\text{train}}$, postprocessing set $\mathcal{D}_{\text{post}}$, test set $\mathcal{D}_{\text{test}}$; groups $\mathcal{A}$; tolerances $\{\delta_k\}$; constraint indices $\mathcal{K}_{\text{L}}, \mathcal{K}_{\text{LF}}$ with $\mathcal{K}_{\text{L}} \sqcup \mathcal{K}_{\text{LF}} = [K]$; centroid intervals $\{\mathcal{Q}_k\}_{k \in \mathcal{K}_{\text{LF}}}$; margins $\{\varepsilon_k > 0\}$.

1: **Pre-train predictor.** Fit a probabilistic predictor $s : \mathcal{X} \times \mathcal{A} \to [0, 1]$ on $\mathcal{D}_{\text{train}}$.
2: **Postprocess ROC curve.** On $\mathcal{D}_{\text{post}}$, compute probabilistic predictor scores $s(x_i, a_i)$ and retrieve labels $y_i$. For each $a \in \mathcal{A}$: form the empirical ROC curve and keep the upper convex hull supports $\widehat{\mathcal{H}}_a$; build lifted points $\widehat{r}_a^{(j)} = (\widehat{\text{TPR}}_a^{(j)}, \widehat{\text{FPR}}_a^{(j)}, 1)^\top$. Form plug-in coefficients $\widehat{\gamma}_a, \widehat{u}_{k,a}, \widehat{v}_{k,a}$.
3: **Search region (Algorithm 3).** Provide $\{\widehat{\mathcal{H}}_a\}, \{\widehat{r}_a^{(j)}\}, \{\widehat{\gamma}_a, \widehat{u}_{k,a}, \widehat{v}_{k,a}\}, \{\delta_k\}, \mathcal{K}_{\text{L}}, \mathcal{K}_{\text{LF}}, \{\mathcal{Q}_k\},$ $\{\varepsilon_k\}, \{\tau_\alpha\}$ to Algorithm 3; obtain target operating points $\{\widetilde{\rho}_a\}$.
4: **Construct classifier (Algorithm 4).** Using $\{\widetilde{\rho}_a\}$ and $s$, construct a classifier $\widehat{f}$ that attains $\widetilde{\rho}_a$ for each group (optionally with additional desiderata).
5: **Evaluate.** On $\mathcal{D}_{\text{test}}$, evaluate $\widehat{f}$ to report loss and fairness metrics $\max_{a,a' \in \mathcal{A}} |G_{k,a}(\widehat{f}) - G_{k,a'}(\widehat{f})|$ for $k \in [K]$.
6: **Return** $\{\widetilde{\rho}_a\}, \widehat{f}$, and test metrics.

---

---

**Algorithm 2** `RegionSearch`

---

**Require:** Groups $\mathcal{A}$; hull supports $\{\widehat{\mathcal{H}}_a\}$; plug-in $\widehat{\gamma}_a, \widehat{u}_{k,a}, \widehat{v}_{k,a}$; tolerances $\{\delta_k\}$; index sets $\mathcal{K}_{\text{L}}, \mathcal{K}_{\text{LF}}$; centroid intervals $\{\mathcal{Q}_k\}_{k \in \mathcal{K}_{\text{LF}}}$; margins $\{\varepsilon_k > 0\}$.

1: best $\leftarrow +\infty, \vec{q}_{\text{opt}} \leftarrow$ None, $\{\widetilde{\lambda}_{a,j}\} \leftarrow$ None, $\{\widetilde{\rho}_a\} \leftarrow$ None.
2: Construct $\widehat{r}_a^{(j)} = (\widehat{\text{TPR}}_a^{(j)}, \widehat{\text{FPR}}_a^{(j)}, 1)^\top$ from hull supports $\widehat{\mathcal{H}}_a = \{(\widehat{\text{TPR}}_a^{(j)}, \widehat{\text{FPR}}_a^{(j)})\}_{j \in S_a}$.
3: **for** $\vec{q} = (q_k)_{k \in \mathcal{K}_{\text{LF}}} \in \prod_{k \in \mathcal{K}_{\text{LF}}} \mathcal{Q}_k$ **do**          ▷ e.g., coarse grid
4:      **Inner LP:** $\min_{\{\lambda_{a,j}\}, \{q_\ell\}_{\ell \in \mathcal{K}_{\text{L}}}} \sum_{a \in \mathcal{A}} \langle \widehat{\gamma}_a, \widehat{\rho}_a \rangle$

$$\text{s.t.} \begin{cases} \sum_{j \in S_a} \lambda_{a,j} = 1, \quad \lambda_{a,j} \geq 0 \quad \forall j \in S_a, \\[2mm] \widehat{\rho}_a = \sum_{j \in S_a} \lambda_{a,j} \cdot \widehat{r}_a^{(j)} \end{cases} \quad \forall a \in \mathcal{A}$$

$$-\frac{\delta_\ell}{2} \leq \langle \widehat{u}_{\ell,a}, \widehat{\rho}_a \rangle - q_\ell \leq \frac{\delta_\ell}{2} \qquad \forall a \in \mathcal{A}, \ell \in \mathcal{K}_{\text{L}}$$

$$\begin{cases} \langle \widehat{u}_{k,a}, \widehat{\rho}_a \rangle - \left(q_k + \frac{\delta_k}{2}\right) \langle \widehat{v}_{k,a}, \widehat{\rho}_a \rangle \leq 0, \\[2mm] \left(q_k - \frac{\delta_k}{2}\right) \langle \widehat{v}_{k,a}, \widehat{\rho}_a \rangle - \langle \widehat{u}_{k,a}, \widehat{\rho}_a \rangle \leq 0, \end{cases} \quad \forall a \in \mathcal{A}, k \in \mathcal{K}_{\text{LF}}$$

$$\langle \widehat{v}_{k,a}, \widehat{\rho}_a \rangle \geq \varepsilon_k \qquad \forall a \in \mathcal{A}, k \in \mathcal{K}_{\text{LF}}.$$

5:      Let $\widehat{\Phi}(\vec{q})$ be the optimal value and $\{\lambda_{a,j}\}, \{\widehat{\rho}_a\}$ the optimal variables.
6:      **if** $\widehat{\Phi}(\vec{q}) <$ best **then** best $\leftarrow \widehat{\Phi}(\vec{q}), \vec{q}_{\text{opt}} \leftarrow \vec{q}, \widetilde{\lambda}_{a,j} \leftarrow \lambda_{a,j}, \widetilde{\rho}_a \leftarrow \widehat{\rho}_a$.
7:      **end if**
8: **end for**
9: **Return** Return the target operating points $\{\widetilde{\rho}_a\}_{a \in \mathcal{A}}$.

---

---

**Algorithm 3** `RegionSearch-FeasibilityGuard` ($\alpha$-uniform expansion policy)

---

**Require:** Groups $\mathcal{A}$; hull supports $\{\widehat{\mathcal{H}}_a\}$; plug-in $\{\widehat{\gamma}_a\}$, $\{\widehat{u}_{k,a}\}$, $\{\widehat{v}_{k,a}\}$; tolerances $\vec{\delta} = \{\delta_k\}$; index
   sets $\mathcal{K}_{\mathrm{L}}, \mathcal{K}_{\mathrm{LF}}$; centroid intervals $\{\mathcal{Q}_k\}_{k \in \mathcal{K}_{\mathrm{LF}}}$; margins $\{\varepsilon_k > 0\}$; bisection tolerance $\tau_\alpha > 0$.

**Ensure:** Minimal feasible expansion $\widetilde{\alpha}$ and operating points $\{\widetilde{\rho}_a\}_{a \in \mathcal{A}}$.

    **Step 0: Instantiate search interval.**

1: **for** each $a \in \mathcal{A}$ **do**

2:     $\widetilde{j}_a \in \arg\min_{j \in S_a} \langle \widehat{\gamma}_a, \widehat{r}_a^{(j)} \rangle$; set $\widehat{\rho}_a^{\mathrm{base}} \leftarrow \widehat{r}_a^{(\widetilde{j}_a)}$.

3: **end for**

4: Compute baseline metrics: $g_{\ell,a}^{\mathrm{base}} = \langle \widehat{u}_{\ell,a}, \widehat{\rho}_a^{\mathrm{base}} \rangle$ for $\ell \in \mathcal{K}_{\mathrm{L}}$, $r_{k,a}^{\mathrm{base}} = \dfrac{\langle \widehat{u}_{k,a}, \widehat{\rho}_a^{\mathrm{base}} \rangle}{\langle \widehat{v}_{k,a}, \widehat{\rho}_a^{\mathrm{base}} \rangle}$ for $k \in \mathcal{K}_{\mathrm{LF}}$

    with $\langle \widehat{v}_{k,a}, \widehat{\rho}_a^{\mathrm{base}} \rangle \geq \varepsilon_k$.

5: Baseline disparities: $\Delta_\ell = \max_a g_{\ell,a}^{\mathrm{base}} - \min_a g_{\ell,a}^{\mathrm{base}}$,   $\Delta_k = \max_a r_{k,a}^{\mathrm{base}} - \min_a r_{k,a}^{\mathrm{base}}$.

6: Minimal per-metric expansions: $\alpha_\ell^{\min} = \Delta_\ell / \delta_\ell$,   $\alpha_k^{\min} = \Delta_k / \delta_k$.

7: Set bracket: $\alpha_{\mathrm{lo}} \leftarrow 1$,   $\alpha_{\mathrm{hi}} \leftarrow \max\left\{ 1, \max_{\ell \in \mathcal{K}_{\mathrm{L}}} \alpha_\ell^{\min}, \max_{k \in \mathcal{K}_{\mathrm{LF}}} \alpha_k^{\min} \right\}$.

    **Step 1: Exit early at nominal tolerances.**

8: Run `RegionSearch` with $\vec{\delta}(1) = \vec{\delta}$;

9: **if** feasible **then**

10:     **return** $\widetilde{\alpha} = 1$ and the returned $\{\widetilde{\rho}_a\}_{a \in \mathcal{A}}$.

11: **end if**

    **Step 2: Bisection-search on $\alpha$.**

12: **while** $\alpha_{\mathrm{hi}} - \alpha_{\mathrm{lo}} > \tau_\alpha$ **do**

13:     $\alpha \leftarrow (\alpha_{\mathrm{lo}} + \alpha_{\mathrm{hi}})/2$; set $\vec{\delta}(\alpha) = \{\alpha \delta_k\}$.

14:     Run `RegionSearch` with $\vec{\delta}(\alpha)$.

15:     **if** feasible **then**

16:         $\alpha_{\mathrm{hi}} \leftarrow \alpha$; cache current solution $\{\widehat{\rho}_a\}_{a \in \mathcal{A}}$.

17:     **else**

18:         $\alpha_{\mathrm{lo}} \leftarrow \alpha$.

19:     **end if**

20: **end while**

21: **return** $\widetilde{\alpha} \leftarrow \alpha_{\mathrm{hi}}$ and the cached solution $\{\widetilde{\rho}_a\}_{a \in \mathcal{A}}$ at $\alpha_{\mathrm{hi}}$

---

randomized classifier applied to a mixed-GWTR: `LabelFlipping` flips the mixed-GWTR labels with outcome-dependent probabilities. We begin by defining interventions.

**Definition A.3** (Interventions). Fix a baseline post-processor $f^{(0)} : \mathcal{X} \times \mathcal{A} \to \{0,1\}$ as per equation 5. For any candidate post-processor $f$, the population *intervention rate* is $\mathrm{Int}(f; f^{(0)}) := \Pr\left(f(X, A) \neq f^{(0)}(X, A)\right)$, with group-wise intervention rates $\mathrm{Int}_a(f; f^{(0)}) := \Pr\left(f(X, A) \neq f^{(0)}(X, A) \mid A=a\right)$. On a dataset $\mathcal{D}$, the *expected empirical intervention rate*[13] is

$$\widetilde{\mathrm{Int}}(f; f^{(0)}; \mathcal{D}) := \frac{1}{|\mathcal{D}|} \sum_{(x,a,y) \in \mathcal{D}} \Pr\left(f(x, a) \neq f^{(0)}(x, a)\right),$$

where the probability is taken over any randomization introduced by $f$ and/or $f^{(0)}$. The expected number of interventions is $|\mathcal{D}| \cdot \widetilde{\mathrm{Int}}$.

Similarly, on a dataset $\mathcal{D}$, the *empirical intervention rate* removes the expectation over the randomization procedures induced by the post-processors, and is given by

$$\widehat{\mathrm{Int}}(f; f^{(0)}; \mathcal{D}) := \frac{1}{|\mathcal{D}|} \sum_{(x,a,y) \in \mathcal{D}} \mathbb{1}\left(f(x, a) \neq f^{(0)}(x, a)\right),$$

so that the empirical number of interventions is $|\mathcal{D}| \cdot \widehat{\mathrm{Int}}$.

The intervention rate is simply the misclassification error when the true labels are provided by the baseline post-processor $f^{(0)}$. However, we find it helpful to define this quantity separately.[14]

**Definition A.4** (Minimal intervention). Consider a class $\mathcal{F}_N$ of post-processors as per equation 5. We define the following minimal intervention post-processing objective:

$$\min_{f \in \mathcal{F}_N} \widetilde{\mathrm{Int}}(f; f^{(0)}; \mathcal{D}_{\mathrm{post}}) \quad \text{s.t.} \quad \widehat{\rho}_a(f) = \widetilde{\rho}_a, \forall a \in \mathcal{A}, \tag{18}$$

where $\widehat{\rho}_a(f)$ are the empirical operating characteristic rates of post-processors $f \in \mathcal{F}_N$ on the post-processing set $\mathcal{D}_{\mathrm{post}}$ (see §A.2.1), and $\widetilde{\rho}_a = (\widetilde{\mathrm{TPR}}_a, \widetilde{\mathrm{FPR}}_a, 1)^\top$ are target operating characteristics (e.g., the output of Algorithm 3).

### A.3.2 FORMULATING THE MININTERVENTION PROBLEM

Given a fixed post-processing set, the expected number of interventions (equation A.3) can be computed in closed form for any mixed-GWTR. In particular, for a base post-processor $f^{(0)}$, let $s_a^+ := \Pr\left(f^{(0)}=1 \mid A=a\right)$ denote the baseline selection rate. Then, the expected empirical intervention rates for a post-processor $\widetilde{f}$ on $f^{(0)}$ under our randomization procedures are given as follows.

Inspired by the label flipping procedure[15] of Hardt et al. (2016) let $(p_{a,0}, p_{a,1})$ be chosen to match the target rates via equation 13. The expected empirical intervention rate in group $a$ is then

$$\mathcal{I}_a^{\mathrm{LF}} := \mathrm{Int}_a^{\mathrm{LF}}(\widetilde{f}; f^{(0)}) = \mathbb{E}\left[\mathbb{1}\left(\widetilde{f} \neq f^{(0)}\right)\Big| A=a\right] = s_a^+(1 - p_{a,1}) + (1 - s_a^+)p_{a,0}. \tag{19}$$

On a finite split $\mathcal{D}_{\mathrm{post}}^{(a)}$, this can be estimated by

$$\widetilde{\mathcal{I}}_a^{\mathrm{LF}} := \widetilde{\mathrm{Int}}_a^{\mathrm{LF}}(\widetilde{f}; f^{(0)}; \mathcal{D}_{\mathrm{post}}) = \left(n_{1,a}^{(0)}(1 - p_{a,1}) + (n_a - n_{1,a}^{(0)})p_{a,0}\right)/n_a.$$

Taking the base post-processor $f^{(0)}$ to be of the form given by equation 11, we can substitute the closed-form expressions for the parameters of the randomization procedure,

---

[13] To clarify, we are computing an expectation over the randomization procedure induced by post-processors $f$ and/or $f^{(0)}$ for a *fixed* dataset $\mathcal{D}$.

[14] We do not claim that interventions are the best desiderata to consider when developing classifiers for real-world deployment. We provide a general procedure for constructing classifiers to achieve any target operating characteristic realizable by post-processing (see Algorithm 4), and leave the question of designing additional ethically and legally sound objectives to practitioners (see, e.g., Heidari et al. (2019); Regoli et al. (2023)).

[15] Developed for the case of perfect fairness for equality of opportunity and equalized odds.

$$p_{a,1}(\theta) = \frac{\widetilde{\mathrm{FPR}}_a \mathrm{FNR}_{a,\theta}^{(0)} - \left(1 - \widetilde{\mathrm{FNR}}_a\right)\left(1 - \mathrm{FPR}_{a,\theta}^{(0)}\right)}{\det(\theta)},$$

$$p_{a,0}(\theta) = \frac{\left(1 - \widetilde{\mathrm{FNR}}_a\right)\mathrm{FPR}_{a,\theta}^{(0)} - \widetilde{\mathrm{FPR}}_a\left(1 - \mathrm{FNR}_{a,\theta}^{(0)}\right)}{\det(\theta)}, \tag{20}$$

into the closed-form intervention expressions (equation 19). This yields one-dimensional objectives in $\theta$:

$$\mathcal{I}_{a,\theta}^{\mathrm{LF}} = s_a^+\left(1 - p_{a,1}(\theta)\right) + \left(1 - s_a^+\right)p_{a,0}(\theta). \tag{21}$$

This suggests enumerating adjacent hull pairs $\mathsf{Edges}_a = \{(h, h+1)\}$ for each group $a \in \mathcal{A}$, and minimizing the expected empirical intervention rate $\widetilde{\mathcal{I}}_{a,\theta}^{\mathrm{LF}}$ over $\theta \in [0,1]$ for each edge, either with a coarse grid or golden-section search (see e.g., Section 7.2 of Chong et al. (2023)). We then select the edge and $\theta$ with the smallest empirical intervention, thereby yielding the minimal-intervention instantiation of the chosen randomization mechanism while matching the target operating characteristics.

Since not all operating characteristics are attainable by applying a randomization rule to a base post-processor of the form in equation 11, we check for feasibility by ensuring that the solved randomization parameters lie in the appropriate set (i.e, $(p_{a,0}, p_{a,1}) \in [0,1]^2$ for `LabelFlipping`). We skip values of $\theta$ if they are infeasible.

**Snapping to the convex hull boundary.** In practice, for stability and computational efficiency, we check whether the target operating characteristic for a group lies near or on its group-wise ROC convex hull before conducting the search over adjacent hull pairs.

If the target operating characteristic lies exactly on the boundary of the convex hull, we set the intervention rate to zero and skip the edge search procedure for this group. We check closeness of the target point within adaptive tolerance values—$\mathrm{tol}_{\mathrm{fpr}} = \xi/n_{a,0}$ and $\mathrm{tol}_{\mathrm{fnr}} = \xi/n_{a,1}$ for tunable $\xi$ where $n_{a,0}$ and $n_{a,1}$ are the number of negative and positive labels in group $a$, respectively, cf. §A.2.1. If the target point is within these tolerance levels to the convex hull, we snap to the nearest edge and, again, skip the edge search procedure.

Altogether, this procedure is detailed in Algorithm 5 and provides the final module of the post-processing procedure outlined in our ROCF-pipeline (Algorithm 1).

---

**Algorithm 4** Construct Classifier (MinIntervention)

---

**Require:** probabilistic predictor $s$; groups $\mathcal{A}$; postprocessing set $\mathcal{D}_{\mathrm{post}}$; target rates $\{\widetilde{\rho}_a\}_{a \in \mathcal{A}}$ from Algorithm 3; a construction subroutine $\mathsf{Construct} \in \{\texttt{LabelFlipping}, \text{any admissible}\}$.
1: **for** each $a \in \mathcal{A}$ **do**
2:    Extract group subset $\mathcal{D}_{\mathrm{post}}^{(a)} = \{(x_i, a, y_i) \in \mathcal{D}_{\mathrm{post}} : A_i = a\}$.
3:    **Call subroutine:** $\{\vec{\zeta}_a\}_{a \in \mathcal{A}} \leftarrow \mathsf{Construct}\left(s(\cdot, a), \mathcal{D}_{\mathrm{post}}^{(a)}, \widetilde{\rho}_a\right)$.
4:    Define $f_a(\cdot)$ and $f_a^{(0)}(\cdot)$ from parameters $\vec{\zeta}_a$ (e.g., mixing thresholds; label-flip rates).
5: **end for**
6: Assemble $f(x, a) := f_a(x)$, $f^{(0)}(x, a) := f_a^{(0)}(x)$ and compute $\widehat{\rho}_a(f)$ on $\mathcal{D}_{\mathrm{post}}$.
7: Report $\widehat{\mathrm{Int}}(f; f^{(0)}; \mathcal{D}_{\mathrm{post}})$ and $\{\widehat{\mathrm{Int}}_a(f; f^{(0)}; \mathcal{D}_{\mathrm{post}})\}$.
8: **Return** $f$, realized rates $\{\widehat{\rho}_a\}$, and intervention statistics.

---

**On the global optimality for minimum interventions.** The construction of LABELFLIPPING chooses intervention parameters that are optimal conditional on the empirical ROC curve: for a given target operating point, the intervention objective in equation 21 computes the minimal mixing or flipping probabilities needed to achieve that point. In this sense, this mechanism minimizes the population-level expected number of interventions for their particular functional form.

---

**Algorithm 5** `MinIntervention` subroutine (`LabelFlipping` modality)

---

**Require:** probabilistic predictor $s$; post-processing data $\mathcal{D}_{\text{post}}$; target operating characteristics $\{(\widetilde{\text{TPR}}_a, \widetilde{\text{FPR}}_a)\}_{a \in \mathcal{A}}$; rate tolerance $\xi$; 1D line-search routine $G$; mode $= \{\text{LF}\}$.

**Ensure:** parameter recipe and mechanism-specific parameters $\{\vec{\zeta}_a\}_{a \in \mathcal{A}}$

1: **for** group $a \in \mathcal{A}$ **do**

2:     **Compute empirical hull.** Extract the subpopulation $\{(x_i, a, y_i) \in \mathcal{D}_{\text{post}}^{(a)}\}$, and use $s(x_i, a)$ to form the empirical ROC and upper convex hull (cf.§A.2.1) and order it:

$$\widehat{\mathcal{H}}_a^{\text{ord}} = \left\{ (t_{1,a}, \ldots, t_{S_a,a}) \text{ along with } (\widehat{\text{FNR}}_a^{(h)}, \widehat{\text{FPR}}_a^{(h)}, \widehat{s}_{a,+}^{(h)}) \right\}_{h=1}^{S_a};$$

    here, $\widehat{s}_{a,+}$ is the plug-in estimate for $s_{a,+}^{(h)} := \Pr(f^{(0)}{=}1 \mid A{=}a)$ where $f^{(0)}(x, a) = \mathbb{1}(s(x, a) \geq t_{h,a})$.

3:     best $\leftarrow +\infty$.

4:     **Snap to convex hull boundary.** If $(\widetilde{\text{TPR}}_a, \widetilde{\text{FPR}}_a)$ lies on boundary of $\widehat{\mathcal{H}}_a^{\text{ord}}$ with snap tolerance $\xi$ (cf. §A.3.2), compute the interpolant $\theta^{\text{edge}} \in [0, 1]$ and set:

      • mode=LF: use $p_{a,0}{=}0, p_{a,1}{=}1$ (no flips); objective $= 0$.

    Update $\vec{\zeta}_a \leftarrow (t_{h,a}, t_{h+1,a}, \theta^{\text{edge}}, \text{mechanism params})$, best $\leftarrow 0$, and **continue** to the next group.

5:     **for** each adjacent pair $(h, h{+}1)$ in $\widehat{\mathcal{H}}_a^{\text{ord}}$ **do,**

6:       **Search over $\theta$ on this edge** via a 1D line search $G$ (e.g., grid search, golden-section search):

      1. Read off

$$\widehat{\text{FPR}}_{a,\theta}^{(0)} = (1 - \theta)\widehat{\text{FPR}}_a^{(h)} + \theta\widehat{\text{FPR}}_a^{(h+1)}, \quad \widehat{\text{FNR}}_{a,\theta}^{(0)} = (1 - \theta)\widehat{\text{FNR}}_a^{(h)} + \theta\widehat{\text{FNR}}_a^{(h+1)},$$

        and the baseline selection rate $\widehat{s}_{a,+}(\theta) = (1 - \theta)\widehat{s}_{a,+}^{(h)} + \theta\widehat{s}_{a,+}^{(h+1)}$.

      2. **Compute mechanism parameters.**

        • If mode=LF: compute $(p_{a,0}(\theta), p_{a,1}(\theta))$ via equation 20; feasibility requires $(p_{a,0}(\theta), p_{a,1}(\theta)) \in [0, 1]^2$.

        Skip if infeasible.

      3. **Compute expected number of interventions.**

        • mode=LF: evaluate $\widetilde{\mathcal{I}}_{a,\theta}^{\text{LF}}$ (cf. equation 21).

      4. If $\widetilde{\mathcal{I}}_{a,\theta}^{\text{LF}} < $ best, set $\vec{\zeta}_a \leftarrow (t_{h,a}, t_{h+1,a}, \theta, \text{mechanism params at } \theta)$ and best $\leftarrow \widetilde{\mathcal{I}}_{a,\theta}^{\text{LF}}$.

7:     **end for**

8:     **Assemble outputs.** Return the parameter recipe

$$\vec{\zeta}_a = \left( t_{h,a}, t_{h+1,a}, \theta, \underbrace{(p_{a,0}, p_{a,1})}_{\text{LF}} \right),$$

    which specifies a base post-processor $f_a^{(0)}(\cdot)$ by equation 11 and the final $f_a(\cdot)$ via equation 12 (LF).

9:     **return** $\{\vec{\zeta}_a\}_{a \in \mathcal{A}}$

10: **end for**

---

More generally, this mechanism is not necessarily "globally optimal" among all post-processing rules for minimizing the intervention rate (cf. Definition A.3). Establishing such optimality is challenging because the minimal achievable intervention rate depends on the geometry of the empirical ROC curve and the baseline post-processor.

In light of this, one can view these intervention methods as interpretable, analytically tractable, and, most importantly, practically effective instantiations of the broader design space of all post-processing rules (see, e.g., §4.3 and §5).

### A.3.3 MULTIPLE PROTECTED ATTRIBUTES

In `RegionSearch` (see Algorithm 2), the inner LP optimizes group-wise operating characteristics over the empirical ROC-hull supports, yielding $\mathcal{O}(|\mathcal{A}|)$ linear constraints and $\mathcal{O}(\sum_{a \in \mathcal{A}} |S_a|)$ variables, where $|S_a|$ is the number of support points in the empirical convex hull of group $a$. As discussed in Remark A.2 and as observed in our experiments (see runtimes in §A.6.2), the set of support points is modestly sized. Given the optimal operating characteristics, our classifier construction decouples across groups, so the randomization hyperparameters of Algorithm 4 can be found in parallel.

### A.3.4 MULTIPLE CLASSES

In the multi-class setting, the natural analog of operating characteristics (cf. equation 1) is a confusion-rate matrix, which records a classifier's joint misclassification probabilities for each pair of true and predicted classes (see equation 22).

In this section, we extend our method to this setting by introducing a direct intervention mechanism on the confusion-rate matrix, preserving the intuition and theoretical guarantees of the binary class case.

**Setup in the multi-class setting.** Let $X \in \mathcal{X}$ be features, $A \in \mathcal{A}$ a discrete protected attribute, and $Y \in [C] := \{1, \ldots, C\}$ the multiclass label. We are given a pre-trained probabilistic predictor $s : \mathcal{X} \times \mathcal{A} \to \Delta^{C-1}$, $s(x, a) = (s_1(x, a), \ldots, s_C(x, a))$ and a post-processing set $\mathcal{D}_{\text{post}} = \{(x_i, a_i, y_i)\}_{i=1}^N$ which is drawn i.i.d. from the population.

A (possibly randomized) post-processor $f : \mathcal{X} \times \mathcal{A} \to [C]$ induces, for each group $a$, the *confusion–rate matrix*[16]

$$M_a(y, \hat{c}) := \Pr\left(\hat{Y} = \hat{c}, Y = y \mid A = a\right), \qquad y, \hat{c} \in [C]. \tag{22}$$

Row sums equal the class priors $\pi_{a,y} := \Pr(Y = y \mid A = a)$, so $\sum_{\hat{c}} M_a(y, \hat{c}) = \pi_{a,y}$. Let $u_{a,\hat{c}} := \sum_y M_a(y, \hat{c}) = \Pr(\hat{Y} = \hat{c} \mid A = a)$ denote the predicted-class marginals. The misclassification loss is linear in $M_a$: $\Pr(\hat{Y} \neq Y \mid A = a) = \sum_{y \neq \hat{c}} M_a(y, \hat{c})$.

**Post-processing confusion rate region.** As in the binary classification setting, we restrict to post-processors that only depend on $s$ and $\mathcal{D}_{\text{post}}$:

$$\mathcal{F}_N = \mathcal{F}_N(s; \mathcal{D}_{\text{post}}) := \{f(\cdot, a) \text{ obtained from } s(\cdot, a) \text{ and } \mathcal{D}_{\text{post}} \text{ (possibly randomized)}\}. \tag{23}$$

Define the *realizable (or post-processing) confusion-rate region*

$$\mathcal{R}_a(s) := \{M_a(f) : f \in \mathcal{F}_N\}. \tag{24}$$

Randomizing the post-processors by taking a mixture of the confusion rate matrices makes $\mathcal{R}_a(s)$ convex.

**Multi-class group fairness for common linear-fractional and linear constraints.**

We provide details for extending our ROC feasibility region approach to the multi-class classification setting for common linear-fractional and linear group fairness constraints, and defer extending our method in full generality for future work.

Similar to the works of Baumann et al. (2022); Alghamdi et al. (2022); Xian & Zhao (2024), we present here the definitions of common group fairness metrics for the multi-class setting. Similar

---

[16]In the binary classification setting, since the rate-matrix is row-stochastic, this reduces to the familiar group-wise operating characteristics.

to the binary setting, we work in the approximate group-fairness setting, where we seek to equalize group performance metrics across protected attributes and/or predicted/true class labels.

*Multi-class demographic parity (DP).* For each predicted class $\hat{c}$, multi-class DP requires $u_{a,\hat{c}} = \Pr(\hat{Y} = \hat{c} \mid A = a) = \sum_y M_a(y, \hat{c})$ to be approximately equal across groups. Introducing centroids $q_{\hat{c}}^{\mathrm{DP}} \in [0, 1]$ with $\sum_{\hat{c}} q_{\hat{c}}^{\mathrm{DP}} = 1$ and imposing the linear bands, this constraint is written as

$$-\frac{\delta_{\mathrm{DP}}}{2} \leq \sum_y M_a(y, \hat{c}) - q_{\hat{c}}^{\mathrm{DP}} \leq \frac{\delta_{\mathrm{DP}}}{2} \qquad \text{for all } a, \hat{c}. \qquad (25)$$

*Multi-class equalized odds (EO).* For each true class $y$, multi-class EO equalizes the conditional distribution of $\hat{Y}$ given $Y = y$ across groups:

$$\Pr(\hat{Y} = \hat{c} \mid Y = y, A = a) = \frac{M_a(y, \hat{c})}{\pi_{a,y}} \approx q_{\hat{c}|y}^{\mathrm{EO}} \qquad \text{for all } y, \hat{c} \in [C],$$

Since $\pi_{a,y}$ is a fixed row-sum, clearing the denominator yields linear bands:

$$-\frac{\delta_{\mathrm{EO}}}{2}\pi_{a,y} \leq M_a(y, \hat{c}) - q_{\hat{c}|y}^{\mathrm{EO}}\pi_{a,y} \leq \frac{\delta_{\mathrm{EO}}}{2}\pi_{a,y}, \quad \sum_{\hat{c}} q_{\hat{c}|y}^{\mathrm{EO}} = 1, q_{\hat{c}|y}^{\mathrm{EO}} \in [0, 1]. \qquad (26)$$

*Multi-class sufficiency (predictive parity & false-omission rate parity).* For each predicted class $\hat{c}$, multi-class sufficiency equalizes the conditional distribution of $Y$ given $\hat{Y} = \hat{c}$ across groups:

$$\Pr(Y = y \mid \hat{Y} = \hat{c}, A = a) = \frac{M_a(y, \hat{c})}{u_{a,\hat{c}}} \approx q_{y|\hat{c}}^{\mathrm{Suff}} \qquad \text{for all } y, \hat{c} \in [C],$$

with $\vec{q}_{\cdot|\hat{c}}^{\mathrm{Suff}} \in \Delta^{C-1}$. For a *fixed* $\vec{q}_{\cdot|\hat{c}}^{\mathrm{Suff}}$, clearing the denominator yields linear bands:

$$-\frac{\delta_{\mathrm{Suff}}}{2}u_{a,\hat{c}} \leq M_a(y, \hat{c}) - q_{y|\hat{c}}^{\mathrm{Suff}}u_{a,\hat{c}} \leq \frac{\delta_{\mathrm{Suff}}}{2}u_{a,\hat{c}}, \quad \sum_y q_{y|\hat{c}}^{\mathrm{Suff}} = 1, q_{y|\hat{c}}^{\mathrm{Suff}} \in [0, 1]. \qquad (27)$$

**Population level optimization in confusion-rate space.**

Let $\Gamma_a \in \mathbb{R}^{C \times C}$ encode an arbitrary loss criterion, so that $\langle \Gamma_a, M_a \rangle$ is the group-wise risk (e.g., for 0-1 misclassification loss, $\Gamma_a$ has 0 on the diagonal and 1 off the diagonal). The population problem is then

---

**Population-level Optimal Fair Classification via Post-Processor Confusion Rates**

$$\min_{\{M_a\}_{a \in \mathcal{A}}} \quad \sum_{a \in \mathcal{A}} \langle \Gamma_a, M_a \rangle$$

$$\text{s.t.} \quad M_a \in \mathcal{R}_a(s), \quad M_a \geq 0, \quad M_a\vec{1} = \pi_a, \qquad (28)$$

$$\text{with DP/EO/Suff. constraints equation 25–equation 27.}$$

---

For a fixed $\{\vec{q}_{y|\hat{c}}^{\mathrm{Suff}}\}_{\hat{c} \in [C]}$, we obtain an inner linear program in the variables $\{M_a\}_{a \in \mathcal{A}}$ and centroids for the linear-fractional and linear group fairness constraints (cf. equation 25–equation 27). By the following theorem, searching $\{\vec{q}_{y|\hat{c}}^{\mathrm{Suff}}\}_{\hat{c} \in [C]}$ over $\prod_{\hat{c}} \Delta^{C-1}$ recovers the optimum of equation 28:

**Theorem A.5** (Centroid reduction in the multiclass setting). *Define the value function $\Phi$ such that $\Phi(\vec{q})$ is the optimal objective value of the inner linear program in equation 28 obtained by fixing the centroid parameters $\vec{q} := \{q_{y|\hat{c}}^{\mathrm{Suff}}\}_{\hat{c} \in [C]}$. Then, the optimal value of equation 28 is equal to*

$$\min_{\vec{q} \in \mathcal{Q}} \Phi(\vec{q}),$$

*where $\mathcal{Q} := \underbrace{\Delta^{C-1} \times \cdots \times \Delta^{C-1}}_{C \text{ times}}$. Moreover, an optimizer $\{M_a^\star\}_{a \in \mathcal{A}}$ of the original objective in equation 28 can be obtained by selecting any minimizer $\vec{q}^\star \in \arg\min_{\vec{q} \in \mathcal{Q}} \Phi(\vec{q})$ and then solving the corresponding linear program at fixed $\vec{q}^\star$.*

The proof mirrors the binary class setting case, where pairwise disparities are equivalent to the existence of a common centroid; linear–fractional bands become linear once the (sufficiency-based) centroid is fixed; and taking a union over centroids recovers the original feasible set.

*Sample complexity.* Our finite-sample guarantees for the binary-class setting can be directly translated to the multi-class case. Recall, in the binary setting, Theorem 4.2 controls the excess risk and fairness slack in terms of $n_{\min} = \min_{a,j} n_{a,j}$ over group-label counts $j \in \{0,1\}$. In the multi-class setting, these group-label pairs simply become group-class pairs $(a,y)$ with $y \in [C]$, so that $n_{\min} := \min_{a,y} n_{a,y}$ appears in the bounds. The same Hoeffding- and union-bound argument yields a $\tilde{O}(1/\sqrt{n_{\min}})$ rate, which only incurs a mild polynomial dependence on $C$ due to the $C(C-1)$ confusion-rate entries. Thus, the guarantees remain conditional on the fixed score function $s$, require neither calibration nor Bayes-optimality, and preserve the same $1/\sqrt{n}$ convergence rate as in the binary case.

**Empirical level optimization in confusion-rate space.**

Similar to the binary class setting, we use a post-processing set to generate a collection of confusion-rate matrices that approximates the frontier of the realizable confusion-rate region (cf. equation 24).

We then prune to support points and allow mixtures between these points, thereby ensuring that the final optimization has considerably fewer variables without losing any expressivity in the resulting family of randomized confusion-rate matrices.

*Building the empirical convex hull of confusion-rate matrices.* Define the group-wise empirical confusion-rate matrix of a post-processor $f$ as

$$\widehat{M}_a(y, \hat{c}) := \frac{1}{n_a} \sum_{i:\, a_i = a} 1\left(Y_i = y, f(X_i, a) = \hat{c}\right),$$

where $n_a = \#\{i : a_i = a\}$ is the number of post-processing samples in group $a$.[17]

We seek to construct a finite support of empirical confusion-rate matrices $\{\widehat{M}_a^{(j)}\}_{j \in S_a}$ with index sets $\{S_a\}_{a \in \mathcal{A}}$ whose convex hull

$$\widehat{\mathcal{R}}_a(s) = \text{conv}\{\widehat{M}_a^{(j)} : j \in S_a\}$$

approximates the realizable confusion-rate region $\mathcal{R}_a(s)$.

This is achieved by instantiating several simple, deterministic rule families $G \in \mathcal{G}$ that map the learned score vectors $s(x, a) \in \Delta^{C-1}$ to a label in $[C]$ (e.g., one versus rest threshold classifiers (Chow, 2003; Geifman & El-Yaniv, 2017)) or margin-based classifiers (Bartlett & Wegkamp, 2008)).[18]

For each family $G$, we then collect empirical confusion-rate matrices $S_a^G := \{\widehat{M}_a(g) : g \in \mathcal{G}\}$ and keep the *extreme points* $\text{ext}(S_a^G)$ of their convex hull. Finally, we form the union over all families and extract the global extreme points

$$S_a = \text{ext}\left(\bigcup_{G \in \mathcal{G}} \text{ext}(S_a^G)\right).$$

The set $S_a$ defines the empirical support of attainable confusion-rate matrices for group $a$, and randomization over $S_a$ recovers the entire empirical region $\widehat{\mathcal{R}}_a(s)$.

*Empirical level inner LP.* For fixed linear-fractional centroids $\{q_{y|\hat{c}}^{\text{Suff}}\}_{\hat{c} \in [C]}$, the inner optimization reduces to an LP in convex weights and linear centroids.

Analogous the binary class setting, we use empirical plug-ins for proportion based quantities (e.g., $\widehat{\pi}_{a,y} = 1/n_a \sum_{i:\, a_i = a} 1\,(Y_i = y)$) and empirical group-wise confusion-rate matrices $\{\widehat{M}_a^{(j)}\}_{j \in S_a}$ obtained from the post-processing dataset.

---

[17] If scores are tied, then ties can be broken with a deterministic rule (e.g., lexicographic ordering), resulting in a well-defined confusion matrix.

[18] In the binary-class setting, it suffices to consider group-wise thresholding rules (cf. equation 3) since the realizable operating-characteristic region (cf. equation 6) is completely determined by these deterministic rules. In the multiclass setting, however, there is no analogously simple one-parameter family that recovers all extreme confusion-rate points; multiple families are needed to adequately explore the frontier of $\mathcal{R}_a(s)$ (cf. equation 24).

We also impose additional positivity guards with negligibly small $\varepsilon_{\hat{c}} > 0$ to ensure the centroid-based linearization is valid for the sufficiency based constraints.

---

**(Empirical) Inner Optimization for Fixed Linear Fractional Centroids in the Multi-class Setting**

$$\min_{\substack{\{\lambda_{a,j}\},\{q_{\hat{c}}^{\mathrm{DP}}\}, \\ \{q_{\hat{c}|y}^{\mathrm{EO}}\}}} \quad \sum_{a \in \mathcal{A}} \left\langle \Gamma_a, \sum_{j \in S_a} \lambda_{a,j} \widehat{M}_a^{(j)} \right\rangle$$

$$\text{s.t.} \quad \sum_{j \in S_a} \lambda_{a,j} = 1, \quad \lambda_{a,j} \geq 0, \qquad \forall a, j,$$

$$\text{DP bands: } -\tfrac{\delta_{\mathrm{DP}}}{2} \leq \sum_y \left( \sum_j \lambda_{a,j} \widehat{M}_a^{(j)}(y, \hat{c}) \right) - q_{\hat{c}}^{\mathrm{DP}} \leq \tfrac{\delta_{\mathrm{DP}}}{2}, \forall a, \hat{c},$$

$$\text{EO bands: } -\tfrac{\delta_{\mathrm{EO}}}{2} \widehat{\pi}_{a,y} \leq \sum_j \lambda_{a,j} \widehat{M}_a^{(j)}(y, \hat{c}) - q_{y|\hat{c}}^{\mathrm{EO}} \widehat{\pi}_{a,y} \leq \tfrac{\delta_{\mathrm{EO}}}{2} \widehat{\pi}_{a,y}, \forall a, y, \hat{c},$$

$$\text{Suff. bands: } -\tfrac{\delta_{\mathrm{Suff}}}{2} \sum_{y'} \sum_j \lambda_{a,j} \widehat{M}_a^{(j)}(y', \hat{c}) \leq$$

$$\sum_j \lambda_{a,j} \widehat{M}_a^{(j)}(y, \hat{c}) - q_{y|\hat{c}}^{\mathrm{Suff}} \left( \sum_{y'} \sum_j \lambda_{a,j} \widehat{M}_a^{(j)}(y', \hat{c}) \right) \leq$$

$$\tfrac{\delta_{\mathrm{Suff}}}{2} \sum_{y'} \sum_j \lambda_{a,j} \widehat{M}_a^{(j)}(y', \hat{c}), \ \forall a, y, \hat{c},$$

$$\text{Positivity guards: } \sum_y \sum_j \lambda_{a,j} \widehat{M}_a^{(j)}(y, \hat{c}) \geq \varepsilon_{\hat{c}}, \forall a, \hat{c}.$$

$$\tag{29}$$

---

An outer search over $\bar{q}^{\mathrm{Suff}}$ then produces target confusion rates $\{\widetilde{M}_a\}_{a \in \mathcal{A}}$. Similar to the binary class setting, a simple feasibility guard can minimally relax the tolerances when the user-specified fairness levels are empirically infeasible.

*Computational complexity.* In the multi-class setting, each group-wise confusion–rate matrix lies in an affine space of intrinsic dimension $d = C(C-1)$, so the inner problem remains a polynomial-size LP in the convex weights and fairness centroids, and the outer search over $\bar{q}^{\mathrm{Suff}} \in \prod_{\hat{c}} \Delta^{C-1}$ has the same structure as in the binary case.

The principal open question is whether the empirical hull, constructed from pruning many deterministic classifiers over $\mathcal{D}_{\mathrm{post}}$, remains small in practice, as it does in the binary setting (e.g., $\approx$ 20-30 support points per group even for datasets with over one million samples like ACSINCOME; see also §A.5).

A direct analogue of the theory available in the binary class setting (e.g., Grenander-style arguments for the ROC frontier as in Groeneboom (2021)) is not yet available for multi-class confusion-rate matrices. We view developing a full theoretical and/or empirical characterization of the number of support points of this empirical hull as an interesting direction for future work, but note that it does not affect the polynomial-time solvability of each inner-optimization subproblem for any fixed number of classes.

**Constructing classifiers.**

Given a baseline post-processor with confusion matrix $M_a^{(0)}$ for group $a$ and a target $\widetilde{M}_a$, we give one particular multi-class construction.

*Multi-class label-flipping.* Let $R_a \in \mathbb{R}^{C \times C}$ be a column-stochastic matrix. If the baseline post-processor predicts $\hat{c} \in [C]$, the final prediction is drawn randomly from the $\hat{c}$-th column of $R_a$. The resulting confusion-rate matrix then requires

$$\widetilde{M}_a = M_a^{(0)} R_a, \qquad R_a^\top \vec{1} = \vec{1}, R_a \geq 0. \tag{30}$$

The expected intervention rate in group $a$ equals $\text{Int}_a^{\text{LF}} = 1 - \sum_{\hat{c}} u_{a,\hat{c}} R_a(\hat{c}, \hat{c})$.

*Choosing the baseline post-processor and finding the min-intervention classifier.* Unlike the binary class setting, where any feasible point on the convex hull of the ROC curve lies on a one-dimensional segment between two adjacent threshold rules (i.e., a mixed-GWTR cf. equation 4), the multi-class feasible region is a higher-dimensional polytope whose extreme points correspond to many deterministic post-processors with empirical confusion-rate matrices $\{\widehat{M}_a^{(j)}\}_{j \in S_a}$. Consequently, it is non-trivial to identify which mixture of deterministic rules, indexed by $j \in S_a$, we should take as the baseline post-processor.

Instead, one can adopt a simple and fully empirical strategy. For each group $a$, we iterate over all baseline candidates $\{\widehat{M}_a^{(j)}\}_{j \in S_a}$, and, for each baseline $j$, we solve the small convex feasibility problem associated with the chosen intervention mechanism. That is, for the label-flipping mechanism, we find parameters $R_a$ such that equation 30 hold, respectively, while minimizing the expected intervention rate. If no such parameters exist for baseline $j$, we deem that baseline post-processor infeasible under the mechanism and skip it.

Among all baselines $j$ for which a feasible mechanism parameterization exists, we simply return the one attaining the smallest intervention rate. This procedure guarantees that the final randomized classifier is realizable and chosen to minimize the expected number of interventions.

### A.3.5   ROC-F PROCEDURE FOR A SIMPLIFIED SETTING: DP AND PP

In this section, we provide a single self-contained algorithm with the full end-to-end implementation of our procedure for a single linear-fractional and linear constraint of predictive parity (PP) and demographic parity (DP), respectively – in particular, Algorithm 6 below is a specific instantiation of the end-to-end pipeline outlined in Algorithm 1.

Note that the construction of classifiers for target operating characteristics rates does not require these specific group fairness constraints (i.e., the "Construct classifier" step in Algorithms 1 and 6; see, also, §4.3, §A.3.1); nonetheless, we present our classifier construction procedure as a self-contained algorithm inside Algorithm 6 for completeness and increased digestibility of our methodology.

### A.4   THEORETICAL DETAILS AND PROOFS

**Admissible centroid sets.**   For each linear–fractional (LF) constraint $k \in \mathcal{K}_{\text{LF}}$, let $[L_k, U_k] \subseteq [0,1]$   for all $a \in \mathcal{A}$, denote the range of the LF group performance function (cf. Definition 3.1). Given a disparity level $\delta_k \in [0, U_k - L_k]$, we define the *admissible centroid interval* as

$$Q_k := \left[ L_k + \tfrac{\delta_k}{2}, U_k - \tfrac{\delta_k}{2} \right]. \tag{31}$$

Denote $z_a := G_{k,a}(\vec{\rho}_a)$, $a \in \mathcal{A}$, which satisfy $\max_{a,a'} |z_a - z_{a'}| \le \delta_k$. Then denote

$$\bigcap_{a \in \mathcal{A}} \left[ z_a - \tfrac{\delta_k}{2}, z_a + \tfrac{\delta_k}{2} \right] = \left[ \max_a z_a - \tfrac{\delta_k}{2}, \min_a z_a + \tfrac{\delta_k}{2} \right] =: \mathcal{I}_k,$$

so that $\mathcal{I}_k \cap \left[ L_k + \tfrac{\delta_k}{2}, U_k - \tfrac{\delta_k}{2} \right] \ne \varnothing$, and any centroid $q_k$ chosen from the intersection automatically lies in $Q_k$.

For common LF constraints like predictive parity and false omission rate parity, we clearly have $[L_k, U_k] = [0,1]$ and $\delta_k \in [0,1]$. Hence, the admissible set of centroids for these metrics is $Q_k = [\delta_k/2, 1 - \delta_k/2]$.

**Definition A.6** (Operating characteristic feasibility region)**.** Let $\mathcal{K}_{\text{L}}$ and $\mathcal{K}_{\text{LF}}$ index linear and linear–fractional group performance functions, respectively, and let $\vec{q} = (q_k)_{k \in \mathcal{K}_{\text{LF}}}$ be fixed centroids for the LF constraints. Define $\mathfrak{F}(\vec{q}; s, \vec{\delta})$ to be the set of $\left\{ \{\vec{\rho}_a\}_{a \in \mathcal{A}} \right.$ such that there are $q_\ell$, $\ell \in \mathcal{K}_{\text{L}}$, such that for all $a$ and $\ell \in \mathcal{K}_{\text{L}}$, $-\tfrac{\delta_\ell}{2} \le \langle \vec{u}_{\ell,a}, \vec{\rho}_a \rangle - q_\ell \le \tfrac{\delta_\ell}{2}$. Also, for all $a$ and $k \in \mathcal{K}_{\text{LF}}$, $\langle \vec{u}_{k,a}, \vec{\rho}_a \rangle - (q_k + \tfrac{\delta_k}{2})\langle \vec{v}_{k,a}, \vec{\rho}_a \rangle \le 0$, $(q_k - \tfrac{\delta_k}{2})\langle \vec{v}_{k,a}, \vec{\rho}_a \rangle - \langle \vec{u}_{k,a}, \vec{\rho}_a \rangle \le 0$, $\langle \vec{v}_{k,a}, \vec{\rho}_a \rangle > 0 \left. \right\}$.

The (centroid-free) *operating characteristic feasibility region* is then the union $\mathfrak{F}(s, \vec{\delta}) := \bigcup_{\vec{q} \in \mathcal{Q}} \mathfrak{F}(\vec{q}; s, \vec{\delta})$, where $\mathcal{Q}$ is the collection of admissible centroids for the LF constraints (cf. Eq. 31).

---

**Algorithm 6** ROC-F pipeline for $\delta$-approximate demographic parity (DP) and predictive parity (PP)

---

**Require:** Training set $\mathcal{D}_{\text{train}}$, postprocessing set $\mathcal{D}_{\text{post}}$, test set $\mathcal{D}_{\text{test}}$; groups $\mathcal{A}$; tolerances $\delta_{\text{DP}}, \delta_{\text{PP}} \geq 0$; centroid interval $\mathcal{Q}_{\text{PP}} \subset [0,1]$.

**Ensure:** Target operating points $\{\widetilde{\rho}_a\}_{a \in \mathcal{A}}$ and (optionally) a post-processed classifier $\widehat{f}$.

1: **Pre-train predictor.** Fit a probabilistic predictor $s : \mathcal{X} \times \mathcal{A} \to [0,1]$ on $\mathcal{D}_{\text{train}}$.

2: **Postprocess ROC curve.** On $\mathcal{D}_{\text{post}}$, compute probabilistic predictor scores $s(x_i, a_i)$ and retrieve labels $y_i$. For each $a \in \mathcal{A}$: form the empirical ROC curve and keep the upper convex hull supports $\widehat{\mathcal{H}}_a = \{\widehat{\rho}_a^{(j)}\}_{j=1}^{S_a} = \{(\widehat{\text{TPR}}_a^{(j)}, \widehat{\text{FPR}}_a^{(j)})\}_{j=1}^{S_a}$; build lifted points $\widehat{r}_a^{(j)} = (\widehat{\text{TPR}}_a^{(j)}, \widehat{\text{FPR}}_a^{(j)}, 1)^\top$.

   Using $\mathcal{D}_{\text{post}}$, form plug-in coefficients to estimate group rates $\widehat{p}_a$ for $p_a := \Pr(A{=}a)$ and base prevalence rates $\widehat{\pi}_a$ for $\widehat{\pi}_a := \Pr(Y{=}1 \mid A{=}a)$;
   mis-classification rate vector $\widehat{\gamma}_a = \big(-\widehat{p}_a\widehat{\pi}_a, \ \widehat{p}_a(1-\widehat{\pi}_a), \ \widehat{p}_a\widehat{\pi}_a\big)$;
   DP (linear) coefficients: $\widehat{u}_{\text{DP},a} = (\widehat{\pi}_a, 1-\widehat{\pi}_a, 0), \widehat{v}_{\text{DP},a} = (0,0,1)$; and,
   PP (linear-fractional) coefficients: $\widehat{u}_{\text{PP},a} = (\widehat{\pi}_a, 0, 0), \widehat{v}_{\text{PP},a} = (\widehat{\pi}_a, 1-\widehat{\pi}_a, 0)$.

3: **Search region (DP + PP)**

4: **for** $q_{\text{PP}} \in \mathcal{Q}_{\text{PP}}$ **do**                    ▷ coarse grid over PP centroid

5:     **Inner LP (DP+PP):**

$$\min_{\{\lambda_{a,j}\}, \, q_{\text{DP}}} \sum_{a \in \mathcal{A}} \langle \widehat{\gamma}_a, \widehat{\rho}_a \rangle$$

$$\text{s.t.} \quad \begin{cases} \displaystyle\sum_{j \in S_a} \lambda_{a,j} = 1, \quad \lambda_{a,j} \geq 0 \quad \forall j \in S_a, \\[2mm] \displaystyle\widehat{\rho}_a = \sum_{j \in S_a} \lambda_{a,j} \cdot \widehat{r}_a^{(j)} \end{cases} \quad \forall a \in \mathcal{A}$$

(DP as linear constraint via centroid $q_{\text{DP}}$)

$$-\frac{\delta_{\text{DP}}}{2} \leq \langle \widehat{u}_{\text{DP},a}, \widehat{\rho}_a \rangle - q_{\text{DP}} \leq \frac{\delta_{\text{DP}}}{2} \qquad \forall a \in \mathcal{A}$$

(PP as linear–fractional constraint with fixed $q_{\text{PP}}$)

$$\langle \widehat{u}_{\text{PP},a}, \widehat{\rho}_a \rangle - \left(q_{\text{PP}} + \frac{\delta_{\text{PP}}}{2}\right)\langle \widehat{v}_{\text{PP},a}, \widehat{\rho}_a \rangle \leq 0, \qquad \forall a \in \mathcal{A}$$

$$\left(q_{\text{PP}} - \frac{\delta_{\text{PP}}}{2}\right)\langle \widehat{v}_{\text{PP},a}, \widehat{\rho}_a \rangle - \langle \widehat{u}_{\text{PP},a}, \widehat{\rho}_a \rangle \leq 0, \qquad \forall a \in \mathcal{A}$$

(PP denominator positivity / guard)

$$\langle \widehat{v}_{\text{PP},a}, \widehat{\rho}_a \rangle \geq \varepsilon_{\text{PP}} = 1e{-}7 \qquad \forall a \in \mathcal{A}.$$

6:     Let $\widehat{\Phi}(q_{\text{PP}})$ be the optimal value and $\{\lambda_{a,j}\}, \{\widehat{\rho}_a\}, q_{\text{DP}}$ the optimal variables.

7:     **if** $\widehat{\Phi}(q_{\text{PP}}) <$ best **then** best $\leftarrow \widehat{\Phi}(q_{\text{PP}})$, $q_{\text{PP,opt}} \leftarrow q_{\text{PP}}$, $q_{\text{DP,opt}} \leftarrow q_{\text{DP}}$, $\widetilde{\lambda}_{a,j} \leftarrow \lambda_{a,j}$, $\widetilde{\rho}_a \leftarrow \widehat{\rho}_a$.

8:     **end if**

9: **end for**

10: **Return** Return the target operating points $\{\widetilde{\rho}_a\}_{a \in \mathcal{A}}$.

11: **Construct classifier.**

12: **for** group $a \in \mathcal{A}$ **do**

13:     **Compute base prevalence rates of empirical hull.** Extract the subpopulation $\{(x_i, a, y_i) \in \mathcal{D}_{\text{post}}^{(a)}\}$, and use $s(x_i, a)$ to form the convex hull of the (mirrored) empirical ROC:

$$\widehat{\mathcal{H}}_a^{\text{ord}} = \left\{(t_{1,a}, \ldots, t_{S_a,a}) \text{ along with } (\widehat{\text{FNR}}_a^{(j)}, \widehat{\text{FPR}}_a^{(j)}, \widehat{s}_{a,+}^{(j)})\right\}_{j=1}^{S_a};$$

here, $\widehat{s}_{a,+}$ is the plug-in estimate for $s_{a,+}^{(j)} := \Pr(f^{(0)}{=}1 \mid A{=}a)$ where $f^{(0)}(x, a) = 1\,(s(x,a) \geq t_{j,a})$.

14:     best $\leftarrow +\infty$

15:     **for** each adjacent pair $(j, j{+}1)$ in $\widehat{\mathcal{H}}_a^{\text{ord}}$ **do,**

16:         **Search over $\theta$ on this edge** via a 1D line search $G$ (e.g., grid search, golden-section search):

      1. Read off

$$\widehat{\text{FPR}}_{a,\theta}^{(0)} = (1-\theta)\widehat{\text{FPR}}_a^{(j)} + \theta\widehat{\text{FPR}}_a^{(j+1)}, \quad \widehat{\text{FNR}}_{a,\theta}^{(0)} = (1-\theta)\widehat{\text{FNR}}_a^{(j)} + \theta\widehat{\text{FNR}}_a^{(j+1)},$$

      and the baseline selection rate $\widehat{s}_{a,+}(\theta) = (1-\theta)\widehat{s}_{a,+}^{(j)} + \theta\widehat{s}_{a,+}^{(j+1)}$.

      (Continued on next page.)

---

17:     **2. Compute mechanism parameters.**
18:     **if** mode = LF **then**                                              ▷ label-flipping
19:         Define mechanism-specific operating characteristics

$$\text{TPR}_{a,\theta}^{\text{LF}} = (1 - p_{a,1}(\theta))\widehat{\text{TPR}}_{a,\theta}^{(0)} + p_{a,0}(\theta)\big(1 - \widehat{\text{TPR}}_{a,\theta}^{(0)}\big),$$

$$\text{FPR}_{a,\theta}^{\text{LF}} = (1 - p_{a,1}(\theta))\widehat{\text{FPR}}_{a,\theta}^{(0)} + p_{a,0}(\theta)\big(1 - \widehat{\text{FPR}}_{a,\theta}^{(0)}\big).$$

20:         Solve for the solution $p_{a,0}(\theta), p_{a,1}(\theta) \in [0,1]$ of $\text{TPR}_{a,\theta}^{\text{LF}} = \widetilde{\text{TPR}}_a, \text{FPR}_{a,\theta}^{\text{LF}} = \widetilde{\text{FPR}}_a$.
21:         The expected intervention rate is

$$\widetilde{\mathcal{I}}_{a,\theta}^{\text{LF}} = p_{a,1}(\theta)\widehat{s}_{a,+}(\theta) + p_{a,0}(\theta)\big(1 - \widehat{s}_{a,+}(\theta)\big).$$

22:     **end if**
23:     **Update best solution & Assemble outputs.**
24:     **if** $\widetilde{\mathcal{I}}_{a,\theta}^{\text{LF}} < \text{best}$ **then**
25:         set $\theta_a = \theta$ and

$$\vec{\zeta}_a \leftarrow (t_{j,a}, t_{j+1,a}, \theta_a, p_{a,0}(\theta), p_{a,1}(\theta)),$$

26:         and $\text{best} \leftarrow \widetilde{\mathcal{I}}_{a,\theta}^{\text{LF}}$.
27:     **end if**
28:     **end for**
29: **end for**
30: **return** parameter recipes $\{\vec{\zeta}_a\}_{a \in \mathcal{A}}$.
31: **Evaluate.**
32: Construct $\widehat{f}$ from parameter recipes $\{\vec{\zeta}_a\}_{a \in \mathcal{A}}$ by defining a baseline post-processor $f_a^{(0)}$ as

$$f_a^{(0)}(x) = (1 - \theta_a)\mathbb{1}\left(s(x,a) \geq t_{j,a}\right) + \theta_a \mathbb{1}\left(s(x,a) \geq t_{j+1,a}\right),$$

and its positive prediction rate

$$q_a(x) := \Pr\left(f_a^{(0)}(x) = 1 \mid X = x, A = a\right).$$

33: The final (fair) post-processor is

$$\Pr\left(\widehat{f}(x,a) = 1 \mid X = x, A = a\right) = p_{a,0}\big(1 - q_a(x)\big) + p_{a,1}q_a(x).$$

34: On $\mathcal{D}_{\text{test}}$, evaluate $\widehat{f}$ and report loss and the induced DP and PP metrics of

$$\max_{a,a' \in \mathcal{A}} \big| \Pr(\widehat{Y}{=}1|A{=}a) - \Pr(\widehat{Y}{=}1|A{=}a') \big|; \quad \max_{a,a' \in \mathcal{A}} \big| \Pr(Y{=}1|\widehat{Y}{=}1, A{=}a) - \Pr(Y{=}1|\widehat{Y}{=}1, A{=}a') \big|.$$

35: **return** $\{\widetilde{\rho}_a\}$, $\widehat{f}$, and test metrics.

### A.4.1 PROOF OF THEOREM 4.1

Denote the feasible set of the post-processing problem with fairness constraints encoded as pairwise disparities (cf. equation 7 and the discussion below it) as $\mathfrak{F}_0(s, \vec{\delta})$.

We prove this theorem in two parts: (i) first, we argue that $\mathfrak{F}_0(s, \vec{\delta}) = \mathfrak{F}(s, \vec{\delta})$ (cf. Definition A.6); then, (ii) we show that the optimal values are the same.

*(i) Equality of feasible sets.*

**Part 1.** The first required inclusion is that $\mathfrak{F}_0(s, \vec{\delta}) \subseteq \mathfrak{F}(s, \vec{\delta})$.

Suppose $\{\vec{\rho}_a\}_{a \in \mathcal{A}} \in \mathfrak{F}_0(s, \vec{\delta})$. Clearly, for any collection of numbers $\{z_a\}_{a \in \mathcal{A}}$, $z_a \in [0, 1]$ and $\delta \geq 0$,

$$\max_{a, a'} |z_a - z_{a'}| \leq \delta \quad \Longleftrightarrow \quad \exists q \in [0, 1] \text{such that} \max_a |z_a - q| \leq \delta/2. \tag{32}$$

Now consider the disjoint sets of constraints, $\mathcal{K}_L, \mathcal{K}_{LF}$, in turn:

- If $k \in \mathcal{K}_L$, take $\{z_a\}_{a \in \mathcal{A}} = \{\langle \vec{u}_{k,a}, \vec{\rho}_a \rangle\}_{a \in \mathcal{A}}$. Then the pairwise disparity constraints and equation 32 guarantee the existence of $q_k \in \mathcal{Q}_k$ s.t. $-\frac{\delta_k}{2} \leq \langle \vec{u}_{k,a}, \vec{\rho}_a \rangle - q_k \leq \frac{\delta_k}{2}$ for all $a \in \mathcal{A}$. This centroid is admissible since it is associated with a linear constraint.

- If $k \in \mathcal{K}_{LF}$, take $\{z_a\}_{a \in \mathcal{A}} = \{G_{a,k}\}_{a \in \mathcal{A}} = \{\frac{\langle \vec{u}_{k,a}, \vec{\rho}_a \rangle}{\langle \vec{v}_{k,a}, \vec{\rho}_a \rangle}\}_{a \in \mathcal{A}}$, which is well-defined by the denominator positivity constraint.
  Since $\max_a G_{a,k} - \min_a G_{a,k} \leq \delta_k$ for all $k \in \mathcal{K}_{LF} \subseteq \mathcal{K}$ and $z_a \in [0, 1]$, the set

  $$\bigcap_{a \in \mathcal{A}} \left[ z_a - \frac{\delta_k}{2}, z_a + \frac{\delta_k}{2} \right] = \left[ \max_a z_a - \frac{\delta_k}{2}, \min_a z_a + \frac{\delta_k}{2} \right].$$

  is nonempty. Moreover, it is included in $Q_k = [\frac{\delta_k}{2}, 1 - \frac{\delta_k}{2}]$ (admissible set of centroids cf. equation 31).
  Pick any $q_k \in Q_k$ in this overlap. The pairwise disparity constraints $|z_a - q_k| \leq \delta_k/2$ and denominator positivity are then equivalent to

  $$\langle \vec{u}_{k,a}, \vec{\rho}_a \rangle - (q_k + \tfrac{\delta_k}{2})\langle \vec{v}_{k,a}, \vec{\rho}_a \rangle \leq 0, \quad (q_k - \tfrac{\delta_k}{2})\langle \vec{v}_{k,a}, \vec{\rho}_a \rangle - \langle \vec{u}_{k,a}, \vec{\rho}_a \rangle \leq 0$$

  for all $a \in \mathcal{A}$.
  Since $q_k \in Q_k$ for all $k \in \mathcal{K}_{LF}$, we conclude $\vec{q} \in \mathcal{Q}$.

It follows that $\{\vec{\rho}_a\}_{a \in \mathcal{A}} \in \mathfrak{F}(\vec{q}; s, \vec{\delta})$ for some admissible $\vec{q}$.

**Part 2.** The second required inclusion is that $\mathfrak{F}_0(s, \vec{\delta}) \supseteq \mathfrak{F}(s, \vec{\delta})$.

Suppose $\{\vec{\rho}_a\}_{a \in \mathcal{A}} \in \mathfrak{F}(\vec{q}; s, \vec{\delta})$ for some $\vec{q} \in \mathcal{Q}$. Consider again the disjoint sets of constraints $\mathcal{K}_L, \mathcal{K}_{LF}$, in turn.

- If $k \in \mathcal{K}_L$, then $|\langle \vec{u}_{k,a}, \vec{\rho}_a \rangle - q_k| \leq \delta_k/2$ for all $a \in \mathcal{A}$. Hence, for any $a, a'$,

  $$\left| \langle \vec{u}_{k,a}, \vec{\rho}_a \rangle - \langle \vec{u}_{k,a'}, \vec{\rho}_{a'} \rangle \right| \leq \left| \langle \vec{u}_{k,a}, \vec{\rho}_a \rangle - q_k \right| + \left| \langle \vec{u}_{k,a'}, \vec{\rho}_{a'} \rangle - q_k \right| \leq \delta_k$$

  so that the pairwise disparity constraints hold.

- If $k \in \mathcal{K}_{LF}$, then for each $a \in \mathcal{A}$,

  $$\langle \vec{u}_{k,a}, \vec{\rho}_a \rangle - (q_k + \tfrac{\delta_k}{2})\langle \vec{v}_{k,a}, \vec{\rho}_a \rangle \leq 0; \qquad (q_k - \tfrac{\delta_k}{2})\langle \vec{v}_{k,a}, \vec{\rho}_a \rangle - \langle \vec{u}_{k,a}, \vec{\rho}_a \rangle \leq 0.$$

  By denominator positivity, dividing by $V_{k,a}(\vec{\rho}_a)$ gives $q_k - \frac{\delta_k}{2} \leq G_{k,a}(\vec{\rho}_a) \leq q_k + \frac{\delta_k}{2}$, so by the triangle inequality, $|G_{k,a}(\vec{\rho}_a) - G_{k,a'}(\vec{\rho}_{a'})| \leq \delta_k$ for all $a, a'$.

Therefore $\{\vec{\rho}_a\}_{a \in \mathcal{A}} \in \mathfrak{F}_0(s, \vec{\delta})$.

*(ii) Equality of optimal values.* Let $J(\{\vec{\rho}_a\}_{a \in \mathcal{A}})$ denote the linear objective in equation 7. From (i) we have $\mathfrak{F}_0(s, \vec{\delta}) = \mathfrak{F}(s, \vec{\delta}) = \bigcup_{\vec{q} \in \mathcal{Q}} \mathfrak{F}(\vec{q}; s, \vec{\delta})$. Hence

$$\min_{\{\vec{\rho}_a\} \in \mathfrak{F}_0(s, \vec{\delta})} J(\{\vec{\rho}_a\}) = \min_{\{\vec{\rho}_a\} \in \cup_{\vec{q} \in \mathcal{Q}} \mathfrak{F}(\vec{q}; s, \vec{\delta})} J(\{\vec{\rho}_a\}) = \min_{\vec{q} \in \mathcal{Q}} \min_{\{\vec{\rho}_a\} \in \mathfrak{F}(\vec{q}; s, \vec{\delta})} J(\{\vec{\rho}_a\}) = \min_{\vec{q} \in \mathcal{Q}} \Phi(\vec{q}),$$

since the inner problem at fixed $\vec{q}$ is exactly equation 9 with optimal value $\Phi(\vec{q})$. Moreover, the above argument also implies that the minimizers can be recovered as stated in the theorem. This concludes the proof.

### A.4.2 PROOF OF THEOREM 4.2

We begin by introducing relevant notation and then proving two helper lemmas that provide uniform convergence rates for the group-wise operating characteristics (cf. Lemma A.7) and a deviation bound on the fairness constraints using an appropriate Lipschitz constant (cf. Lemma A.8).

**Notation.** For each group $a$, denote the population-level ROC curve as $R_a(s)$ and the empirical convex hull built from $D_{\text{post}}$ as $\widehat{R}_a(s)$ (cf. §A.2.1). For arbitrary lifted operating characteristic rates $\{\vec{\rho}_a\}_{a \in \mathcal{A}}$ with $\vec{\rho}_a \in [0,1]^2 \times \{1\}$, the objective is

$$J(\{\vec{\rho}_a\}_{a \in \mathcal{A}}) = \sum_{a \in \mathcal{A}} \langle \vec{\gamma}_a, \vec{\rho}_a \rangle,$$

with empirical version $\widehat{J}(\{\vec{\rho}_a\}_{a \in \mathcal{A}})$ defined by plugging in empirical coefficients $\widehat{\vec{\gamma}}_a$ constructed from $D_{\text{post}}$. Define the group-wise maximum-$\ell_1$ norm of $\vec{\gamma}_a$ as $B_\gamma := \max_{a \in \mathcal{A}} \|\vec{\gamma}_a\|_1$.

**Lemma A.7** (Uniform control of groupwise rates). *For any confidence level $\delta \in (0,1)$, with probability at least $1 - \delta$, we have simultaneously for all $a \in \mathcal{A}$ and all thresholds $t \in \mathbb{R}$ that*

$$\sup_t \left| \widehat{TPR}_a(t) - TPR_a(t) \right| \leq \tilde{\eta}_{a,1}, \qquad \sup_t \left| \widehat{FPR}_a(t) - FPR_a(t) \right| \leq \tilde{\eta}_{a,0}.$$

*where $\tilde{\eta}_{a,j} = \sqrt{\frac{1}{2n_{a,j}} \log \frac{2m}{\delta}}$ for $j \in \{0,1\}$. Consequently, for every convex combination $\vec{\rho}_a = \sum_j \lambda_j (TPR_a(t_j), FPR_a(t_j), 1)$ and its empirical analog $\widehat{\vec{\rho}_a} = \sum_j \lambda_j (\widehat{TPR}_a(t_j), \widehat{FPR}_a(t_j), 1)$, we have $\|\widehat{\vec{\rho}_a} - \vec{\rho}_a\|_1 \leq \tilde{\eta}_{a,1} + \tilde{\eta}_{a,0}$. In particular, for all $a \in \mathcal{A}$, the Hausdorff distance between $R_a(s)$ and $\widehat{R}_a(s)$ in the $\ell_\infty$ metric is at most $\tilde{\eta}_{a,1} + \tilde{\eta}_{a,0}$.*

*Proof.* Let $F_{a,1}(t) = \Pr\left(s(X,a) \leq t \mid Y = 1, A = a\right)$ and $\widehat{F}_{a,1}$ be its empirical conditional CDF over the $n_{a,1}$ positives in group $a$. By the DKW inequality (Dvoretzky et al., 1956; Massart, 1990), $\Pr\left(\sup_t \left| \widehat{F}_{a,1}(t) - F_{a,1}(t) \right| > \varepsilon\right) \leq 2e^{-2n_{a,1}\varepsilon^2}$. Since $TPR_a(t) = 1 - F_{a,1}(t^-)$ and $\widehat{TPR}_a(t) = 1 - \widehat{F}_{a,1}(t^-)$, the same bound holds for $\sup_t \left| \widehat{TPR}_a(t) - TPR_a(t) \right|$. Setting the RHS of the inequality to $\delta/m$ and applying a union bound yields the result for deviation in the TPR. A similar argument holds bounding the FPR from its empirical version in the $Y = 0$ case.

The convexity statement follows from linearity of convex combinations and the triangle inequality:

$$\|\widehat{\vec{\rho}_a} - \vec{\rho}_a\|_1 \leq \sum_j \lambda_j \left( \left| \widehat{TPR}_a(t_j) - TPR_a(t_j) \right| + \left| \widehat{FPR}_a(t_j) - FPR_a(t_j) \right| \right) \leq \tilde{\eta}_{a,1} + \tilde{\eta}_{a,0}.$$

The Hausdorff claim follows immediately, since we can approximate any $\vec{\rho}_a \in R_a(s)$ with the same mixture in the empirical hull and vice versa. $\square$

Define the following Lipschitz constants

$$L_k := \begin{cases} \max_{a \in \mathcal{A}} \|u_{k,a}\|_\infty, & k \in K_{\mathrm{L}}, \\[2ex] \max_{a \in \mathcal{A}} \dfrac{\|u_{k,a}\|_1 \|v_{k,a}\|_\infty + \|v_{k,a}\|_1 \|u_{k,a}\|_\infty}{\varepsilon_k^2}, & k \in K_{\mathrm{LF}}. \end{cases}$$

**Lemma A.8** (Lipschitz control for linear and linear-fractional fairness metrics). *For $\rho, \rho' \in [0,1]^2 \times \{1\}$ that satisfy the denominator guards $\langle v_{k,a}, \rho \rangle \geq \varepsilon_k$ and $\langle v_{k,a}, \rho \rangle \geq \varepsilon_k$ for $k \in \mathcal{K}_{LF}$,*

$$\left| G_{k,a}(\rho) - G_{k,a}(\rho') \right| \leq L_k \|\rho - \rho'\|_1, \quad \forall k \in \mathcal{K} = \mathcal{K}_L \sqcup \mathcal{K}_{LF}, \tag{33}$$

*where the group-performance metrics (cf. Definition 3.1) $G_{k,a}(\rho_a) = \langle u_{k,a}, \rho_a \rangle$ for $k \in \mathcal{K}_{\mathrm{L}}$ or $G_{k,a}(\rho_a) = \langle u_{k,a}, \rho_a \rangle / \langle v_{k,a}, \rho_a \rangle$ for $k \in \mathcal{K}_{\mathrm{LF}}$.*

*Proof.* For linear constraints $k \in K_{\mathrm{L}}$, the claim is immediate since

$$\big|G_{k,a}(\rho) - G_{k,a}(\rho')\big| = |\langle u_{k,a}, \rho - \rho' \rangle| \leq \|u_{k,a}\|_\infty \|\rho - \rho'\|_1$$

with $L_k = \max_a \|u_{k,a}\|_\infty$.

For linear-fractional constraints $k \in K_{\mathrm{LF}}$, write $U_{k,a}(\rho) := \langle u_{k,a}, \rho \rangle$ and $V_{k,a}(\rho) := \langle v_{k,a}, \rho \rangle$. Now, for any $\rho, \rho' \in [0,1]^2 \times \{1\}$ with $V_{k,a}(\rho), V_{k,a}(\rho') \geq \varepsilon_k$,

$$\left| \frac{U_{k,a}(\rho)}{V_{k,a}(\rho)} - \frac{U_{k,a}(\rho')}{V_{k,a}(\rho')} \right| = \left| \frac{U_{k,a}(\rho)V_{k,a}(\rho') - U_{k,a}(\rho')V_{k,a}(\rho)}{V_{k,a}(\rho)V_{k,a}(\rho')} \right|$$
$$\leq \frac{|U_{k,a}(\rho) - U_{k,a}(\rho')|\,|V_{k,a}(\rho')| + |V_{k,a}(\rho) - V_{k,a}(\rho')|\,|U_{k,a}(\rho')|}{\varepsilon_k^2}.$$

By Hölder's inequality, $|U_{k,a}(\rho) - U_{k,a}(\rho')| \leq \|u_{k,a}\|_\infty \|\rho - \rho'\|_1$ and $|V_{k,a}(\rho) - V_{k,a}(\rho')| \leq \|v_{k,a}\|_\infty \|\rho - \rho'\|_1$, while $|U_{k,a}(\rho')| \leq \|u_{k,a}\|_1$ and $|V_{k,a}(\rho')| \leq \|v_{k,a}\|_1$. Thus

$$\big|G_{k,a}(\rho) - G_{k,a}(\rho')\big| \leq L_k \|\rho - \rho'\|_1,$$

as desired. $\qquad\square$

We can now present the proof of our main result.

**Proof of Theorem 4.2** We work on the intersection of two high–probability events that control (i) uniform convergence of the empirical ROC curves, and (ii) concentration of the plug-in fairness coefficients.

Let $\mathcal{E}_1$ denote the (DKW) event of Lemma A.7 on which

$$\sup_t \big|\widehat{\mathrm{TPR}}_a(t) - \mathrm{TPR}_a(t)\big| \leq \tilde{\eta}_{a,1}, \qquad \sup_t \big|\widehat{\mathrm{FPR}}_a(t) - \mathrm{FPR}_a(t)\big| \leq \tilde{\eta}_{a,0}, \quad \forall a \in \mathcal{A},$$

where $\tilde{\eta}_{a,j} := \sqrt{\frac{1}{2n_{a,j}} \log \frac{4m}{\delta}}$ for $j \in \{0,1\}$; this event holds with probability $\geq 1 - \delta/2$.

Let $\mathcal{E}_2$ denote the Hoeffding event on which all plug-in coefficients for the fairness functionals (group proportions, base rates, and all quantities entering $\widehat{u}_{k,a}, \widehat{v}_{k,a}$, and the objective coefficients $\widehat{\gamma}_a$) satisfy

$$\|\widehat{u}_{k,a} - u_{k,a}\|_\infty, \quad \|\widehat{v}_{k,a} - v_{k,a}\|_\infty, \quad \|\widehat{\gamma}_a - \gamma_a\|_\infty \leq C \cdot \bar{\eta}_a, \quad \forall a \in \mathcal{A}, \ k \in \mathcal{K},$$

where $\bar{\eta}_a := \sqrt{\frac{1}{2n_a} \log \frac{4mK}{\delta}}$, $n_a := n_{a,0} + n_{a,1}$ is the total sample size for group $a$, and $C > 0$ is a constant depending only on $\{u_{k,a}, v_{k,a}, \gamma_a\}$ (but not on $n_{a,j}, m, K$, or $\delta$); by Hoeffding's inequality and a union bound, this event holds with probability $\geq 1 - \delta/2$.

By a union bound, $\Pr(\mathcal{E}_1 \cap \mathcal{E}_2) \geq 1 - \delta$, and we henceforth work on the event $\mathcal{E} := \mathcal{E}_1 \cap \mathcal{E}_2$. Furthermore, for convenience, combine the sample rates as

$$\eta_{a,j} := \tilde{\eta}_{a,j} + C \cdot \bar{\eta}_a, \qquad j \in \{0,1\}.$$

Now, on the event $\mathcal{E}_1$, there exists $\widehat{\varrho}^\sharp = \{\widehat{\rho}_a^\sharp\}_{a \in \mathcal{A}}$ with $\widehat{\rho}_a^\sharp \in \widehat{R}_a(s)$ such that

$$\|\widehat{\rho}_a^\sharp - \rho_a^\star\|_1 \leq \eta_{a,1} + \eta_{a,0}, \qquad \forall a \in \mathcal{A},$$

by the convexity argument in Lemma A.7.

Moreover, for every $a \in \mathcal{A}$ and $k \in \mathcal{K}$, Lemma A.8 guarantees a bound on the deviation of the fairness metrics

$$\big|G_{k,a}(\widehat{\rho}_a^\sharp) - G_{k,a}(\rho_a^\star)\big| \leq L_k(\eta_{a,1} + \eta_{a,0}).$$

Since $\varrho^\star$ satisfies the bands $|G_{k,a}(\rho_a^\star) - q_k| \leq \delta_k/2$ for some centroids $q_k$ (cf. Theorem 4.1), the $\widehat{\varrho}^\sharp$ satisfy

$$\big|G_{k,a}(\widehat{\rho}_a^\sharp) - q_k\big| \leq \frac{\delta_k}{2} + L_k(\eta_{a,1} + \eta_{a,0}) \leq \frac{\delta_k}{2} + L_k \eta_\star, \qquad \eta_\star := \max_a(\eta_{a,1} + \eta_{a,0}).$$

Furthermore, using Lemma A.8, rounding $q_k$ to the nearest grid point on a grid with stepsize $h_k$ introduces an additional slack of at most $L_k h_k$ in $|G_{k,a}(\widehat{\rho}_a^\sharp) - q_k|$. Similarly, using Lemma A.8, the denominator guard contributes at most a further constant multiple of $\varepsilon_k$ to the bands.

Consequently, there exist $C_{\mathrm{grid}}$, $C_\varepsilon$ depending on $\{u_{k,a}, v_{k,a}\}_{a \in \mathcal{A},\, k \in \mathcal{K}}$ and the margins $\{\varepsilon_k\}_{k \in \mathcal{K}}$ such that $\widehat{\rho}^\sharp$ is feasible for the population-level problem with (inflated) bands

$$\delta_k^{\mathrm{infl}} := \delta_k + 2L_k \eta_\star + 2L_k h_k + 2C_\varepsilon \varepsilon_k.$$

*Risk generalization.* Let $\widehat{\varrho} = \{\widehat{\rho}_a\}$ and $\widehat{\varrho}^\dagger = \{\widehat{\rho}_a^\dagger\}$ be the optimizer returned by our empirical region search procedure (Algorithm 2) for nominal disparity levels $\{\delta_k\}$ and $\{\delta_k^{\mathrm{infl}}\}$, respectively. In particular, both procedures use the same grid stepsize, same empirical ROC hulls $\widehat{R}_a(s)$, and same empirical plug-in coefficients.

We now decompose the risk as

$$J(\widehat{\varrho}) - J(\varrho^\star) = \underbrace{J(\widehat{\varrho}) - \widehat{J}(\widehat{\varrho})}_{A} + \underbrace{\widehat{J}(\widehat{\varrho}) - \widehat{J}(\widehat{\varrho}^\dagger)}_{B} + \underbrace{\widehat{J}(\widehat{\varrho}^\dagger) - \widehat{J}(\widehat{\varrho}^\sharp)}_{C} + \underbrace{\widehat{J}(\widehat{\varrho}^\sharp) - J(\widehat{\varrho}^\sharp)}_{D} + \underbrace{J(\widehat{\varrho}^\sharp) - J(\varrho^\star)}_{E}$$

and control each term as follows.

- *Terms A and D.* On the event $\mathcal{E}_2$, for any $\varrho = \{\rho_a\}_{a \in \mathcal{A}}$ with $\vec{\rho}_a \in [0,1]^2 \times \{1\}$, we have

$$|J(\varrho) - \widehat{J}(\varrho)| = \Big| \sum_{a \in \mathcal{A}} \langle \gamma_a - \widehat{\gamma}_a, \rho_a \rangle \Big| \leq B_\gamma \sum_{a \in \mathcal{A}} \|\rho_a\|_1 \|\widehat{\gamma}_a - \gamma_a\|_\infty \leq B_\gamma \sum_{a \in \mathcal{A}} (\eta_{a,1} + \eta_{a,0}),$$

  where $B_\gamma := \max_{a \in \mathcal{A}} \|\gamma_a\|_1$. Applying this with $\varrho = \widehat{\varrho}$ and $\varrho = \widehat{\varrho}^\sharp$ yields

$$|A|, |D| \leq B_\gamma \sum_{a \in \mathcal{A}} (\eta_{a,1} + \eta_{a,0}).$$

- *Term B.* Define the value functions $\widehat{\Phi}_{\mathrm{nom}}(\vec{q})$ and $\widehat{\Phi}_{\mathrm{infl}}(\vec{q})$ as the optimal values of the inner LP of the region search algorithm for fixed centroids $\vec{q}$ under, respectively, nominal and inflated bands. Since $\widehat{\varrho}$ and $\widehat{\varrho}^\dagger$ minimize these value functions over $\vec{q} \in Q$ at centroids $\vec{q}_{\mathrm{nom}}$ and $\vec{q}_{\mathrm{infl}}$, respectively, we have

$$B = \widehat{\Phi}_{\mathrm{nom}}(\vec{q}_{\mathrm{nom}}) - \widehat{\Phi}_{\mathrm{infl}}(\vec{q}_{\mathrm{infl}}) \leq \sup_{\vec{q} \in Q} \big( \widehat{\Phi}_{\mathrm{nom}}(\vec{q}) - \widehat{\Phi}_{\mathrm{infl}}(\vec{q}) \big).$$

  For any fixed $\vec{q}$, the two LPs differ only in the right-hand sides of the fairness-band constraints, with per-coordinate inflation at most $\delta_k^{\mathrm{infl}} - \delta_k = 2L_k \eta_\star + L_k h_k + C_\varepsilon \varepsilon_k$. By standard parametric LP sensitivity (e.g. Bonnans & Shapiro, 2013), there exist constants $C_{\mathrm{stat}}, C_{\mathrm{grid}}, C_\varepsilon > 0$ (depending only on $\{\gamma_a\}$ and $\{u_{k,a}, v_{k,a}\}$) such that

$$0 \leq B \leq C_{\mathrm{stat}} \sum_a (\eta_{a,1} + \eta_{a,0}) + C_{\mathrm{grid}} \sum_k h_k + C_\varepsilon \max_k \varepsilon_k.$$

- *Term C.* On the event $\mathcal{E}_2$, we have a uniform bound

$$\big| \widehat{G}_{k,a}(\rho) - G_{k,a}(\rho) \big| \leq C_{\mathrm{param}} (\eta_{a,1} + \eta_{a,0}) \qquad \forall \rho \in \widehat{R}_a(s),$$

  for some constant $C_{\mathrm{param}} > 0$ depending only on $\{u_{k,a}, v_{k,a}\}$. Since $\widehat{\varrho}^\sharp$ is feasible for the inflated *population* bands, $\widehat{\varrho}^\sharp$ violates the inflated *empirical* bands by at most an amount of order $\sum_a (\eta_{a,1} + \eta_{a,0})$, in addition to the grid and guard terms $\sum_k h_k$ and $\max_k \varepsilon_k$ already accounted for in $\delta_k^{\mathrm{infl}}$. Thus, by a standard LP sensitivity argument, there exist constants $C_{\mathrm{stat}}', C_{\mathrm{grid}}', C_\varepsilon' > 0$ such that

$$|C| = \big| \widehat{J}(\widehat{\varrho}^\dagger) - \widehat{J}(\widehat{\varrho}^\sharp) \big| \leq C_{\mathrm{stat}}' \sum_{a \in \mathcal{A}} (\eta_{a,1} + \eta_{a,0}) + C_{\mathrm{grid}}' \sum_{k \in \mathcal{K}} h_k + C_\varepsilon' \max_{k \in K_{\mathrm{LF}}} \varepsilon_k.$$

- *Term E.* Using linearity of the population objective,

$$|E| = |J(\widehat{\varrho}^\sharp) - J(\varrho^\star)| = \Big| \sum_{a \in \mathcal{A}} \langle \gamma_a, \widehat{\rho}_a^\sharp - \rho_a^\star \rangle \Big| \leq B_\gamma \sum_{a \in \mathcal{A}} \|\widehat{\rho}_a^\sharp - \rho_a^\star\|_1 \leq B_\gamma \sum_{a \in \mathcal{A}} (\eta_{a,1} + \eta_{a,0}),$$

  where the last inequality uses the construction of $\widehat{\varrho}^\sharp$ from Lemma A.7.

Collecting terms, we obtain the desired bound on the risk

$$
\begin{aligned}
J(\widehat{\varrho}) - J(\varrho^\star) &\leq |A| + B + |C| + |D| + |E| \\
&\leq \big(3B_\gamma + C_{\mathrm{stat}} + C'_{\mathrm{stat}}\big) \sum_{a \in \mathcal{A}} (\eta_{a,1} + \eta_{a,0}) + (C_{\mathrm{grid}} + C'_{\mathrm{grid}}) \sum_{k \in \mathcal{K}} h_k \\
&\quad + (C_\varepsilon + C'_\varepsilon) \max_{k \in K_{\mathrm{LF}}} \varepsilon_k \\
&\lesssim \sum_{a \in \mathcal{A}} (\eta_{a,1} + \eta_{a,0}) + \sum_{k \in K_{\mathrm{LF}}} h_k + \max_{k \in K_{\mathrm{LF}}} \varepsilon_k.
\end{aligned}
$$

*Fairness attainment.*

On the event $\mathcal{E}_2$, the plug-in fairness metrics satisfy

$$
\big|\widehat{G}_{k,a}(\rho) - G_{k,a}(\rho)\big| \leq C_{\mathrm{param}}(\eta_{a,1} + \eta_{a,0}), \qquad \forall \rho \in \widehat{R}_a(s).
$$

for some constant $C_{\mathrm{param}} > 0$ depending only on $\{u_{k,a}, v_{k,a}\}$. The triangle inequality then gives

$$
\begin{aligned}
\big|G_{k,a}(\widehat{\rho}_a) - G_{k,a'}(\widehat{\rho}_{a'})\big| &\leq \big|G_{k,a}(\widehat{\rho}_a) - \widehat{G}_{k,a}(\widehat{\rho}_a)\big| + \big|\widehat{G}_{k,a}(\widehat{\rho}_a) - \widehat{G}_{k,a'}(\widehat{\rho}_{a'})\big| \\
&\quad + \big|\widehat{G}_{k,a'}(\widehat{\rho}_{a'}) - G_{k,a'}(\widehat{\rho}_{a'})\big| \\
&\leq \delta_k + C_{\mathrm{param}}\big[(\eta_{a,1} + \eta_{a,0}) + (\eta_{a',1} + \eta_{a',0})\big],
\end{aligned}
$$

so that the maximum pairwise fairness disparity satisfies

$$
\max_{a,a'}\big|G_{k,a}(\widehat{\rho}_a) - G_{k,a'}(\widehat{\rho}_{a'})\big| \lesssim \delta_k + \eta_\star, \qquad \eta_\star := \max_a(\eta_{a,1} + \eta_{a,0}).
$$

This completes the proof.

### A.5 PRACTICAL IMPLEMENTATION DETAILS

We provide here additional implementation details for the experiments in §5.

**Datasets.**

- COMPAS (Larson & Angwin, 2016) is a recidivism dataset where the goal is to predict re-offense within two years. We take race as the protected attribute, restrict to the two largest groups (African-American and Caucasian, $|\mathcal{A}|=2$), and perform preprocessing as in Cho et al. (2020) (e.g., removing traffic offenses and enforcing a screening–arrest window) yielding $5,278$ individuals in the cleaned dataset.

  We start from the ProPublica two-year cohort and retain only the columns [age, c_charge_degree, race, sex, priors_count, days_b_screening_arrest, is_recid, c_jail_in, c_jail_out].

  We remove entries with inconsistent screening/arrest timing by keeping records with days_b_screening_arrest in $[-30, 30]$ days and excluding rows with is_recid $= -1$. We then drop traffic offenses (c_charge_degree = "O"). Next, we compute *length of stay* as the day difference between c_jail_out and c_jail_in, subsequently dropping the original jail timestamp columns and the screening-arrest column. We restrict the analysis to the two largest racial groups (African-American and Caucasian), binarize the label from is_recid, and define groups by race.

  When training the attribute-aware classifier, we remove the sensitive column from $X$ (here, race) and apply one-hot encoding to all remaining categorical features; the protected attribute $A$ is later concatenated as an extra input feature.

- Lawschool (Wightman, 1998; Fabris et al., 2022) is a law school admissions dataset from the Law School Admission Council (LSAC). Following common practice, we keep the core features LSAT, GPA, Gender, resident, a race indicator (e.g., White), and the binary admission label. After removing unused columns and dropping rows with missing values, the resulting dataset contains 96,584 individuals.

Table 4: Training hyperparameters used for the probabilistic predictor $s$.

| dataset | epochs | learning rate | batch size | hidden width | # layers | optimizer |
|---|---|---|---|---|---|---|
| COMPAS | 500 | $5 \times 10^{-4}$ | 2048 | 32 | 3 | Adam |
| Lawschool | 200 | $2 \times 10^{-4}$ | 2048 | 32 | 3 | Adam |
| BiasBios | 20 | $1 \times 10^{-3}$ | 128 | 32 | 3 | Adam |
| ACSIncome | 20 | $1 \times 10^{-3}$ | 128 | 32 | 3 | Adam |

We treat $|\mathcal{A}|{=}2$ race groups as the protected attribute, integer-encode them, and, consistent with our attribute-aware setup, drop the sensitive column from $X$ before concatenating the protected group index as an additional input feature. Numeric features are standardized, and categorical features are one-hot encoded.

- BiasBios (De-Arteaga et al., 2019; Ravfogel et al., 2020) is a large-scale biographical text dataset containing short professional biographies paired with ground truth annotated labels for one of 28 occupation categories (e.g., "physician", "software_engineer", "nurse", "teacher") and an associated binary gender attribute provided in the original metadata. We extract a subset of 25,000 biographies by uniformly sampling from the full corpus of roughly $400,000$ individuals.

  We designate *gender* (female vs. male) as the protected attribute ($|\mathcal{A}|{=}2$), and construct a binary prediction task by grouping the original profession labels into *STEM* and *non-STEM* categories. Specifically, we map {software_engineer, physician, surgeon, dentist, architect, composer} to the STEM class, and assign all remaining occupations to the non-STEM class.

  To handle the text-based data, we encode each biography using a frozen, pretrained DistilBERT encoder by computing one CLS-based embedding per biography and subsequently standardize these embeddings before feeding them to the downstream pre-training and (post-processing) fairness algorithms (Devlin et al., 2019). Ultimately, all experiments operate on a fixed 768-dimensional feature representation.

- ACSIncome (Ding et al., 2021) comes from the 2018 one-Year American Community Survey and contains data on a total of $1,664,500$ individuals. The goal is to predict whether an individual has an income of above or below \$50k. We consider both a binary protected attribute ($|\mathcal{A}|{=}2$) and a multi-group setting ($|\mathcal{A}|{=}5$) by either using gender (Male and Female) or grouping by race. We follow the preprocessing steps of Xian & Zhao (2024) for precise variable definitions and bucketing procedures.

  We use Folktables' ACS (2018, 1-Year, person) slice and apply adult_filter. The feature set includes AGEP, COW, SCHL, MAR, OCCP, POBP, RELP, WKHP, SEX, RAC1P, with category maps supplied for interpretability. For binary classification, the target is $\mathbf{1}\{\text{PINCP} > \$50{,}000\}$. We set the protected attribute to either SEX in the binary protected attribute setting (cf. §5.1), or bin by RACE for the multiple protected attribute setting.

  Races are binned as follows: White (RAC1P=1), Black or African American (2), American Indian or Alaska Native (merge 3,4,5), Asian, Native Hawaiian or Other Pacific Islander (merge 6,7), Other (merge 8,9). We convert to a data frame with dummy variables, map group labels to integer codes, and—consistent with the attribute-aware setup—drop the sensitive columns from $X$ before concatenating the group index as an additional input feature.

**Base classifier** The probabilistic predictor $s$ is a small MLP with three layers and 32 hidden nodes per hidden layer, ending in a sigmoid output. We train with Adam (without weight decay) and binary cross-entropy, using dataset-specific epochs, learning rates, and batch sizes listed in Table 4.

**Baselines.** We compare with two post-processing methods that seek to simultaneously control LF fairness constraints (META and MFOpt), and one state-of-the-art post-processing method that controls for linear fairness constraints (LPP). In addition, we record the performance of the unconstrained probabilistic classifier $s$ (Baseline) and an oracle post-processor returned by the RegionSearch-FG routine of Algorithm 3 (Oracle).

- `Baseline` refers to the probabilistic predictor $s$ with no fairness constraints imposed. It does not use the POST set at all (neither for post-processing/additional calibration nor as additional training data for the TRAIN set) and instead directly evaluates on the TEST set.

- `Oracle` denotes the optimal operating point returned by `RegionSearch-FG` (Algorithm 3) using the TEST set – this requires access to the true labels of the test set. That is, these rates maximize accuracy over operating characteristics but do not correspond to a valid classifier, since they use the test data.[19]

- `META` (Celis et al., 2019) learns a deterministic GWTR (see equation 3) that maximizes classification accuracy subject to fairness constraints involving ratios of LF/F group performance functions (cf. equation 3.1). This meta-algorithm reduces these ratio-based constraints to bounds on group performance functions controlled by a hyperparameter $\tau$; we tune $\tau$ on POST and evaluate the fixed rule on TEST.

- `MFOpt` (Hsu et al., 2022) learns a group-conditional randomized post-processor—i.e., a label-flipping rule parameterized by a $2\times2$ transition matrix per group that maps base predictions to final labels as in Hardt et al. (2016). This method seeks to minimize the expected number of flipped labels (see also §4.3) subject to the fairness constraints on POST. The learned mapping is then fixed and applied to TEST.

- `LPP-DP/EOpp/EO` (Xian & Zhao, 2024) learns a linear post-processor of the base probabilistic predictor $s$ that satisfies $\vec{\delta}$-approximate fairness for common linear metrics (see §3). Though it can handle multiple classes $|\mathcal{Y}| > 2$, specializing it to the binary setting results in a deterministic GWTR, similar to Celis et al. (2019). `LPP-DP/EO` refers to either imposing $\delta$-approximate demographic parity, equality of opportunity, or equalized odds, respectively.

**Baseline configurations**

- `META` (Celis et al., 2019): We implement the group-fair reduction with simultaneous DP, EO (TPR/FPR), and PP. A *ratio band* of width $\tau$ is enforced by sweeping the lower endpoint $a$ on a grid with step $\varepsilon = 0.01$ and setting the upper endpoint to $\min(1, a/\tau)$. We select the band by first determining whether the resulting fairness constraints are satisfied on the POST set, and then choosing the band with the highest accuracy. Finally, we deploy the resulting deterministic score rule on TEST. We use $\tau \in \{0.1, 0.2, \ldots, 1.0\}$ unless stated otherwise.

- `MFOpt` (Hsu et al., 2022): We export the probabilistic predictor 's scores on POST/TEST, run their provided solver on POST to obtain a stochastic matrix mapping the transition rates between bins. Since the optimizer sometimes does not return a row-statistic matrix mapping, we normalize it to ensure a valid classifier. We use this normalized mapping on TEST and report the sampled metrics from the randomized transitions.

- `LPP-DP/EOpp/EO` (Xian & Zhao, 2024): We use the authors' Linear Post-Processing algorithm (`LPP`) to solve the empirical LP on POST with tolerance $\alpha=\delta$ under either demographic parity (`DP`), equality of opportunity (`EOpp`) or equalized odds (`EO`). The resulting (deterministic) decision rule is obtained by linearly adjusting per-class risks, and is then applied unchanged to TEST to report accuracy and disparities. We use CVXPY with GUROBI (fallback SCS) and do not sweep any hyperparameters beyond the nominal $\delta$.

**Sources of randomness.** The stochasticity of the entire procedure is induced by the random TRAIN/POST/TEST split, the stochastic training of $s$, and any randomness introduced by sampled predictions (e.g., `MFOpt` of Hsu et al. (2022) and our randomization schema `LabelFlipping`).

**Configurations for `ROCF-LF` (our method).** For the LF-fairness constraint grid $\mathcal{Q}_{\text{PP}}$ of the `RegionSearch` subroutine (Algorithm 2), we sweep across $q=1000$ equidistantly spaced points within the range of the admissible centroids. For the setting with multiple LF-fairness constraints,

---

[19]The optimality of this operating point depends on how refined the grid-search over the LF fairness constraints is—however, this approximation is a fundamental feature of the centroid linearization technique, see, e.g., Theorem 4.4 of Celis et al. (2019). Our constructed classifiers often achieve these optimal operating characteristics (§5.1-§5.2).

we sweep across $q=100$ equidistant points within the admissible bands $\mathcal{Q}_{\text{PP}}$ and $\mathcal{Q}_{\text{FOR}}$ instead. The denominator positivity margins $\varepsilon_k$ are uniformly set to $10^{-7}$; see §A.5.1 for full details.

We set the bisection search tolerance of the wrapper algorithm `RegionSearch-FG` (Algorithm 3) to $\tau_\alpha=0.01$.

When constructing the minimum intervention classifier (Algorithm 5), we set the snap tolerance parameter $\xi$ to be $0.75$. We use a golden-section search by first evaluating a coarse 101-point uniform grid over the mixing parameter $\theta$ and then, for contiguous feasible intervals, running a golden-section search with tolerance $10^{-5}$ and a cap of 40 iterations.

On a computational level, incorporating an additional LF constraint amounts to an extra (low-dimensional) grid search in `RegionSearch` (Algorithm 2). Though this yields a polynomial-time search in the grid size, we find that a modest, refined grid of $q=100$ equidistant points for both FOR-parity and PP performs well in practice (see §A.5.1 and §A.6.2).

### A.5.1 THEORETICAL AND PRACTICAL IMPLEMENTATION OF COMMON LF CONSTRAINTS

Let $\pi_a = \Pr(Y=1|A=a)$ be the prevalence/base rate. The following conditions ensure equation 8 and denominator positivity ($V_{k,a}(\rho) = \langle \vec{v}_{k,a}, \rho \rangle \geq \varepsilon_k > 0$ for $k \in \mathcal{K}_{\text{LF}}$; see §4.2) are met for predictive parity and false omission rate parity.

*Predictive parity (PP).* For $G_{\text{PP},a} = \frac{\pi_a \text{TPR}_a}{\pi_a \text{TPR}_a + (1-\pi_a)\text{FPR}_a} \in (0,1)$, the centroid band $|G_{\text{PP},a} - q_{\text{PP}}| \leq \delta_{\text{PP}}/2$ is equivalent to the linear inequalities in $(\text{FPR}_a, \text{FNR}_a)$:

$$1 - \text{FNR}_a \in \left[\alpha_a^{\text{lo}}(q_{\text{PP}})\text{FPR}_a, \alpha_a^{\text{hi}}(q_{\text{PP}})\text{FPR}_a\right], \quad \alpha_a^{\text{lo/hi}}(q) = \frac{(q \mp \frac{\delta_{\text{PP}}}{2})(1-\pi_a)}{\left(1 - (q \mp \frac{\delta_{\text{PP}}}{2})\right)\pi_a},$$

which is valid when $(1 - (q \mp \delta_{\text{PP}}/2))\pi_a > 0$ for all $a \in \mathcal{A}$.

Denominator positivity requires $\pi_a(1 - \text{FNR}_a) + (1 - \pi_a)\text{FPR}_a \geq \varepsilon_{\text{PP}} > 0$ for all $a \in \mathcal{A}$.

*False omission rate (FOR).* For $G_{\text{FOR},a} = \frac{\pi_a \text{FNR}_a}{(1-\pi_a)(1-\text{FPR}_a) + \pi_a \text{FNR}_a} \in (0,1)$, let $L = q_{\text{FOR}} - \delta_{\text{FOR}}/2$ and $U = q_{\text{FOR}} + \delta_{\text{FOR}}/2$. The centroid band $|G_{\text{FOR},a} - q_{\text{FOR}}| \leq \delta_{\text{FOR}}/2$ is equivalent to the two linear constraints:

$$(1-L)\pi_a\text{FNR}_a + L(1-\pi_a)\text{FPR}_a \geq L(1-\pi_a), \qquad (1-U)\pi_a\text{FNR}_a + U(1-\pi_a)\text{FPR}_a \leq U(1-\pi_a).$$

Denominator positivity requires $(1 - \pi_a)(1 - \text{FPR}_a) + \pi_a\text{FNR}_a \geq \varepsilon_{\text{FOR}} > 0$ for all $a \in \mathcal{A}$.

**Practical implementation of bands for common LF constraints.** As derived above, admissible centroids require (i) the band coefficients to be well-defined PP : $(1 - (q \pm \delta_{\text{PP}}/2))\pi_a > 0$ and (ii) positivity of the LF denominators

$$\text{PP} : \pi_a(1 - \text{FNR}_a) + (1 - \pi_a)\text{FPR}_a \geq \varepsilon_{\text{PP}}; \qquad \text{FOR} : (1 - \pi_a)(1 - \text{FPR}_a) + \pi_a\text{FNR}_a \geq \varepsilon_{\text{FOR}}.$$

In practice, we perform an outer grid search over $\mathcal{Q}_k$ and impose additional box constraints on the inner-LP of equation 9.

*Intervals and grids.* We ensure band coefficients are well defined by restricting centroids to the outer grids,

$$\mathcal{Q}_{\text{PP}} = [\delta_{\text{PP}}/2, 1-\delta_{\text{PP}}/2], \quad \mathcal{Q}_{\text{FOR}} = [\delta_{\text{FOR}}/2, 1-\delta_{\text{FOR}}/2].$$

- If only one LF metric is active (PP or FOR), we sample a uniform grid of 1000 points over its interval.
- If both PP and FOR are active, we sample 100 PP points and 100 FOR points and iterate over the pairs in their Cartesian product ($100 \times 100$ pairs in total, barring denominator guards).

*LP box-constraints.* To enforce denominator positivity, we add the box-constraints $\text{FPR}_a \geq \varepsilon$, $0 \leq \text{FNR}_a \leq 1$ in the inner-LP of equation 9, with $\varepsilon = 1e{-}7$ set to be negligibly small. This is a slight relaxation; in practice, with $\pi_a \in (0,1)$ and our specified grids $\mathcal{Q}_{\text{PP}}, \mathcal{Q}_{\text{FOR}}$, these boxes are sufficient to ensure the more formal condition ($V_{k,a}(\rho) > 0$) and work well in practice.

### A.6 ADDITIONAL EXPERIMENTAL RESULTS

#### A.6.1 ADDITIONAL RESULTS FOR EXPERIMENTS IN MAIN BODY

We present here experimental results for the binary protected attribute setting on LAWSCHOOL, BIASBIOS, and ACSINCOME and the multiple linear-fractional constraint setting for COMPAS, LAWSCHOOL, and ACSINCOME for completeness; see Tables 5 and 6.

Our method `ROCF-LF` performs favorably on LAWSCHOOL, BIASBIOS, and ACSINCOME with binary protected attributes for the setting where DP, EOpp, PEq, and PP are controlled for at either level 0.02 (for LAWSCHOOL) or 0.05 (for BIASBIOS and ACSINCOME) (Table 5).

For the multiple linear fractional setting, we observe that our method can recover a mixed GWTR with no added randomization as the final classifier and perform as well as state-of-the-art post-processing methods (Table 6(C)).

We also report the behavior of the feasibility guards triggered during our calls to the region search algorithm (Algorithm 3). This algorithm was called on both the post-processing dataset and on the test set (Table 7(A) and (B), respectively), since `ROCF-LF` (or `Oracle`) finds and uses optimal operating characteristics from the post-processing (test) dataset (cf. §5).

For the experiments which had a substantial number of triggers (i.e., the setting of multiple LF fairness constraints for BIASBIOS with binary protected attributes and ACSINCOME with multiple protected attributes), we found that the resulting expansion was quite small ($\alpha \approx 1.48$ and $\alpha \approx 1.04$ at nominal levels 0.05 and 0.10 for BIASBIOS and ACSINCOME, respectively, resulting in an additive constant in the disparity level of about 0.025 and 0.005) and yielded strong empirical results (Tables 3, 6, and 7).

For the experiment for controlling multiple LF constraints on BIASBIOS, the failure of our ORACLE and ROCF-LF methods to achieve the nominal disparity levels of 0.05 is not the result of any inherent shortcoming of their methodology, but rather serves to illustrate the difficulty of simultaneously attaining a nontrivial level of approximate fairness for multiple linear/linear-fractional constraints for this dataset (see also the discussion in §5.2).

On a computational level, any runs that trigger the feasibility guard also require an additional bisection search on the expansion parameter $\alpha$ and thus requires multiple calls to Algorithm 2, but we find that a modest level of tolerance for terminating the search ($\tau_\alpha = 0.01$) performs well (see Tables 3 and 7); for further discussion of computational runtimes, see §A.6.2.

#### A.6.2 COMPUTATIONAL RUNTIMES

In this section, we report wall-clock running times of our method and baselines on the experiments in §5. Figures A.6.2 and A.6.2 display the runtimes measuring the average execution time of a single run per seed for the COMPAS, Lawschool, and ACSIncome datasets. All experiments are conducted on a computing cluster equipped with Intel Xeon Platinum 8375C CPUs @ 2.90GHz processors. For the COMPAS dataset, each run is allowed to use up to 10 CPU cores with 1 GB of RAM, while Lawschool and BiasBios each use up to 2 CPU cores with 8 GB of RAM, and ACSIncome uses a single core with access of up to 32 GB of RAM.

When the nominal levels are well specified and the guard does not trigger, runtimes are modest with small error bars (Figure A.6.2 and Figure A.6.2(B)+(D)). We observe that this setting occurs often in practice; see Table 7. By contrast, under the multiple LF constraint setting for BiasBios with binary protected attributes and ACSIncome with multiple protected attributes, the feasibility guard is often triggered, so we require a bisection section that adds multiple `RegionSearch` (Algorithm 2) solves. This leads to both larger means and larger standard errors in Figure A.6.2(A), (C), and (E).

Although `ROCF-LF` is slower than classic post-processing baselines, the longest observed runtime (on ACSINCOME with $|\mathcal{A}|=5$ and $\approx 1.6M$ individuals) is on the order of minutes ($\approx 60$ minutes) and remains practical for offline model selection. More importantly, these additional solves deliver strong empirical results: `ROCF-LF` attains the nominal disparity levels while achieving nearly optimal accuracy-fairness tradeoffs (cf. §5).

Table 5: Performance on the test set for (A) Lawschool ($|\mathcal{A}|=2$), (B) BiasBios ($|\mathcal{A}| = 2$) and (C) ACSIncome ($|\mathcal{A}|=2$). The disparities $\delta_{\text{DP}}, \delta_{\text{EOpp}}, \delta_{\text{PEq}}, \delta_{\text{PP}}$ are controlled at level 0.02 (for Lawschool) or 0.05 (for BiasBios and ACSIncome) whenever they are active. **Interv.** is the empirical intervention rate on the test set (see Definition A.3). Cells in green indicate that the fairness constraint is satisfied within two standard deviations of the nominal tolerance level, whereas cells in red indicate violation. Entries in the Oracle rows are shaded in lighter colors to denote that they are not practically feasible baselines.

| (A) Lawschool ($|\mathcal{A}|=2$) | | | | | | | |
|---|---|---|---|---|---|---|---|
| Method | Acc | DP | EOpp | PEq | PP | FOR | Interv. |
| Baseline | $0.79 \pm 0.00$ | $0.07 \pm 0.01$ | $0.10 \pm 0.02$ | $0.03 \pm 0.01$ | $0.04 \pm 0.02$ | $0.04 \pm 0.00$ | $0.00 \pm 0.00$ |
| Oracle | $0.78 \pm 0.00$ | $0.02 \pm 0.00$ | $0.01 \pm 0.01$ | $0.01 \pm 0.00$ | $0.02 \pm 0.00$ | $0.06 \pm 0.00$ | N/A |
| **ROCF-LF** (ours) | $0.77 \pm 0.00$ | $0.02 \pm 0.01$ | $0.01 \pm 0.01$ | $0.01 \pm 0.00$ | $0.03 \pm 0.02$ | $0.06 \pm 0.00$ | $0.01 \pm 0.00$ |
| MFOpt | $0.67 \pm 0.00$ | $0.05 \pm 0.01$ | $0.03 \pm 0.02$ | $0.05 \pm 0.01$ | $0.04 \pm 0.01$ | $0.07 \pm 0.01$ | $0.16 \pm 0.01$ |
| META | $0.75 \pm 0.01$ | $0.02 \pm 0.01$ | $0.04 \pm 0.03$ | $0.03 \pm 0.02$ | $0.11 \pm 0.01$ | $0.05 \pm 0.01$ | $0.00 \pm 0.00$ |
| LPP-DP | $0.79 \pm 0.00$ | $0.02 \pm 0.01$ | $0.01 \pm 0.01$ | $0.01 \pm 0.01$ | $0.12 \pm 0.01$ | $0.05 \pm 0.00$ | $0.00 \pm 0.00$ |
| LPP-EO | $0.79 \pm 0.00$ | $0.03 \pm 0.01$ | $0.03 \pm 0.01$ | $0.01 \pm 0.00$ | $0.10 \pm 0.01$ | $0.05 \pm 0.00$ | $0.00 \pm 0.00$ |

| (B) BiasBios ($|\mathcal{A}|=2$) | | | | | | | |
|---|---|---|---|---|---|---|---|
| Method | Acc | DP | EOpp | PEq | PP | FOR | Interv. |
| Baseline | $0.93 \pm 0.00$ | $0.14 \pm 0.01$ | $0.06 \pm 0.03$ | $0.02 \pm 0.01$ | $0.05 \pm 0.03$ | $0.02 \pm 0.01$ | $0.00 \pm 0.00$ |
| Oracle | $0.87 \pm 0.01$ | $0.05 \pm 0.00$ | $0.05 \pm 0.00$ | $0.00 \pm 0.00$ | $0.05 \pm 0.00$ | $0.10 \pm 0.01$ | N/A |
| **ROCF-LF** (ours) | $0.88 \pm 0.01$ | $0.05 \pm 0.01$ | $0.05 \pm 0.03$ | $0.00 \pm 0.00$ | $0.05 \pm 0.01$ | $0.10 \pm 0.01$ | $0.00 \pm 0.00$ |
| MFOpt | $0.71 \pm 0.01$ | $0.12 \pm 0.05$ | $0.05 \pm 0.04$ | $0.08 \pm 0.04$ | $0.15 \pm 0.03$ | $0.07 \pm 0.02$ | $0.15 \pm 0.03$ |
| META | $0.93 \pm 0.00$ | $0.11 \pm 0.02$ | $0.04 \pm 0.03$ | $0.01 \pm 0.01$ | $0.06 \pm 0.03$ | $0.04 \pm 0.01$ | $0.00 \pm 0.00$ |
| LPP-DP | $0.92 \pm 0.00$ | $0.05 \pm 0.01$ | $0.09 \pm 0.02$ | $0.04 \pm 0.01$ | $0.24 \pm 0.03$ | $0.06 \pm 0.01$ | $0.00 \pm 0.00$ |
| LPP-EO | $0.93 \pm 0.00$ | $0.14 \pm 0.01$ | $0.06 \pm 0.03$ | $0.02 \pm 0.01$ | $0.05 \pm 0.03$ | $0.02 \pm 0.01$ | $0.00 \pm 0.00$ |

| (C) ACSIncome ($|\mathcal{A}|=2$) | | | | | | | |
|---|---|---|---|---|---|---|---|
| Method | Acc | DP | EOpp | PEq | PP | FOR | Interv. |
| Baseline | $0.79 \pm 0.01$ | $0.17 \pm 0.02$ | $0.14 \pm 0.02$ | $0.08 \pm 0.03$ | $0.05 \pm 0.01$ | $0.06 \pm 0.02$ | $0.00 \pm 0.00$ |
| Oracle | $0.75 \pm 0.00$ | $0.05 \pm 0.00$ | $0.05 \pm 0.00$ | $0.02 \pm 0.00$ | $0.05 \pm 0.00$ | $0.14 \pm 0.00$ | N/A |
| **ROCF-LF** (ours) | $0.75 \pm 0.00$ | $0.05 \pm 0.00$ | $0.05 \pm 0.00$ | $0.02 \pm 0.00$ | $0.05 \pm 0.00$ | $0.14 \pm 0.00$ | $0.02 \pm 0.00$ |
| MFOpt | $0.74 \pm 0.01$ | $0.20 \pm 0.01$ | $0.09 \pm 0.01$ | $0.15 \pm 0.01$ | $0.06 \pm 0.01$ | $0.07 \pm 0.01$ | $0.06 \pm 0.01$ |
| META | $0.79 \pm 0.00$ | $0.11 \pm 0.02$ | $0.03 \pm 0.02$ | $0.03 \pm 0.01$ | $0.11 \pm 0.01$ | $0.09 \pm 0.01$ | $0.00 \pm 0.00$ |
| LPP-DP | $0.78 \pm 0.01$ | $0.05 \pm 0.00$ | $0.04 \pm 0.01$ | $0.02 \pm 0.01$ | $0.14 \pm 0.01$ | $0.12 \pm 0.01$ | $0.00 \pm 0.00$ |
| LPP-EO | $0.78 \pm 0.01$ | $0.10 \pm 0.01$ | $0.03 \pm 0.01$ | $0.03 \pm 0.01$ | $0.10 \pm 0.01$ | $0.09 \pm 0.01$ | $0.00 \pm 0.00$ |

Table 6: Performance on the test set for (A) COMPAS ($|\mathcal{A}|{=}2$), (B) BiasBios ($|\mathcal{A}|{=}2$), and (C) ACSIncome ($|\mathcal{A}|{=}2$). The disparities $\delta_{\text{EOpp}}, \delta_{\text{PP}}$ and $\delta_{\text{FOR}}$ are controlled at level 0.10 (for COMPAS and ACSIncome) whenever they are active; for the dataset BiasBios, the disparities $\delta_{\text{DP}}, \delta_{\text{PP}}$ and $\delta_{\text{FOR}}$ are controlled at level 0.05 whenever they are active. **Interv.** is the empirical intervention rate on the test set (see Definition A.3). Cells in green indicate that the fairness constraint is satisfied within two standard deviations at the nominal level, whereas cells in red indicate violation. Entries in the Oracle rows are shaded in lighter colors to denote that they are not practically feasible baselines.

| (A) COMPAS ($|\mathcal{A}|{=}2$) | | | | | | |
|---|---|---|---|---|---|---|
| Method | Acc | DP | EOpp | PEq | PP | FOR | Interv. |
| Baseline | $0.68 \pm 0.01$ | $0.28 \pm 0.05$ | $0.27 \pm 0.05$ | $0.19 \pm 0.05$ | $0.07 \pm 0.04$ | $0.03 \pm 0.02$ | $0.00 \pm 0.00$ |
| Oracle | $0.65 \pm 0.04$ | $0.15 \pm 0.04$ | $0.10 \pm 0.01$ | $0.12 \pm 0.09$ | $0.10 \pm 0.01$ | $0.09 \pm 0.01$ | N/A |
| **ROCF-LF** (ours) | $0.65 \pm 0.04$ | $0.15 \pm 0.04$ | $0.11 \pm 0.04$ | $0.11 \pm 0.09$ | $0.11 \pm 0.04$ | $0.09 \pm 0.04$ | $0.01 \pm 0.02$ |
| LPP-EOpp | $0.68 \pm 0.01$ | $0.15 \pm 0.04$ | $0.14 \pm 0.04$ | $0.08 \pm 0.04$ | $0.12 \pm 0.03$ | $0.06 \pm 0.03$ | $0.00 \pm 0.00$ |

| (A) BiasBios ($|\mathcal{A}|{=}2$) | | | | | | |
|---|---|---|---|---|---|---|
| Method | Acc | DP | EOpp | PEq | PP | FOR | Interv. |
| Baseline | $0.93 \pm 0.00$ | $0.14 \pm 0.01$ | $0.06 \pm 0.03$ | $0.02 \pm 0.01$ | $0.05 \pm 0.03$ | $0.02 \pm 0.01$ | $0.00 \pm 0.00$ |
| Oracle | $0.91 \pm 0.00$ | $0.08 \pm 0.00$ | $0.04 \pm 0.02$ | $0.01 \pm 0.00$ | $0.07 \pm 0.01$ | $0.08 \pm 0.00$ | N/A |
| **ROCF-LF** (ours) | $0.91 \pm 0.01$ | $0.08 \pm 0.01$ | $0.04 \pm 0.02$ | $0.01 \pm 0.00$ | $0.08 \pm 0.02$ | $0.07 \pm 0.01$ | $0.00 \pm 0.00$ |
| LPP-DP | $0.92 \pm 0.00$ | $0.05 \pm 0.01$ | $0.09 \pm 0.02$ | $0.04 \pm 0.01$ | $0.24 \pm 0.03$ | $0.06 \pm 0.01$ | $0.00 \pm 0.00$ |

| (C) ACSIncome ($|\mathcal{A}|{=}2$) | | | | | | |
|---|---|---|---|---|---|---|
| Method | Acc | DP | EOpp | PEq | PP | FOR | Interv. |
| Baseline | $0.79 \pm 0.01$ | $0.17 \pm 0.02$ | $0.14 \pm 0.02$ | $0.08 \pm 0.03$ | $0.05 \pm 0.01$ | $0.06 \pm 0.02$ | $0.00 \pm 0.00$ |
| Oracle | $0.79 \pm 0.00$ | $0.16 \pm 0.00$ | $0.10 \pm 0.00$ | $0.07 \pm 0.01$ | $0.07 \pm 0.01$ | $0.07 \pm 0.00$ | N/A |
| **ROCF-LF** (ours) | $0.79 \pm 0.00$ | $0.16 \pm 0.00$ | $0.10 \pm 0.00$ | $0.07 \pm 0.01$ | $0.07 \pm 0.01$ | $0.07 \pm 0.00$ | $0.00 \pm 0.00$ |
| LPP-EOpp | $0.79 \pm 0.01$ | $0.14 \pm 0.01$ | $0.09 \pm 0.01$ | $0.05 \pm 0.02$ | $0.08 \pm 0.01$ | $0.07 \pm 0.01$ | $0.00 \pm 0.00$ |

Table 7: Summary statistics for feasibility guard triggers of `RegionSearch-FG` (Algorithm 3) for the nominal disparity levels considered in §5.1-§5.2; $p$ is the proportion of runs and $\mu$ is the average expansion multiplicative factor $\alpha$ with s.d. and is presented as a tuple $(p, \mu)$. "No expansion (N/E)" is used to indicate that no expansion was needed for all seeds; that is, when listed for an experiment, $(p, \mu) = (0, 1)$ across all seeds.

| (A) `ROCF-LF` | | | | | |
|---|---|---|---|---|---|
| Active $\vec{\delta}$ components | COMPAS | Lawschool | BiasBios | ACSIncome ($|\mathcal{A}|{=}2$) | ACSIncome ($|\mathcal{A}|{=}5$) |
| $\delta_{\text{DP}}, \delta_{\text{EOpp}}, \delta_{\text{PEq}}, \delta_{\text{PP}}$ active | N/E | N/E | N/E | N/E | N/E |
| $\delta_{\text{DP/EOpp}}, \delta_{\text{PP}}, \delta_{\text{FOR}}$ active | $(0.10, 1.01 \pm 0.02)$ | N/E | $(1.00, 1.48 \pm 0.07)$ | N/E | $(0.66, 1.04 \pm 0.05)$ |
| (B) `Oracle` | | | | | |
| Active $\vec{\delta}$ components | COMPAS | Lawschool | BiasBios | ACSIncome ($|\mathcal{A}|{=}2$) | ACSIncome ($|\mathcal{A}|{=}5$) |
| $\delta_{\text{DP}}, \delta_{\text{EOpp}}, \delta_{\text{PEq}}, \delta_{\text{PP}}$ active | N/E | N/E | N/E | N/E | N/E |
| $\delta_{\text{DP/EOpp}}, \delta_{\text{PP}}, \delta_{\text{FOR}}$ active | $(0.08, 1.00 \pm 0.02)$ | N/E | $(1.00, 1.52 \pm 0.08)$ | N/E | $(0.64, 1.05 \pm 0.08)$ |

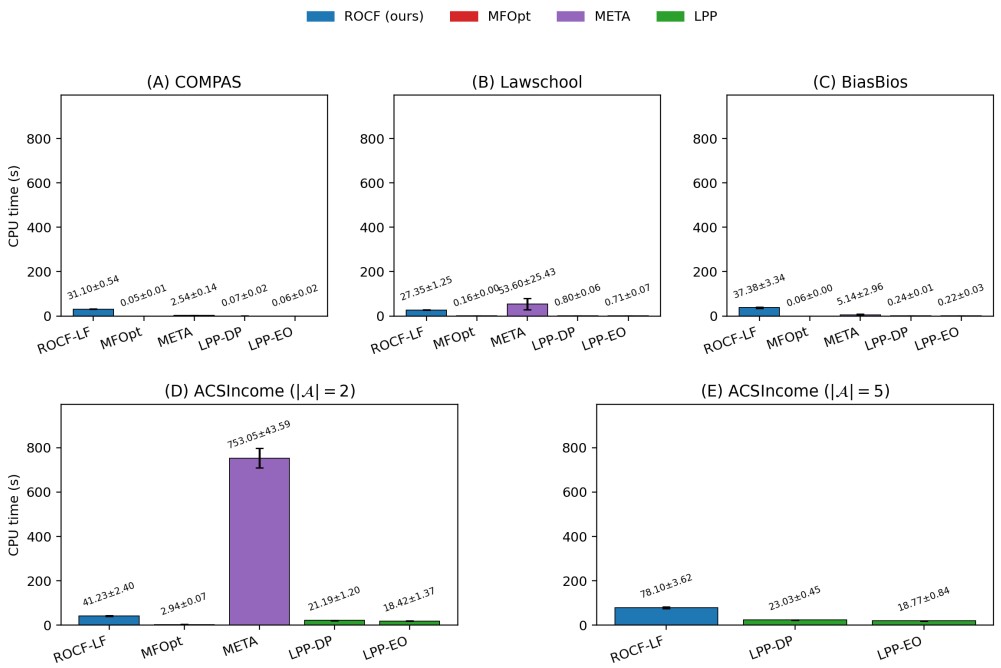

Figure 1: CPU runtime (mean ± s.d., seconds) for (A) COMPAS, (B) Lawschool, (C) BiasBios, (D) ACSIncome ($|\mathcal{A}|=2$), and (E) ACSIncome ($|\mathcal{A}|=5$). As in §5.1 and §A.3.3, the disparities $\delta_{DP}, \delta_{EOpp}, \delta_{PEq}$ and $\delta_{PP}$ are controlled at either level 0.05 (COMPAS, BiasBios, and ACSIncome) or 0.03 (Lawschool) whenever active.

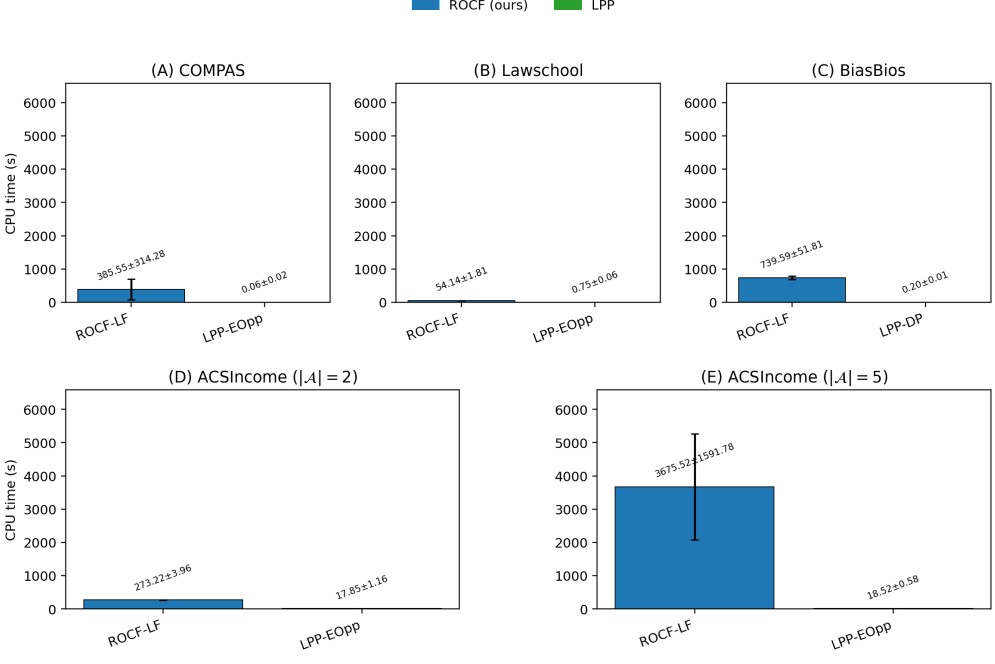

Figure 2: CPU runtime (mean ± s.d., seconds) for (A) COMPAS, (B) Lawschool, (C) BiasBios, (D) ACSIncome ($|\mathcal{A}|=2$), and (E) ACSIncome ($|\mathcal{A}|=5$). As in §5.2, our method ROCF-LF controls for the disparities $\delta_{DP}/\delta_{EOpp}, \delta_{PP}$ and $\delta_{FOR\text{-}parity}$ at either level 0.10 (for COMPAS and ACSIncome), 0.05 (for BiasBios), or 0.03 (for Lawschool), while LPP-DP/EOpp only controls for demographic parity or equality of opportunity, respectively. For full experimental details, please see §5.2.

Table 8: Performance on the test set for ACSIncome ($|\mathcal{A}| = 5$) over various grid sizes. The disparities $\delta_{\text{EOpp}}$, $\delta_{\text{PP}}$ and $\delta_{\text{FOR}}$ are controlled at level $0.10$ whenever they are active. **Interv.** is the empirical intervention rate on the test set (see Definition A.3). Cells in green indicate that the fairness constraint is satisfied within two standard deviations at level $0.10$, whereas cells in red indicate violation. Entries in the Oracle rows are shaded in lighter colors to denote that they are not practically feasible baselines.

| **(A): Grid size 10** | | | | | | | |
|---|---|---|---|---|---|---|---|
| **Method** | **Acc** | **DP** | **EOpp** | **PEq** | **PP** | **FOR** | **Interv.** |
| Baseline | 0.79 ± 0.00 | 0.25 ± 0.01 | 0.24 ± 0.02 | 0.09 ± 0.02 | 0.23 ± 0.02 | 0.08 ± 0.01 | N/A |
| Oracle | 0.66 ± 0.04 | 0.41 ± 0.04 | 0.12 ± 0.01 | 0.41 ± 0.07 | 0.12 ± 0.01 | 0.07 ± 0.03 | N/A |
| **ROCF-LF** (ours) | 0.65 ± 0.02 | 0.42 ± 0.03 | 0.14 ± 0.01 | 0.44 ± 0.05 | 0.13 ± 0.01 | 0.07 ± 0.02 | 0.02 ± 0.02 |

| **(B): Grid size 25** | | | | | | | |
|---|---|---|---|---|---|---|---|
| **Method** | **Acc** | **DP** | **EOpp** | **PEq** | **PP** | **FOR** | **Interv.** |
| Baseline | 0.79 ± 0.00 | 0.25 ± 0.01 | 0.24 ± 0.02 | 0.09 ± 0.02 | 0.23 ± 0.02 | 0.08 ± 0.01 | N/A |
| Oracle | 0.58 ± 0.03 | 0.47 ± 0.03 | 0.11 ± 0.00 | 0.55 ± 0.06 | 0.11 ± 0.01 | 0.09 ± 0.01 | N/A |
| **ROCF-LF** (ours) | 0.58 ± 0.03 | 0.47 ± 0.03 | 0.12 ± 0.01 | 0.55 ± 0.05 | 0.11 ± 0.01 | 0.10 ± 0.02 | 0.05 ± 0.01 |

| **(C): Grid size 50** | | | | | | | |
|---|---|---|---|---|---|---|---|
| **Method** | **Acc** | **DP** | **EOpp** | **PEq** | **PP** | **FOR** | **Interv.** |
| Baseline | 0.79 ± 0.00 | 0.25 ± 0.01 | 0.24 ± 0.02 | 0.09 ± 0.02 | 0.23 ± 0.02 | 0.08 ± 0.01 | N/A |
| Oracle | 0.57 ± 0.03 | 0.48 ± 0.03 | 0.10 ± 0.00 | 0.57 ± 0.05 | 0.10 ± 0.01 | 0.10 ± 0.01 | N/A |
| **ROCF-LF** (ours) | 0.56 ± 0.02 | 0.48 ± 0.03 | 0.11 ± 0.01 | 0.57 ± 0.04 | 0.11 ± 0.01 | 0.12 ± 0.09 | 0.05 ± 0.01 |

| **(D): Grid size 100** | | | | | | | |
|---|---|---|---|---|---|---|---|
| **Method** | **Acc** | **DP** | **EOpp** | **PEq** | **PP** | **FOR** | **Interv.** |
| Baseline | 0.79 ± 0.00 | 0.25 ± 0.01 | 0.24 ± 0.02 | 0.09 ± 0.02 | 0.23 ± 0.02 | 0.08 ± 0.01 | N/A |
| Oracle | 0.57 ± 0.04 | 0.48 ± 0.05 | 0.11 ± 0.01 | 0.56 ± 0.08 | 0.10 ± 0.01 | 0.10 ± 0.01 | N/A |
| **ROCF-LF** (ours) | 0.56 ± 0.02 | 0.49 ± 0.02 | 0.11 ± 0.01 | 0.59 ± 0.04 | 0.11 ± 0.01 | 0.11 ± 0.03 | 0.05 ± 0.01 |

### A.6.3 SENSITIVITY ANALYSIS

Due to centroid linearization (cf. §4.2), we require an outer grid search over admissible centroids for the linear-fractional group fairness constraints. We perform a sensitivity analysis of the effect of the grid's granularity on our method's accuracy performance & fairness attainment (Table 8), computational runtimes (Figure A.6.3), and feasibility guard triggers (Table 9) for the ACSIncome experiment with multiple protected attributes and linear-fractional constraints.

The grid size of either 10, 25, 50, or 100, refers to the number of the equidistantly spaced centroids in both the predictive parity and false omission rate parity bands that are swept over in the outer grid search of Algorithm 2 (e.g., a grid size of $q$ would result in $q \times q$ pairs in total for the grid search). In the related experiment of the main text (cf. §5.2), we use $q = 100$ points.

**Discussion.** We observe that our method's performance is quite poor when we have an overly sparse grid of 10 points: this requires considerable relaxation of the nominal disparity levels and, consequently, does not attain the desired fairness levels even within 2 s.d.s. (see Table 8(A) and first row of Table 9).

In contrast, having a moderately coarse grid works well – even with 25 and 50 grid points, our method achieves similar accuracy performance and fairness attainment for 100 grid points, while exhibiting considerably shorter computational runtimes.

This demonstrates that, though our method is susceptible to overly granular grids (e.g., $q = 10$), a moderately spaced grid performs well in practice and is comparable with the denser grids considered in §5.

### A.6.4 EXPERIMENTAL RESULTS FOR INCREMENTAL EXPANSION POLICY

We present experimental results on COMPAS using a $\Delta$-incremental expansion policy (cf. §A.2.1). Similar to §5.2, we consider the setting of multiple linear fractional constraints, except we now

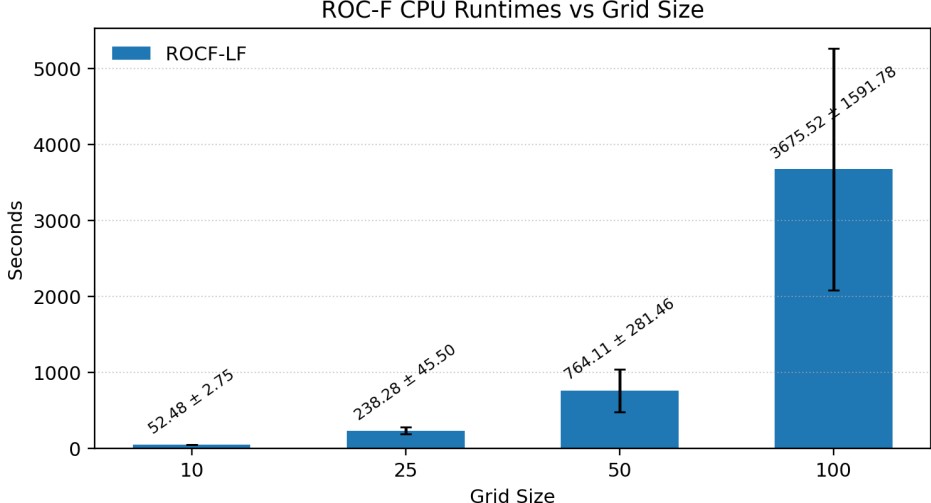

Figure 3: CPU runtime (mean $\pm$ s.d., seconds) for ACSIncome ($|\mathcal{A}|$=5) across various grid sizes. The grid size refers to the number of equidistantly spaced points searched over in the outer grid search of Algorithm 2. As in §5.2, the disparities $\delta_{\text{EOpp}}, \delta_{\text{PP}}$ and $\delta_{\text{FOR-parity}}$ are controlled at level 0.10 for our method ROCF-LF.

Table 9: Summary statistics for feasibility guard triggers of RegionSearch-FG (Algorithm 3) for the nominal disparity levels considered in §5.2 across various grid sizes; $p$ is the proportion of runs and $\mu$ is the average expansion factor $\alpha$ with s.d.

| Grid size | ROCF-LF $p$ | ROCF-LF $\mu$ |
|---|---|---|
| 10 | 1.00 | $1.23 \pm 0.01$ |
| 25 | 0.88 | $1.08 \pm 0.01$ |
| 50 | 0.72 | $1.04 \pm 0.01$ |
| 100 | 0.64 | $1.04 \pm 0.01$ |

wrap an additive expansion policy around our main region search procedure (Algorithm 2) using increments of $\Delta_{\text{EOpp}} = 1e{-}2$ and $\Delta_{\text{PP}} = 5e{-}3$ and constraining the active fairness metrics of $\delta_{\text{EOpp}}, \delta_{\text{PP}}$, and $\delta_{\text{FOR}}$ at nominal levels $0.05, 0.05$ and $0.10$, respectively.

As seen in Table 10, we often need to expand the nominal disparity levels $m = 5$ times in order to find a feasible operating point. For the final attained disparities, we also see that the disparity of predictive parity disparity is lower than that of equality of opportunity, indicating that our non-uniform expansion policy is working as intended.

We would like to note that ORACLE's failure to achieve the nominal equality of opportunity disparity is not the result of any inherent shortcoming of its methodology, but rather serves to illustrate the difficulty of simultaneously attaining a nontrivial level of approximate fairness for multiple linear/linear-fractional constraints for this dataset (see also the discussion in §5.2 and additional experimental results in §A.6).

This preliminary result showcases how practitioners can employ intuitive, non-uniform expansion policies using our method, for which we have proposed and demonstrated the efficacy of a simple additive, incremental procedure.

### A.6.5 EXPERIMENTAL RESULTS ON POST-PROCESSING SET

This section reports the performance of our method along with baselines for various experiments on the post-processing set. By design, our method ROCF-LF attains either the nominal disparity levels when the feasibility guard (cf. §A.2.1) is not activated, or a minimal relaxation of the nominal disparity level when the guard is activated; see Tables 11 and 12.

Table 10: Performance on test set for COMPAS with $\Delta$-incremental expansion policy with $\Delta_{\text{EOpp}} = 0.01$, $\Delta_{\text{PP}} = 0.005$ (see §A.2.1). The disparities $\delta_{\text{EOpp}}, \delta_{\text{PP}}$ and $\delta_{\text{FOR}}$ are controlled at levels $0.05, 0.05$, and $0.10$, respectively, whenever they are active. **Interv.** is the empirical intervention rate on the test set (see Definition A.3). Cells in green indicate that the fairness constraint is satisfied within two standard deviations at the nominal level, whereas cells in red indicate violation. Entries in the Oracle rows are shaded in lighter colors to denote that they are not practically feasible baselines.

| | COMPAS | | | | | | |
|---|---|---|---|---|---|---|---|
| Method | Acc | DP | EOpp | PEq | PP | FOR | Interv. |
| Baseline | $0.68 \pm 0.01$ | $0.28 \pm 0.05$ | $0.27 \pm 0.05$ | $0.20 \pm 0.05$ | $0.07 \pm 0.04$ | $0.03 \pm 0.02$ | $0.00 \pm 0.00$ |
| Oracle | $0.63 \pm 0.04$ | $0.18 \pm 0.05$ | $0.10 \pm 0.02$ | $0.19 \pm 0.10$ | $0.07 \pm 0.01$ | $0.10 \pm 0.00$ | N/A |
| **ROCF-LF** (ours) | $0.62 \pm 0.04$ | $0.18 \pm 0.06$ | $0.12 \pm 0.04$ | $0.19 \pm 0.11$ | $0.08 \pm 0.04$ | $0.10 \pm 0.05$ | $0.03 \pm 0.02$ |
| LPP-EOpp | $0.67 \pm 0.01$ | $0.10 \pm 0.04$ | $0.09 \pm 0.04$ | $0.04 \pm 0.03$ | $0.14 \pm 0.03$ | $0.08 \pm 0.03$ | $0.00 \pm 0.00$ |

Table 11: Performance on the post-processing set for (A) COMPAS ($|\mathcal{A}|=2$), (B) ACSIncome ($|\mathcal{A}|=2$), and (C) ACSIncome ($|\mathcal{A}|=5$). The disparities $\delta_{\text{DP}}, \delta_{\text{EOpp}}, \delta_{\text{PEq}}, \delta_{\text{PP}}$ are controlled at level $0.05$ whenever they are active. **Interv.** is the empirical intervention rate on the post-processing set (see Definition A.3). Cells in green indicate that the fairness constraint is satisfied within two standard deviations at level $0.05$, whereas cells in red indicate violation. Entries in the Oracle rows are shaded in lighter colors to denote that they are not practically feasible baselines.

| | (A) COMPAS | | | | | | |
|---|---|---|---|---|---|---|---|
| Method | Acc | DP | EOpp | PEq | PP | FOR | Interv. |
| Baseline | $0.68 \pm 0.01$ | $0.27 \pm 0.04$ | $0.26 \pm 0.06$ | $0.19 \pm 0.05$ | $0.07 \pm 0.04$ | $0.03 \pm 0.02$ | $0.00 \pm 0.00$ |
| Oracle | $0.62 \pm 0.02$ | $0.05 \pm 0.01$ | $0.03 \pm 0.01$ | $0.05 \pm 0.01$ | $0.05 \pm 0.00$ | $0.15 \pm 0.02$ | N/A |
| **ROCF-LF** (ours) | $0.62 \pm 0.02$ | $0.05 \pm 0.01$ | $0.03 \pm 0.02$ | $0.05 \pm 0.02$ | $0.05 \pm 0.02$ | $0.16 \pm 0.03$ | $0.06 \pm 0.02$ |
| MFOpt | $0.63 \pm 0.01$ | $0.26 \pm 0.04$ | $0.24 \pm 0.05$ | $0.21 \pm 0.04$ | $0.09 \pm 0.03$ | $0.07 \pm 0.03$ | $0.14 \pm 0.02$ |
| META | $0.51 \pm 0.02$ | $0.05 \pm 0.17$ | $0.05 \pm 0.15$ | $0.04 \pm 0.18$ | $0.06 \pm 0.19$ | $0.15 \pm 0.04$ | $0.00 \pm 0.00$ |
| LPP-DP | $0.67 \pm 0.00$ | $0.05 \pm 0.00$ | $0.04 \pm 0.00$ | $0.03 \pm 0.00$ | $0.16 \pm 0.00$ | $0.09 \pm 0.00$ | $0.00 \pm 0.00$ |
| LPP-EO | $0.67 \pm 0.00$ | $0.10 \pm 0.00$ | $0.09 \pm 0.00$ | $0.03 \pm 0.00$ | $0.14 \pm 0.00$ | $0.07 \pm 0.00$ | $0.00 \pm 0.00$ |

| | (B) ACSIncome ($|\mathcal{A}|=2$) | | | | | | |
|---|---|---|---|---|---|---|---|
| Method | Acc | DP | EOpp | PEq | PP | FOR | Interv. |
| Baseline | $0.79 \pm 0.01$ | $0.17 \pm 0.03$ | $0.14 \pm 0.02$ | $0.08 \pm 0.03$ | $0.05 \pm 0.01$ | $0.06 \pm 0.02$ | $0.00 \pm 0.00$ |
| Oracle | $0.75 \pm 0.00$ | $0.05 \pm 0.00$ | $0.05 \pm 0.00$ | $0.02 \pm 0.00$ | $0.05 \pm 0.00$ | $0.14 \pm 0.00$ | N/A |
| **ROCF-LF** (ours) | $0.75 \pm 0.00$ | $0.05 \pm 0.00$ | $0.05 \pm 0.00$ | $0.02 \pm 0.00$ | $0.05 \pm 0.00$ | $0.14 \pm 0.00$ | $0.02 \pm 0.00$ |
| MFOpt | $0.74 \pm 0.00$ | $0.20 \pm 0.00$ | $0.09 \pm 0.00$ | $0.15 \pm 0.00$ | $0.06 \pm 0.00$ | $0.07 \pm 0.00$ | $0.06 \pm 0.00$ |
| META | $0.79 \pm 0.00$ | $0.11 \pm 0.02$ | $0.03 \pm 0.02$ | $0.03 \pm 0.01$ | $0.11 \pm 0.01$ | $0.09 \pm 0.01$ | $0.00 \pm 0.00$ |
| LPP-DP | $0.78 \pm 0.00$ | $0.05 \pm 0.00$ | $0.04 \pm 0.00$ | $0.02 \pm 0.00$ | $0.14 \pm 0.00$ | $0.12 \pm 0.00$ | $0.00 \pm 0.00$ |
| LPP-EO | $0.78 \pm 0.00$ | $0.10 \pm 0.00$ | $0.03 \pm 0.00$ | $0.03 \pm 0.00$ | $0.10 \pm 0.00$ | $0.09 \pm 0.00$ | $0.00 \pm 0.00$ |

| | (C) ACSIncome ($|\mathcal{A}|=5$) | | | | | | |
|---|---|---|---|---|---|---|---|
| Method | Acc | DP | EOpp | PEq | PP | FOR | Interv. |
| Baseline | $0.79 \pm 0.00$ | $0.25 \pm 0.01$ | $0.24 \pm 0.02$ | $0.10 \pm 0.02$ | $0.23 \pm 0.02$ | $0.08 \pm 0.01$ | $0.00 \pm 0.00$ |
| Oracle | $0.69 \pm 0.01$ | $0.05 \pm 0.00$ | $0.05 \pm 0.00$ | $0.03 \pm 0.01$ | $0.05 \pm 0.00$ | $0.23 \pm 0.01$ | N/A |
| **ROCF-LF** (ours) | $0.69 \pm 0.01$ | $0.05 \pm 0.00$ | $0.05 \pm 0.00$ | $0.03 \pm 0.01$ | $0.05 \pm 0.00$ | $0.23 \pm 0.01$ | $0.03 \pm 0.01$ |
| LPP-DP | $0.78 \pm 0.00$ | $0.05 \pm 0.00$ | $0.07 \pm 0.00$ | $0.09 \pm 0.00$ | $0.35 \pm 0.00$ | $0.15 \pm 0.00$ | $0.00 \pm 0.00$ |
| LPP-EO | $0.78 \pm 0.00$ | $0.12 \pm 0.00$ | $0.06 \pm 0.00$ | $0.06 \pm 0.00$ | $0.33 \pm 0.00$ | $0.13 \pm 0.00$ | $0.00 \pm 0.00$ |

Table 12: Performance on the post-processing set for (A) COMPAS ($|\mathcal{A}|{=}2$), (B) ACSIncome ($|\mathcal{A}|{=}2$), and (C) ACSIncome ($|\mathcal{A}|{=}5$). The disparities $\delta_{\text{EOpp}}, \delta_{\text{PP}}$ and $\delta_{\text{FOR}}$ are controlled at level 0.10 whenever they are active. **Interv.** is the empirical intervention rate on the post-processing set (see Definition A.3). Cells in green indicate that the fairness constraint is satisfied within two standard deviations at level 0.10, whereas cells in red indicate violation. Entries in the Oracle rows are shaded in lighter colors to denote that they are not practically feasible baselines.

| | (A) COMPAS | | | | | | |
| Method | Acc | DP | EOpp | PEq | PP | FOR | Interv. |
| --- | --- | --- | --- | --- | --- | --- | --- |
| Baseline | $0.68 \pm 0.01$ | $0.27 \pm 0.04$ | $0.26 \pm 0.06$ | $0.19 \pm 0.05$ | $0.07 \pm 0.04$ | $0.03 \pm 0.02$ | $0.00 \pm 0.00$ |
| Oracle | $0.65 \pm 0.04$ | $0.15 \pm 0.04$ | $0.10 \pm 0.00$ | $0.12 \pm 0.08$ | $0.10 \pm 0.01$ | $0.08 \pm 0.02$ | N/A |
| **ROCF-LF** (ours) | $0.65 \pm 0.04$ | $0.15 \pm 0.04$ | $0.10 \pm 0.01$ | $0.12 \pm 0.09$ | $0.10 \pm 0.01$ | $0.09 \pm 0.03$ | $0.01 \pm 0.02$ |
| LPP-EOpp | $0.67 \pm 0.01$ | $0.15 \pm 0.01$ | $0.13 \pm 0.03$ | $0.07 \pm 0.02$ | $0.12 \pm 0.03$ | $0.06 \pm 0.03$ | $0.00 \pm 0.00$ |

| | (B) ACSIncome ($|\mathcal{A}|{=}2$) | | | | | | |
| Method | Acc | DP | EOpp | PEq | PP | FOR | Interv. |
| --- | --- | --- | --- | --- | --- | --- | --- |
| Baseline | $0.79 \pm 0.01$ | $0.17 \pm 0.03$ | $0.14 \pm 0.02$ | $0.08 \pm 0.03$ | $0.05 \pm 0.01$ | $0.06 \pm 0.02$ | $0.00 \pm 0.00$ |
| Oracle | $0.79 \pm 0.00$ | $0.16 \pm 0.00$ | $0.10 \pm 0.00$ | $0.07 \pm 0.01$ | $0.07 \pm 0.01$ | $0.07 \pm 0.00$ | N/A |
| **ROCF-LF** (ours) | $0.79 \pm 0.00$ | $0.16 \pm 0.00$ | $0.10 \pm 0.00$ | $0.07 \pm 0.01$ | $0.07 \pm 0.01$ | $0.07 \pm 0.00$ | $0.00 \pm 0.00$ |
| LPP-EOpp | $0.79 \pm 0.01$ | $0.14 \pm 0.01$ | $0.09 \pm 0.01$ | $0.05 \pm 0.02$ | $0.08 \pm 0.01$ | $0.07 \pm 0.01$ | $0.00 \pm 0.00$ |

| | (C) ACSIncome ($|\mathcal{A}|{=}5$) | | | | | | |
| Method | Acc | DP | EOpp | PEq | PP | FOR | Interv. |
| --- | --- | --- | --- | --- | --- | --- | --- |
| Baseline | $0.78 \pm 0.01$ | $0.25 \pm 0.01$ | $0.24 \pm 0.03$ | $0.10 \pm 0.02$ | $0.23 \pm 0.03$ | $0.08 \pm 0.01$ | $0.00 \pm 0.00$ |
| Oracle | $0.57 \pm 0.02$ | $0.48 \pm 0.03$ | $0.10 \pm 0.01$ | $0.57 \pm 0.05$ | $0.10 \pm 0.01$ | $0.10 \pm 0.01$ | N/A |
| **ROCF-LF** (ours) | $0.56 \pm 0.02$ | $0.49 \pm 0.03$ | $0.10 \pm 0.01$ | $0.58 \pm 0.04$ | $0.10 \pm 0.01$ | $0.10 \pm 0.01$ | $0.05 \pm 0.01$ |
| LPP-EOpp | $0.79 \pm 0.00$ | $0.15 \pm 0.01$ | $0.11 \pm 0.01$ | $0.06 \pm 0.01$ | $0.31 \pm 0.02$ | $0.12 \pm 0.01$ | $0.00 \pm 0.00$ |

