# OpenReview forum: "Fair Classification by Direct Intervention on Operating Characteristics"
_ICLR.cc/2026/Conference — ICLR 2026 Poster_

### Official Review · Reviewer_keRQ · 2025-10-18

**Soundness:** 4
**Presentation:** 3
**Contribution:** 4
**Rating:** 8
**Confidence:** 4

**Summary:**

This paper introduces ROCF, a post-processing framework that works directly in rate space, leveraging each group’s ROC convex hull to find realizable regions that satisfy multiple fairness constraints (including linear-fractional ones such as PPV parity). The authors propose (i) a centroid-based linearization of linear-fractional constraints that yields a linear program for any fixed centroid, and (ii) a feasibility guard that minimally relaxes constraints when the desired fairness region is infeasible. Empirical evaluation on COMPAS and ACSIncome shows that ROCF achieves approximate DP, EO, PP, and FOR with minimal intervention rate and only small accuracy drops. The method unifies a variety of fairness notions within one efficient LP-based framework.

**Strengths:**

Originality: Nice and elegant reframing of fairness post-processing as optimization directly over operating characteristics; the centroid-based linearization for fractional constraints is novel and mathematically neat.

Quality: Sound algorithmic development and proofs; empirical evidence across fairness notions and with multiple random seeds; good engineering of a “feasibility guard” to guarantee graceful degradation.

Clarity: Clearly written and well-modularized. Rate-space figures and Algorithm listings support understanding.

Significance: Offers a practical, easily implementable method to jointly achieve fairness notions that were previously handled separately. Useful for any group dealing with fairness auditing or deployment.

**Weaknesses:**

Limited scope and datasets: Evaluation restricted to two small tabular datasets (COMPAS, ACSIncome).

Binary-only setting: The framework and proofs assume binary classification with continuous scores; no discussion of multi-class, where scores live in the probabilistic simplex. The assumption that scores are continuous (“ties broken deterministically”) could fail in applications.

Baselines: Comparisons are mostly to post-processing methods (Hardt et al. 2016, Hsu et al. 2022); in-processing reduction approaches (Agarwal et al. 2018) could be included for a broader view.

Feasibility guard uniformity: The relaxation of fairness constraints is uniform; no discussion of prioritizing one constraint (e.g., EO) over another (e.g., PP). This limits interpretability of fairness–utility trade-offs.

Intervention optimality: The AntiDiagonal and LabelFlipping constructions are heuristically minimal but not formally proven globally optimal; empirical stability to hull noise could be further demonstrated.

**Questions:**

Centroid grid sensitivity: How fine must the centroid grid be to approximate the true feasible region? Any quantitative link between grid resolution and sub-optimality?

Denominator constraints: How are $\epsilon_k$ parameters for denominator stability selected, and do they materially alter fairness feasibility for small groups?

Constraint prioritization: Could the feasibility guard relax only selected constraints (e.g., PP but not EO)? A weighted relaxation could make the framework more flexible.

Calibration drift: How robust is ROCF if the base model’s score calibration shifts between training and deployment?

Multi-attribute scalability: How does ROCF scale when there are many protected attributes or intersecting subgroups?

Dataset expansion: Adding a credit-risk or medical dataset would make results more convincing and highlight ROCF’s linear-fractional handling.

AntiDiagonal vs. LabelFlipping: Under what conditions does one yield strictly fewer interventions? An ablation or theorem would clarify.

Oracle definition: Clarify what makes the Oracle baseline “infeasible in practice” (e.g., requiring group labels at test time).

---

> ### Author Response · Authors · 2025-11-20
>
> **D1: Limited scope and datasets.**
> Comment A1 from Reviewer YVAP similarly requested an additional dataset for empirical analyses, so we have copy-pasted part of our response to Reviewer YVAP here for convenience.
>
> We have added new experimental results for another dataset in §5 of the revised manuscript.
> This school-admissions dataset, Lawschool (Wightman 1998; Fabris et al. 2022),
> seeks to predict whether an individual will be admitted to law school, and is a moderately sized dataset containing approximately $96{,}500$ individuals and six covariates including LSAT score, GPA, gender, etc.
>
> Similar to our existing experimental setup,
> we are able to simultaneously control (i) demographic parity, equalized odds, and predictive parity at the $\delta=0.02$ level, and (ii) equality of opportunity, predictive parity, and false-omission rate parity at the $\delta=0.03$ level.
> We have chosen these levels to showcase that our method can have advantages compared to existing methods.
> If we choose higher levels of $\delta$, then all methods can satisfy the constraints,
> whereas if we choose smaller values, then the problem becomes too difficult for any method.
> We have displayed the new results of the various methods' performance on \texttt{Lawschool}
> here in Tables 1and 2 below for convenience.
>
> **Table 1**: Performance on the test set for Lawschool  ($|\mathcal{A}|{=}2$).
> The disparities $\delta_{\textrm{DP}}, \delta_{\textrm{EOpp}}, \delta_{\textrm{PEq}}, \delta_{\textrm{PP}}$ are controlled at level $0.02$ whenever they are active. Cells in green indicate that the fairness constraint is satisfied within two standard deviations at the nominal level, whereas cells in red indicate violation.
>
> | Method      | Acc         | DP             | EOpp           | PEq            | PP             | FOR         | Interv.     |
> | ----------- | ----------- | -------------- | -------------- | -------------- | -------------- | ----------- | ----------- |
> | Baseline    | 0.79 ± 0.00 | 🟥 0.07 ± 0.01 | 🟥 0.10 ± 0.02 | 🟩 0.03 ± 0.01 | 🟩 0.04 ± 0.02 | 0.04 ± 0.00 | 0.00 ± 0.00 |
> | Oracle      | 0.78 ± 0.00 | 🟩 0.02 ± 0.00 | 🟩 0.01 ± 0.01 | 🟩 0.01 ± 0.00 | 🟩 0.02 ± 0.00 | 0.06 ± 0.00 | N/A         |
> | **ROCF-AD** (ours) | 0.77 ± 0.00 | 🟩 0.02 ± 0.01 | 🟩 0.01 ± 0.01 | 🟩 0.01 ± 0.00 | 🟩 0.03 ± 0.03 | 0.06 ± 0.00 | 0.01 ± 0.00 |
> | **ROCF-LF** (ours) | 0.77 ± 0.00 | 🟩 0.02 ± 0.01 | 🟩 0.01 ± 0.01 | 🟩 0.01 ± 0.00 | 🟩 0.03 ± 0.02 | 0.06 ± 0.00 | 0.01 ± 0.00 |
> | MFOpt       | 0.67 ± 0.00 | 🟥 0.05 ± 0.01 | 🟩 0.03 ± 0.02 | 🟥 0.05 ± 0.01 | 🟩 0.04 ± 0.01 | 0.07 ± 0.01 | 0.16 ± 0.01 |
> | META        | 0.75 ± 0.01 | 🟩 0.02 ± 0.01 | 🟩 0.04 ± 0.03 | 🟩 0.03 ± 0.02 | 🟥 0.11 ± 0.01 | 0.05 ± 0.01 | 0.00 ± 0.00 |
> | LPP-DP      | 0.79 ± 0.00 | 🟩 0.02 ± 0.01 | 🟩 0.01 ± 0.01 | 🟩 0.01 ± 0.01 | 🟥 0.12 ± 0.01 | 0.05 ± 0.00 | 0.00 ± 0.00 |
> | LPP-EO      | 0.79 ± 0.00 | 🟩 0.03 ± 0.01 | 🟩 0.03 ± 0.01 | 🟩 0.01 ± 0.00 | 🟥 0.10 ± 0.01 | 0.05 ± 0.00 | 0.00 ± 0.00 |
>
> **Table 2**: Performance on the test set for Lawschool ($|\mathcal{A}|{=}2$). The same protocol as in Table 1 is used, except $\delta_{\textrm{EOpp}}, \delta_{\textrm{PP}}$ and $\delta_{\textrm{FOR}}$ are controlled at level $0.03$.
>
> | Method      | Acc         | DP          | EOpp           | PEq         | PP             | FOR            | Interv.     |
> | ----------- | ----------- | ----------- | -------------- | ----------- | -------------- | -------------- | ----------- |
> | Baseline    | 0.79 ± 0.00 | 0.07 ± 0.01 | 🟥 0.11 ± 0.02 | 0.03 ± 0.01 | 🟥 0.04 ± 0.02 | 🟥 0.04 ± 0.00 | 0.00 ± 0.00 |
> | Oracle      | 0.57 ± 0.03 | 0.16 ± 0.02 | 🟩 0.03 ± 0.00 | 0.16 ± 0.03 | 🟩 0.02 ± 0.01 | 🟩 0.03 ± 0.00 | N/A         |
> | **ROCF-AD** (ours) | 0.58 ± 0.04 | 0.15 ± 0.02 | 🟩 0.03 ± 0.01 | 0.16 ± 0.03 | 🟩 0.02 ± 0.01 | 🟩 0.03 ± 0.01 | 0.01 ± 0.01 |
> | **ROCF-LF** (ours) | 0.58 ± 0.04 | 0.15 ± 0.02 | 🟩 0.03 ± 0.01 | 0.16 ± 0.03 | 🟩 0.02 ± 0.01 | 🟩 0.03 ± 0.01 | 0.01 ± 0.01 |
> | LPP-EOpp    | 0.79 ± 0.00 | 0.04 ± 0.01 | 🟥 0.04 ± 0.02 | 0.01 ± 0.00 | 🟥 0.09 ± 0.01 | 🟥 0.05 ± 0.00 | 0.00 ± 0.00 |
>
> We would like to note that the ACSIncome dataset is already a considerably large dataset with approximately
> 1.6 million data points,
> and has been proposed and used as a novel benchmark for the task of fair classification in algorithmic fairness
> (Ding et al., 2021; Ezzeldin et al., 2023; Xian et al., 2024).
>
> With the added dataset during our rebuttal phase, we have now raised the number of datasets analyzed to three, which meets the standards for diversity, depth, and completeness of data analysis examples in papers in the area of algorithmic fairness published in
> recent ML conferences.
> However, if the reviewer has some specific suggestions for an additional data set to consider, we are happy to consider it within
> reasonable limits.

---

> > ### Author Response · Authors · 2025-11-20
> >
> > **D2: Binary-only setting.**
> > Our assumption that the scores are continuous is not strictly necessary for our proposed procedure in the binary classification setting,
> > as the resulting convex hull of the ROC curve will be the same.
> > We have made this clarification by revising Footnote 8 in §A.2.1.
> >
> > Comment A5 of Reviewer YVAP posed a similar question about the possibility of extending our method to the multi-class setting,
> > but we give a complete response here for convenience.
> >
> > In the revised manuscript, we now provide an extension of our framework (namely, a description of the algorithms) for the multi-class case,
> > following the formalizations of multi-class group fairness constraints in Alghamdi et al., 2022; Xian and Zhao, 2024.
> >
> > At a high level, we show that our method naturally extends to settings with $C>2$ classes by replacing the ROC curve with a polytope of confusion-rate matrices,
> > whose vertices correspond to deterministic multi-class post-processors.
> > We describe how to construct this empirical polytope,
> > how to impose multi-class analogues of demographic parity, equalized odds, and sufficiency constraints,
> > and how to recover a realizable randomized classifier via structured mechanisms (e.g., label-flipping or anti-diagonal mixing).
> >
> > This extension requires no new assumptions; it relies only on convexity and the same polytope-based optimization as in the binary case. Full details are provided in §A.3.5. of the updated manuscript.
> > This analysis suggests that our approach is indeed not fundamentally tied to the binary classification setting, but that similar ideas may extend to the setting of a multi-class classification problem.
> >
> > **D3: Baselines.**
> >
> > We agree that in-processing approaches (e.g., Agarwal et al., 2018; Zhang et al., 2018; Prost et al., 2019)
> > provide an important point of comparison in the broader algorithmic fairness field.
> >
> > However,
> > among post-processing methods that do handle linear constraints,
> > Xian and Zhao (2024) provide an extensive empirical study demonstrating that their post-processing framework performs comparably to state-of-the-art methods (in particular, those of Agarwal et al., 2018 and Zhang et al., 2018).
> > Since our baselines already include this method of Xian and Zhao (2024), our evaluation implicitly covers the performance range of those in-processing approaches.
> >
> > More generally speaking, such state-of-the-art in-processing methods do not support sufficiency-based constraints (i.e., the common linear-fractional group fairness constraints of predictive parity and false omission rate parity),
> > making a larger, direct comparison misaligned with the constraint classes studied in this work.
> >
> > We have added this clarification in the amended introduction of §5.
> >
> > **D4: Feasibility guard uniformity.**
> > The feasibility guard, in full generality, doesn't strictly require a uniform relaxation of the fairness tolerances.
> > For ease of exposition, we chose this uniform expansion since it allows a simple bisection search on the (uniform) expansion factor.
> > One can certainly develop more sophisticated search procedures that prioritize certain fairness constraints,
> > but these are application-specific and outside of the scope of this work (see, also, discussion of feasibility guard in §A.2.1)
> >
> > **D5: Intervention optimality.**
> > Our constructions choose intervention parameters that are optimal conditional on the empirical ROC curve: for a given target operating point, the intervention objectives
> > of equation 24 and 25 (in the revised manuscript)
> > compute the minimal mixing or flipping probabilities needed to achieve that point.
> > In this sense, both _AntiDiagonal_ and _LabelFlipping_ minimize the population-level expected number of interventions for that particular functional form.
> >
> > More generally, we do not claim that these mechanisms are ``globally optimal"
> > among all post-processing rules for minimizing the intervention rate (cf. Def. A.3).
> > Establishing such optimality is challenging, even between our two mechanisms _AntiDiagonal_ and _LabelFlipping_,
> > because the minimal achievable intervention rate depends on the geometry of the empirical ROC curve and the baseline post-processor.
> > Nonetheless,
> > it is still possible to prove finite-sample guarantees of our approach -- the manuscript will be further updated with such a new theoretical result soon (see also, above, the global comment to all reviewers).
> >
> > In light of this,
> > we view our intervention methods
> > as interpretable, analytically tractable, and, most importantly,
> > practically effective instantiations of the broader design space
> > of all post-processing rules.
> >
> > We have added this discussion to §A.3.3 in our revised manuscript for clarification.

---

> > > ### Author Response · Authors · 2025-11-20
> > >
> > > **D6: Centroid grid sensitivity.** Comment B3 from Reviewer v8n8 posed a similar question, and we have added a new section with a sensitivity analysis for the granularity of the grid search over the centroids of linear-fractional constraints in §A.6.3.
> > > For convenience, we present the same results and a similar discussion here.
> > >
> > > We have added a new section with a
> > > sensitivity analysis for the granularity of the grid search over the centroid of the linear-fractional constraints
> > > in §A.6.3 of the updated manuscript. For convenience, we present the same results and a similar discussion here.
> > >
> > > We perform a sensitivity analysis of the effect of the grid's granularity on our method's
> > > accuracy performance \& fairness attainment (Table 1),
> > > computational runtimes (Table 2),
> > > and feasibility guard triggers (Table 3)
> > > for the ACSIncome experiment with multiple protected attributes and linear-fractional constraints.
> > >
> > > The grid size of either $10, 25, 50$, or $100$, refers to the number of the equidistantly spaced centroids in both the predictive parity and false omission rate parity bands
> > > (e.g., for two linear-fractional constraints, a grid size of $q$ would result in $q\times q$ pairs in total for the grid search).
> > > In the original experiment with multiple linear-fractional constraints (cf. §5.1.2), we use $q=100$ points.
> > >
> > > _Discussion_. We observe that our method's performance is quite poor when we have an overly sparse grid of 10 points: this requires considerable relaxation of the nominal disparity levels and, consequently,
> > > does not attain the desired fairness levels even within 2 s.d.s. (see Table 1(A) and first row of Table 3).
> > >
> > > In contrast, having a moderately coarse grid works well -- even with 25 and 50 grid points,
> > > our methods achieve similar accuracy performance and fairness attainment for 100 grid points, while exhibiting considerably shorter computational runtimes (see Tables 1(B)-(D) and Table 2).
> > >
> > > This demonstrates that, though our methods are susceptible to overly granular grids (e.g., $q=10$), a moderately spaced grid performs well in practice and is comparable with the denser grids considered in §5.1.2.
> > >
> > > **Table 1(A):** Performance on the test set for ACSIncome ($|\mathcal{A}|=5$) with $q=10$ grid points.
> > > The disparities $\delta_{\textrm{EOpp}}, \delta_{\textrm{PP}}$ and $\delta_{\textrm{FOR}}$ are controlled at level $0.10$ whenever they are active.
> > > Cells in green indicate that the fairness constraint is satisfied within two standard deviations at level $0.10$, whereas cells in red indicate violation.
> > >
> > > | Method      | Acc         | DP          | EOpp           | PEq         | PP             | FOR            | Interv.     |
> > > | ----------- | ----------- | ----------- | -------------- | ----------- | -------------- | -------------- | ----------- |
> > > | Baseline    | 0.79 ± 0.00 | 0.25 ± 0.01 | 🟥 0.24 ± 0.02 | 0.09 ± 0.02 | 🟥 0.23 ± 0.02 | 🟩 0.08 ± 0.01 | N/A         |
> > > | Oracle      | 0.66 ± 0.04 | 0.41 ± 0.04 | 🟥 0.12 ± 0.01 | 0.41 ± 0.07 | 🟥 0.12 ± 0.01 | 🟩 0.07 ± 0.03 | N/A         |
> > > | **ROCF-AD** (ours) | 0.65 ± 0.02 | 0.42 ± 0.03 | 🟥 0.14 ± 0.01 | 0.44 ± 0.05 | 🟥 0.13 ± 0.01 | 🟩 0.07 ± 0.02 | 0.02 ± 0.02 |
> > > | **ROCF-LF** (ours) | 0.65 ± 0.02 | 0.42 ± 0.03 | 🟥 0.14 ± 0.01 | 0.44 ± 0.05 | 🟥 0.13 ± 0.01 | 🟩 0.07 ± 0.02 | 0.02 ± 0.02 |
> > >
> > > **Table 1(B):** Performance on the test set for ACSIncome ($|\mathcal{A}|=5$) with $q=25$ grid points.
> > > | Method      | Acc         | DP          | EOpp           | PEq         | PP             | FOR            | Interv.     |
> > > | ----------- | ----------- | ----------- | -------------- | ----------- | -------------- | -------------- | ----------- |
> > > | Baseline    | 0.79 ± 0.00 | 0.25 ± 0.01 | 🟥 0.24 ± 0.02 | 0.09 ± 0.02 | 🟥 0.23 ± 0.02 | 🟩 0.08 ± 0.01 | N/A         |
> > > | Oracle      | 0.58 ± 0.03 | 0.47 ± 0.03 | 🟥 0.11 ± 0.00 | 0.55 ± 0.06 | 🟩 0.11 ± 0.01 | 🟩 0.09 ± 0.01 | N/A         |
> > > | **ROCF-AD** (ours) | 0.58 ± 0.03 | 0.47 ± 0.03 | 🟥 0.12 ± 0.01 | 0.55 ± 0.05 | 🟩 0.11 ± 0.01 | 🟩 0.10 ± 0.02 | 0.05 ± 0.01 |
> > > | **ROCF-LF** (ours) | 0.58 ± 0.03 | 0.47 ± 0.03 | 🟥 0.12 ± 0.01 | 0.55 ± 0.05 | 🟩 0.11 ± 0.01 | 🟩 0.10 ± 0.02 | 0.05 ± 0.01 |

---

> > > > ### Author Response · Authors · 2025-11-20
> > > >
> > > > **D6: Centroid grid sensitivity (ctd.).**
> > > >
> > > > **Table 1(D):** Performance on the test set for ACSIncome ($|\mathcal{A}|=5$) with $q=100$ grid points.
> > > >
> > > > | Method      | Acc         | DP          | EOpp           | PEq         | PP             | FOR            | Interv.     |
> > > > | ----------- | ----------- | ----------- | -------------- | ----------- | -------------- | -------------- | ----------- |
> > > > | Baseline    | 0.79 ± 0.00 | 0.25 ± 0.01 | 🟥 0.24 ± 0.02 | 0.09 ± 0.02 | 🟥 0.23 ± 0.02 | 🟩 0.08 ± 0.01 | N/A         |
> > > > | Oracle      | 0.57 ± 0.04 | 0.48 ± 0.05 | 🟩 0.11 ± 0.01 | 0.56 ± 0.08 | 🟩 0.10 ± 0.01 | 🟩 0.10 ± 0.01 | N/A         |
> > > > | **ROCF-AD** (ours) | 0.57 ± 0.02 | 0.48 ± 0.03 | 🟩 0.11 ± 0.01 | 0.57 ± 0.05 | 🟩 0.11 ± 0.01 | 🟩 0.11 ± 0.03 | 0.06 ± 0.01 |
> > > > | **ROCF-LF** (ours) | 0.56 ± 0.02 | 0.49 ± 0.02 | 🟩 0.11 ± 0.01 | 0.59 ± 0.04 | 🟩 0.11 ± 0.01 | 🟩 0.11 ± 0.03 | 0.05 ± 0.01 |
> > > >
> > > > **Table 2:** CPU runtime (mean $\pm$ s.d., seconds) for ACSIncome ($|\mathcal{A}|{=}5$) across various grid sizes. The grid size refers to the number of equidistantly spaced points searched over in the outer grid search for linear-fractional constraints.
> > > > | Grid Size | AD Runtime (s)    | LF Runtime (s)    |
> > > > | --------- | ----------------- | ----------------- |
> > > > | 10        | 48.26 ± 1.83      | 52.48 ± 2.75      |
> > > > | 25        | 255.41 ± 49.35    | 238.28 ± 45.50    |
> > > > | 50        | 789.46 ± 294.07   | 764.11 ± 281.46   |
> > > > | 100       | 3903.40 ± 1550.97 | 3675.52 ± 1591.78 |
> > > >
> > > > **Table 3:** Summary statistics for feasibility guard triggers for the nominal disparity levels considered across various grid sizes;
> > > > $p$ is the proportion of runs and $\mu$ is the average expansion factor with listed s.d.
> > > >
> > > > | Grid Size | ROCF-AD p | ROCF-AD μ   | ROCF-LF p | ROCF-LF μ   |
> > > > | --------- | --------- | ----------- | --------- | ----------- |
> > > > | 10        | 1.00      | 1.23 ± 0.01 | 1.00      | 1.23 ± 0.01 |
> > > > | 25        | 0.88      | 1.08 ± 0.01 | 0.88      | 1.08 ± 0.01 |
> > > > | 50        | 0.72      | 1.04 ± 0.01 | 0.72      | 1.04 ± 0.01 |
> > > > | 100       | 0.64      | 1.05 ± 0.01 | 0.64      | 1.04 ± 0.01 |
> > > >
> > > > **D7: Multi-attribute scalability.** Comment A2 of \YVAP asked a similar question, and we present a similar discussion here for completeness.
> > > >
> > > > _Inner subroutine_: The inner LP subroutine of the grid search does depend on the number of groups, but
> > > > only requires solving a modestly sized LP;
> > > > as discussed in Remark A.2., each iteration of the outer grid search solves an LP with
> > > > $\sum_{a\in\mathcal{A}} |S_a| + |\mathcal{K}\_{\text{L}}|$ decision variables,
> > > > and
> > > > $|\mathcal{A}|+1$ linear constraints
> > > > (for the simplex and post-processing per group),
> > > > $2|\mathcal{A}||\mathcal{K}\_{\text{L}}|$ linear constraints (for the linear centroid bands),
> > > > and $2|\mathcal{A}||\mathcal{K}\_{\text{LF}}|$ linear constraints
> > > > (for the LF bands).
> > > > Here, $|S_a|$ is the number of points in the support of the convex hull of the ROC curve for group $a\in\mathcal{A}$ and is typically small
> > > > both theoretically (Groeneboom et al., 2021),
> > > > and empirically in our experiments, where it usually ranges between values such as 20 to 30.
> > > > Moreover,
> > > >  $|\mathcal{K}\_{\text{LF}}|$ and $|\mathcal{K}\_{\text{L}}|$ are the number of linear-fractional (up to 2 in our experiments)
> > > >  and linear group fairness constraints (up to 3 in our experiments), respectively.
> > > >
> > > >  Thus, for instance, if we have five protected attributes (like in the largest example that we consider), the total number of decision variables is roughly 100-150, and the total number of constraints is roughly 20.
> > > >  This leads to modest-sized linear programs that current computational infrastructure can easily handle.
> > > >
> > > > _Outer search:_ The outer grid search over the centroids of the linear-fractional constraints does not depend on the number of groups $|\mathcal{A}|$,
> > > > since we represent all linear-fractional and linear $\binom{|\mathcal{A}|}{2}$ pairwise disparity constraints as $|\mathcal{A}|$ linear disparity constraints via centroid-based linearization (see, e.g., §4.2).
> > > >
> > > > _Intersectional fairness._ We note that our method is not immune to the curse of dimensionality when considering intersectional subgroups —- however, this phenomenon is an inherent challenge in the broader intersectional fairness literature (Kleinberg, 2018; Gohar & Cheng, 2023).
> > > >
> > > > Nonetheless, techniques developed for the intersectional setting can be readily incorporated into our pipeline; e.g.,
> > > > aggregating / smoothing estimates across related subgroups or  employing adaptive constraint selection to prioritize the most informative intersections.
> > > > These strategies have been used in prior work to mitigate small-sample issues in intersectional analysis, and they apply directly to our framework because our post-processing approach operates entirely on estimated group-conditional operating characteristics. We view extending ROC-F with these intersectional tools as an important direction for future work.

---

> > > > > ### Author Response · Authors · 2025-11-20
> > > > >
> > > > > **D8: Denominator constraints for linear-fractional constraints.** The lower bound imposing the denominator constraints for the linear-fractional constraints are set to be  negligibly small positive constants;
> > > > > in practice, we set $ \varepsilon\_k =1e{-}7$. We have clarified this where these denominator constraints are first introduced (cf. §4.2) and in the appendix (cf. §A.5.1.)
> > > > >
> > > > > **D9: Oracle definition.** The Oracle baseline is infeasible in practice since it requires the true labels of the test set. We have clarified this point in the introduction of §5
> > > > > and in the discussion of baseline methods in §A.5.
> > > > >
> > > > > **References**
> > > > >
> > > > > Agarwal, A., Beygelzimer, A., Dudík, M., Langford, J., & Wallach, H. (2018).
> > > > > A reductions approach to fair classification.
> > > > > International Conference on Machine Learning (ICML), 60–69.
> > > > >
> > > > > Alghamdi, W., Hsu, H., Jeong, H., Wang, H., Michalak, P., Asoodeh, S., & Calmon, F. (2022).
> > > > > Beyond adult and compas: Fair multi-class prediction via information projection.
> > > > > Advances in Neural Information Processing Systems, 35, 38747–38760.
> > > > >
> > > > > Celis, L. E., Huang, L., Keswani, V., & Vishnoi, N. K. (2019).
> > > > > Classification with fairness constraints: A meta-algorithm with provable guarantees.
> > > > > FAT* Conference, 319–328.
> > > > >
> > > > > Ding, F., Hardt, M., Miller, J., & Schmidt, L. (2021).
> > > > > Retiring adult: New datasets for fair machine learning.
> > > > > Advances in Neural Information Processing Systems, 34, 6478–6490.
> > > > >
> > > > > Ezzeldin, Y. H., Yan, S., He, C., Ferrara, E., & Avestimehr, A. S. (2023).
> > > > > FairFed: Enabling group fairness in federated learning.
> > > > > AAAI Conference on Artificial Intelligence, 37(6), 7494–7502.
> > > > >
> > > > > Fabris, A., Messina, S., Silvello, G., & Susto, G. A. (2022).
> > > > > Algorithmic fairness datasets: The story so far.
> > > > > Data Mining and Knowledge Discovery, 36(6), 2074–2152.
> > > > >
> > > > > Gohar, U., & Cheng, L. (2023).
> > > > > A survey on intersectional fairness in machine learning: Notions, mitigation, and challenges.
> > > > > arXiv:2305.06969.
> > > > >
> > > > > Groeneboom, P. (2021).
> > > > > Grenander functionals and Cauchy’s formula.
> > > > > Scandinavian Journal of Statistics, 48(1), 275–294.
> > > > >
> > > > > Hsu, B., Mazumder, R., Nandy, P., & Basu, K. (2022).
> > > > > Pushing the limits of fairness impossibility: Who’s the fairest of them all?
> > > > > Advances in Neural Information Processing Systems, 35, 32749–32761.
> > > > >
> > > > > Kleinberg, J. (2018).
> > > > > Inherent trade-offs in algorithmic fairness.
> > > > > Proceedings of the 2018 ACM SIGMETRICS Conference, 40.
> > > > >
> > > > > Prost, F., Qian, H., Chen, Q., Chi, E. H., Chen, J., & Beutel, A. (2019).
> > > > > Toward a better trade-off between performance and fairness with kernel-based distribution matching.
> > > > > arXiv:1910.11779.
> > > > >
> > > > > Wightman, L. F. (1998).
> > > > > LSAC National Longitudinal Bar Passage Study.
> > > > > LSAC Research Report Series.
> > > > >
> > > > > Xian, R., & Zhao, H. (2024).
> > > > > A unified post-processing framework for group fairness in classification.
> > > > > arXiv:2405.04025.
> > > > >
> > > > > Zhang, B. H., Lemoine, B., & Mitchell, M. (2018).
> > > > > Mitigating unwanted biases with adversarial learning.
> > > > > AAAI/ACM Conference on AI, Ethics, and Society, 335–340.

---

> > ### Comment · Reviewer_keRQ · 2025-11-25
> >
> > I would like to thank the authors for a very thorough and well-structured rebuttal. The additional material—particularly the new Lawschool dataset, the grid-sensitivity analysis, the relocation of technical definitions, and the clarification of denominator guards and the Oracle baseline—substantially improves the clarity and self-containment of the submission. The added theorem in §4.2 and the expanded methodological explanations in the appendix also help contextualize some of the earlier claims. That being said, after carefully reviewing the responses, several aspects of my original evaluation remain only partially addressed:
> >
> > 1. Dataset breadth and empirical scope.
> > Although adding Lawschool strengthens the empirical section, the three datasets used (COMPAS, ACSIncome, Lawschool) remain quite similar in structure: medium-dimensional tabular prediction tasks from U.S. socio-economic domains. The method’s ability to scale (and more importantly, to generalize) to substantially different domains still must be inferred rather than demonstrated. This limits the strength of any general empirical claim.
> >
> > 2. Baselines and positioning relative to in-processing methods.
> > I appreciate the argument that post-processing baselines already dominate or approximate reduction-based in-processing methods, especially for linear constraints. Still, the reasoning remains indirect. Since a central contribution of ROCF is handling heterogeneous combinations of fairness metrics, it would be informative to show at least an ablation where in-processing methods are run on the subset of constraints they can handle, so the reader can see concrete trade-offs in constraint families, accuracy, and interventions. As it stands, the empirical comparison remains narrower than ideal for a paper with broad claims.
> >
> > 3. Multi-class extension.
> > The new multi-class discussion in §A.3.5 is conceptually appealing and suggests natural generalization via confusion-rate polytopes. However, this extension is fully theoretical: no experiments, no complexity analysis, and no discussion of how the explosion in the number of vertices or the dimensionality of the rate polytope impacts the runtime of the outer search or the LP. Without empirical verification, the claim that the framework “naturally extends” to multi-class remains preliminary.
> >
> > 4. Theoretical additions.
> > The new theorem in §4.2 provides reassuring finite-sample bounds conditional on the score function. However, in real deployments, the base score function is rarely calibrated or near-optimal. My original concern (how error in the upstream predictor propagates into fairness feasibility and operating-point estimation) remains open. I appreciate that the authors acknowledge the technical difficulty of such guarantees, but the current theory does not substantially change the conceptual positioning of the work.
> >
> > 5. Practical considerations of the outer search and feasibility guard.
> > The sensitivity analysis is helpful and shows that extremely sparse grids perform poorly while moderate grids (25–50) work well. However, runtime in the multi-attribute, multi-LF setting remains high (several thousand seconds per seed in Table 5). For a post-processing method (whose primary promise is low deployment cost) this overhead is substantial. The feasibility guard is well motivated, but the uniform-relaxation design still limits expressiveness in settings where users may want to prioritize certain constraints.
> >
> > 6. Intervention optimality.
> > The authors clarify that AntiDiagonal and LabelFlipping are conditionally minimal interventions given the empirical ROC point, but not globally minimal. This clarification is appreciated. Nevertheless, one practical question from my review remains open: in which empirical regimes does each mechanism dominate the other?
> >
> > Overall, the authors have clearly improved the paper and clarified many technical points. The rebuttal increases my confidence in the correctness and engineering quality of the approach. However, the revisions do not fundamentally change the scope or generality of the work. My main concerns (dataset diversity, limited empirical baselines, the preliminary nature of the multi-class extension, and the practical cost of the outer search) remain only partially resolved. I therefore maintain my original overall assessment. The paper is strong in motivation, clarity, and methodological elegance, but I will keep my score unchanged.

---

> > > ### Author Response · Authors · 2025-12-02
> > >
> > > Thank you for the very detailed reply to our rebuttal. We have done our best below to address each of your further concerns.
> > >
> > > **1. Dataset breadth and empirical scope.**
> > > 	We have added experimental results for a non-tabular NLP-based dataset, BiasBios (De-Arteaga et al., 2019; Ravfogel et al., 2020), with 25k samples and 768 covariates obtained from a pre-trained, frozen BERT encoder (Devlin et al., 2019).
> > > We found that our method continues to demonstrate strong empirical results (see §5 and §A.6.1. in the main text) and, for convenience, we have copy-pasted the new results here (Tables 4 and 5 below).
> > >
> > > For the experiment for controlling multiple LF constraints on BiasBios (Table 5),
> > > the failure of our Oracle and ROCF-AD/LF methods to achieve the nominal disparity levels of $0.05$ is not the result of any inherent shortcoming of their methodology,
> > > but rather serves to illustrate the difficulty of simultaneously attaining a nontrivial level of
> > > approximate fairness for multiple linear/linear-fractional constraints for this dataset.
> > > In fact, this actually serves to demonstrate that the feasibility guard of our method can find a suitable, minimal relaxation of the user-specified tolerances when they are ill-specified.
> > >
> > > **Table 4:** Performance on the test set for BiasBios ($|\mathcal{A}|=2$). The disparities $\delta_{\textrm{DP}}, \delta_{\textrm{EOpp}}, \delta_{\textrm{PEq}}, \delta_{\textrm{PP}}$ are controlled at level $0.05$ whenever they are active.
> > >
> > > | Method   | Acc         | DP             | EOpp           | PEq            | PP             | FOR         | Interv.     |
> > > | -------- | ----------- | -------------- | -------------- | -------------- | -------------- | ----------- | ----------- |
> > > | Baseline | 0.93 ± 0.00 | 🟥 0.14 ± 0.01 | 🟩 0.06 ± 0.03 | 🟩 0.02 ± 0.01 | 🟩 0.05 ± 0.03 | 0.02 ± 0.01 | 0.00 ± 0.00 |
> > > | Oracle   | 0.87 ± 0.01 | 🟩 0.05 ± 0.00 | 🟩 0.05 ± 0.00 | 🟩 0.00 ± 0.00 | 🟩 0.05 ± 0.00 | 0.10 ± 0.01 | N/A         |
> > > | **ROCF-AD**  | 0.88 ± 0.01 | 🟩 0.05 ± 0.01 | 🟩 0.05 ± 0.03 | 🟩 0.00 ± 0.00 | 🟩 0.06 ± 0.01 | 0.10 ± 0.01 | 0.00 ± 0.00 |
> > > | **ROCF-LF**  | 0.88 ± 0.01 | 🟩 0.05 ± 0.01 | 🟩 0.05 ± 0.03 | 🟩 0.00 ± 0.00 | 🟩 0.05 ± 0.01 | 0.10 ± 0.01 | 0.00 ± 0.00 |
> > > | MFOpt    | 0.71 ± 0.01 | 🟩 0.12 ± 0.05 | 🟩 0.05 ± 0.04 | 🟩 0.08 ± 0.04 | 🟥 0.15 ± 0.03 | 0.07 ± 0.02 | 0.15 ± 0.03 |
> > > | META     | 0.93 ± 0.00 | 🟥 0.11 ± 0.02 | 🟩 0.04 ± 0.03 | 🟩 0.01 ± 0.01 | 🟩 0.06 ± 0.03 | 0.04 ± 0.01 | 0.00 ± 0.00 |
> > > | LPP-DP   | 0.92 ± 0.00 | 🟩 0.05 ± 0.01 | 🟩 0.09 ± 0.02 | 🟩 0.04 ± 0.01 | 🟥 0.24 ± 0.03 | 0.06 ± 0.01 | 0.00 ± 0.00 |
> > > | LPP-EO   | 0.93 ± 0.00 | 🟥 0.14 ± 0.01 | 🟩 0.06 ± 0.03 | 🟩 0.02 ± 0.01 | 🟩 0.05 ± 0.03 | 0.02 ± 0.01 | 0.00 ± 0.00 |
> > >
> > > **Table 5:** Performance on the test set for BiasBios ($|\mathcal{A}|{=}2$). The disparities $\delta_{\textrm{DP}}, \delta_{\textrm{PP}}$ and $\delta_{\textrm{FOR}}$ are controlled at level $0.05$ whenever they are active.
> > >
> > > | Method      | Acc         | DP             | EOpp        | PEq         | PP             | FOR            | Interv.     |
> > > | ----------- | ----------- | -------------- | ----------- | ----------- | -------------- | -------------- | ----------- |
> > > | Baseline    | 0.93 ± 0.00 | 🟥 0.14 ± 0.01 | 0.06 ± 0.03 | 0.02 ± 0.01 | 🟩 0.05 ± 0.03 | 🟩 0.02 ± 0.01 | 0.00 ± 0.00 |
> > > | Oracle      | 0.91 ± 0.00 | 🟥 0.08 ± 0.00 | 0.04 ± 0.02 | 0.01 ± 0.00 | 🟩 0.07 ± 0.01 | 🟥 0.08 ± 0.00 | N/A         |
> > > | **ROCF-AD** | 0.91 ± 0.01 | 🟥 0.08 ± 0.01 | 0.04 ± 0.03 | 0.01 ± 0.00 | 🟩 0.08 ± 0.02 | 🟩 0.07 ± 0.01 | 0.00 ± 0.00 |
> > > | **ROCF-LF** | 0.91 ± 0.01 | 🟥 0.08 ± 0.01 | 0.04 ± 0.02 | 0.01 ± 0.00 | 🟩 0.08 ± 0.02 | 🟩 0.07 ± 0.01 | 0.00 ± 0.00 |
> > > | LPP-DP      | 0.92 ± 0.00 | 🟩 0.05 ± 0.01 | 0.09 ± 0.02 | 0.04 ± 0.01 | 🟥 0.24 ± 0.03 | 🟩 0.06 ± 0.01 | 0.00 ± 0.00 |
> > >
> > > We would also like to argue that basing experimental results around such datasets in the socio-economic domain is considered standard in algorithmic fairness literature -- e.g., COMPAS (Larson \& Angwin, 2016) and AdultCensus (Kohavi, 1996) (from which ACSIncome is based on) have been used as benchmark datasets in this subfield since its inception.
> > >
> > > More broadly speaking, these standards are intended less for immediate practical deployment and more so for establishing the method’s effectiveness. To these ends, we have demonstrated that our approach, ROCF-AD/LF, works and favorably scales across a wide range of sample sizes, as reflected in our experiments on the -- now 4 -- COMPAS (\~5k samples), BiasBios (\~25k samples), Lawschool (\~100k samples), and ACSIncome (\~1.6 million samples) datasets.

---

> > > > ### Author Response · Authors · 2025-12-02
> > > >
> > > > **2. Baselines and positioning relative to in-processing methods.**
> > > > Thank you for raising this concern. We have included preliminary experimental results for the Reductions method of Agarwal et al. 2018 on the COMPAS and BiasBios dataset in Tables 6 and 7 directly below, but, due to time constraints, can not provide the full results across all experiments we considered in our revised manuscript. We use the _ExponentiatedGradient_ algorithm in the Fairlearn reductions package (Bird et al., 2020) with the same learning setup for the base score function (i.e., a 3-layer neural network with 32 hidden nodes at each hidden layer).
> > > > For full transparency, we have updated our source code at this anonymized repo: https://anonymous.4open.science/r/Fair-Classification-via-Feasibility-Regions-38DE .
> > > >
> > > > **Table 6:** Performance on the test set of Reductions (RED-DP/EO; Agarwal et al. 2018) on the COMPAS dataset. The disparities $\delta_{\textrm{DP}}$  and $\delta_{\textrm{EO}}$ are fit at levels $0.05$ and are denoted as RED-DP and RED-EO, respectively.
> > > >
> > > > | Method   | Acc         | DP             | EOpp           | PEq            | PP             | FOR         | Interv.     |
> > > > | -------- | ----------- | -------------- | -------------- | -------------- | -------------- | ----------- | ----------- |
> > > > | Baseline | 0.68 ± 0.01 | 🟥 0.28 ± 0.05 | 🟥 0.27 ± 0.05 | 🟥 0.19 ± 0.05 | 🟩 0.07 ± 0.04 | 0.03 ± 0.02 | 0.00 ± 0.00 |
> > > > | Oracle   | 0.62 ± 0.02 | 🟩 0.04 ± 0.01 | 🟩 0.03 ± 0.02 | 🟩 0.05 ± 0.01 | 🟩 0.05 ± 0.00 | 0.16 ± 0.02 | N/A         |
> > > > | LPP-DP   | 0.67 ± 0.01 | 🟩 0.06 ± 0.03 | 🟩 0.04 ± 0.03 | 🟩 0.03 ± 0.03 | 🟥 0.15 ± 0.03 | 0.09 ± 0.03 | 0.00 ± 0.00 |
> > > > | RED-DP   | 0.64 ± 0.01 | 🟩 0.05 ± 0.03 | 🟩 0.05 ± 0.03 | 🟩 0.03 ± 0.02 | 🟥 0.17 ± 0.03 | 0.10 ± 0.03 | 0.00 ± 0.00 |
> > > > | LPP-EO   | 0.67 ± 0.01 | 🟩 0.10 ± 0.04 | 🟩 0.09 ± 0.04 | 🟩 0.04 ± 0.03 | 🟥 0.14 ± 0.03 | 0.08 ± 0.03 | 0.00 ± 0.00 |
> > > > | RED-EO   | 0.63 ± 0.01 | 🟥 0.11 ± 0.03 | 🟩 0.06 ± 0.04 | 🟩 0.08 ± 0.03 | 🟩 0.08 ± 0.04 | 0.13 ± 0.02 | 0.00 ± 0.00 |
> > > >
> > > > **Table 7:** Performance on the test set of Reductions (RED-DP/EO; Agarwal et al. 2018) on the BiasBios dataset. The disparities $\delta_{\textrm{DP}}$  and $\delta_{\textrm{EO}}$ are fit at levels $0.05$ and are denoted as RED-DP and RED-EO, respectively.
> > > >
> > > > | Method   | Acc         | DP             | EOpp           | PEq            | PP             | FOR         | Interv.     |
> > > > | -------- | ----------- | -------------- | -------------- | -------------- | -------------- | ----------- | ----------- |
> > > > | Baseline | 0.93 ± 0.00 | 🟥 0.14 ± 0.01 | 🟩 0.06 ± 0.03 | 🟩 0.02 ± 0.01 | 🟩 0.05 ± 0.03 | 0.02 ± 0.01 | 0.00 ± 0.00 |
> > > > | Oracle  | 0.87 ± 0.01 | 🟩 0.05 ± 0.00 | 🟩 0.05 ± 0.00 | 🟩 0.00 ± 0.00 | 🟩 0.05 ± 0.00 | 0.10 ± 0.01 | N/A         |
> > > > | LPP-DP   | 0.92 ± 0.00 | 🟩 0.05 ± 0.01 | 🟥 0.09 ± 0.02 | 🟩 0.04 ± 0.01 | 🟥 0.24 ± 0.03 | 0.06 ± 0.01 | 0.00 ± 0.00 |
> > > > | RED-DP   | 0.91 ± 0.00 | 🟩 0.07 ± 0.01 | 🟩 0.04 ± 0.02 | 🟩 0.03 ± 0.01 | 🟥 0.23 ± 0.03 | 0.02 ± 0.01 | 0.00 ± 0.00 |
> > > > | LPP-EO   | 0.93 ± 0.00 | 🟥 0.14 ± 0.01 | 🟩 0.06 ± 0.03 | 🟩 0.02 ± 0.01 | 🟩 0.05 ± 0.03 | 0.02 ± 0.01 | 0.00 ± 0.00 |
> > > > | RED-EO   | 0.93 ± 0.00 | 🟥 0.14 ± 0.01 | 🟩 0.06 ± 0.03 | 🟩 0.02 ± 0.01 | 🟩 0.05 ± 0.03 | 0.02 ± 0.01 | 0.00 ± 0.00 |
> > > >
> > > > Nonetheless, as Tables 6 and 7 here demonstrate, Reductions performs similarly to the linear post-processing method of Xian \& Zhao, 2024 and worse than our methods ROCF-AD/LF, suggesting that a more thorough ablation study would likely reinforce the favorable performance of our ROCF-AD/LF methods.
> > > >
> > > > **3. Multi-class extension.**
> > > >
> > > > 	_Sample complexity_ Our finite-sample guarantees for the binary-class setting can be directly translated to the multi-class case.
> > > > In the binary setting, Theorem 4.3 controls the excess risk and fairness slack in terms of $n_{\min}=\min_{a,j} n_{a,j}$ over group-label counts $j\in\{0,1\}$; in the multi-class setting, these group-label pairs simply become group-class pairs
> > > > $(a,y)$ with $y\in[C]$, so that
> > > > $n_{\min} := \min_{a,y} n_{a,y}$ appears in the bounds.
> > > > The same Hoeffding- and union-bound argument yields a
> > > > $\tilde O(1/\sqrt{n_{\min}})$ rate, which only incurs a mild polynomial dependence on $C$
> > > > due to the $C(C{-}1)$ confusion-rate entries.
> > > > Thus, the guarantees remain conditional on the fixed score function $s$,
> > > > require neither calibration nor Bayes-optimality,
> > > > and preserve the same $1/\sqrt{n}$ convergence rate as in the binary case.

---

> > > > > ### Author Response · Authors · 2025-12-02
> > > > >
> > > > > **3. Multi-class extension (ctd.).** _Computational complexity_ In the multi-class setting, each group-wise confusion–rate matrix lies in an
> > > > > affine space of intrinsic dimension $d = C(C{-}1)$,
> > > > > so the inner problem remains a polynomial-size LP in the
> > > > > convex weights and fairness centroids, and the outer search over
> > > > > $\vec q^{\mathrm{Suff}}\in\prod_{\hat c}\Delta^{C-1}$ has the same structure as
> > > > > in the binary case.
> > > > >
> > > > > The principal open question is whether the empirical
> > > > > hull,
> > > > > constructed from pruning many deterministic classifiers over $\mathcal{D}_{\text{post}}$,
> > > > > remains small in practice, as it does in the
> > > > > binary setting (e.g., $\approx 20$--$30$ support points per group even for datasets
> > > > > with over one million samples like ACSIncome; see also §A.5).
> > > > >
> > > > > A direct analogue of the theory available in the
> > > > > binary class setting (e.g., Grenander-style arguments for the
> > > > > ROC frontier as in Groeneboom, 2021) is not yet available for multi-class confusion-rate matrices. We view developing a full theoretical and/or empirical characterization of the number of support points of this empirical hull as an
> > > > > interesting direction for future work,
> > > > > but note that it
> > > > > does not affect the polynomial-time
> > > > > solvability of each inner-optimization subproblem for any fixed number of classes.
> > > > >
> > > > > Ultimately, we agree that the multi-class extension is primarily conceptual in the current work and, due to time constraints, leave concrete empirical results to future work and have made this clear in the revised manuscript. We have also added a similar discussion clarifying the sample and computational complexity of our multi-class extension in §A.3.4.
> > > > >
> > > > > **4. Theoretical additions.**
> > > > > We respectfully note that our finite-sample guarantees do not assume that the upstream score function is calibrated or near-optimal.
> > > > > In fact,
> > > > > our results explicitly quantify how any level of upstream error propagates into feasibility and operating-point estimation, which we believe answers the  reviewer's original concerns about whether "empirical stability to hull noise could be further demonstrated" and quantifying "[Calibration drift:] how robust is ROCF if the base model’s score calibration shifts between training and deployment?". Moreover, our bounds align with the conceptual claim that if the original score function is a poor predictor, then one should not expect any post-processing method to easily achieve fairness constraints (see, e.g., Kleinberg et al., 2016).
> > > > >
> > > > > With this added clarification (see the revised discussion preceding Theorem 4.2 in §4.2),
> > > > > we believe the new theory meaningfully strengthens the conceptual positioning of the work and fully answers the reviewer's original points about the empirical stability and calibration assumptions of a central object of our study, the convex hull of the ROC curve.
> > > > >
> > > > > **5. Practical considerations of the outer search and feasibility guard.**
> > > > > 	We would like to argue that these overhead costs of our post-processing method are reasonable relative to pre- and in-processing methods that require access to the data used to train a model and/or re-training a model entirely, which may be computationally prohibitive (on the order of days/weeks even on state-of-the-art industry-grade HPC infrastructure)
> > > > > or altogether infeasible in practice (see, e.g.,  Dettmers et al., 2023, Groves et al., 2024, and our discussion in §A.1).
> > > > >
> > > > > Even within the realm of post-processing methods,
> > > > > we also re-emphasize that these larger runtimes
> > > > > are a consequence of the adaptive nature of our method which finds a minimal (uniform) relaxation of the nominal disparity levels when they are ill-specified (Table 2 Row 4, in the author reply directly above) -- when the guard is not activated, the runtime is on the order of a few minutes. Most importantly, our method is, to the best of our knowledge, the first result of its kind to demonstrate strong empirical performance for simultaneously controlling linear and linear-fractional group fairness constraints.
> > > > >
> > > > > We have included an additional experiment using a non-uniform expansion policy that asymmetrically inflates the tolerance of a pair of linear and linear fractional constraints (see §A.6.4). For convenience, we copy-paste the results (Table 8 below) and the discussion here.
> > > > >
> > > > > We present experimental results on COMPAS using a $\Delta$-incremental expansion policy (cf. §A.2.1). Similar to §5.2, we consider the setting of multiple linear fractional constraints,
> > > > > except we now wrap an additive expansion policy around our main region search procedure (Algorithm 2)
> > > > > using increments of $\Delta_{\textrm{EOpp}}=1e{-}2$ and $\Delta_{\textrm{PP}}=5e{-}3$ and constraining the active fairness metrics of $\delta_{\text{EOpp}}, \delta_{\text{PP}}$, and $\delta_{\text{FOR}}$ at nominal levels $0.05,0.05$ and $0.10$, respectively.

---

> > > > > > ### Author Response · Authors · 2025-12-02
> > > > > >
> > > > > > **5. Practical considerations of the outer search and feasibility guard (ctd.).**
> > > > > > As seen in Table 8, we often need to expand the nominal disparity levels $m=5$ times in order to find a feasible operating point.
> > > > > > For the final attained disparities,
> > > > > > we also see that the disparity of predictive parity disparity is lower than that of equality of opportunity, indicating that our non-uniform expansion policy is working as intended.
> > > > > >
> > > > > > We would like to note that \textsc{Oracle}'s failure to achieve the nominal equality of opportunity disparity level of $0.05$
> > > > > > is not the result of any inherent shortcoming of its methodology,
> > > > > > but rather serves to illustrate the difficulty of simultaneously attaining a nontrivial level of
> > > > > > approximate fairness for multiple linear/linear-fractional constraints for this dataset (see also the discussion in §5.2 and additional experimental results in §A.6).
> > > > > >
> > > > > > This preliminary result showcases how practitioners can employ intuitive, non-uniform expansion policies using our method,
> > > > > > for which we have proposed and demonstrated the efficacy of a simple additive, incremental procedure.
> > > > > >
> > > > > > **Table 8:** Performance on test set for COMPAS with $\Delta$-incremental expansion policy with $\Delta_{\text{EOpp}}=0.01$, $\Delta_{\text{PP}}=0.005$. The disparities $\delta_{\textrm{EOpp}}, \delta_{\textrm{PP}}$ and $\delta_{\textrm{FOR}}$ are controlled at levels $0.05, 0.05, $ and $0.10$, respectively, whenever they are active.
> > > > > >
> > > > > > | Method      | Acc         | DP          | EOpp           | PEq         | PP             | FOR            | Interv.     |
> > > > > > | ----------- | ----------- | ----------- | -------------- | ----------- | -------------- | -------------- | ----------- |
> > > > > > | Baseline    | 0.68 ± 0.01 | 0.28 ± 0.05 | 🟥 0.27 ± 0.05 | 0.20 ± 0.05 | 🟩 0.07 ± 0.04 | 🟩 0.03 ± 0.02 | 0.00 ± 0.00 |
> > > > > > | Oracle      | 0.63 ± 0.04 | 0.18 ± 0.05 | 🟥 0.10 ± 0.02 | 0.19 ± 0.10 | 🟩 0.07 ± 0.01 | 🟩 0.10 ± 0.00 | N/A         |
> > > > > > | **ROCF-AD** | 0.62 ± 0.04 | 0.19 ± 0.06 | 🟩 0.12 ± 0.04 | 0.19 ± 0.11 | 🟩 0.08 ± 0.03 | 🟩 0.10 ± 0.05 | 0.02 ± 0.02 |
> > > > > > | **ROCF-LF** | 0.62 ± 0.04 | 0.18 ± 0.06 | 🟩 0.12 ± 0.04 | 0.19 ± 0.11 | 🟩 0.08 ± 0.04 | 🟩 0.10 ± 0.05 | 0.03 ± 0.02 |
> > > > > > | LPP-EOpp    | 0.67 ± 0.01 | 0.10 ± 0.04 | 🟩 0.09 ± 0.04 | 0.04 ± 0.03 | 🟥 0.14 ± 0.03 | 🟩 0.08 ± 0.03 | 0.00 ± 0.00 |
> > > > > >
> > > > > > **6. Intervention optimality.**
> > > > > > 	Thank you for continuing to raise this concern. Empirically, we found that both methods are comparable across four standard, diverse datasets
> > > > > > and, in theory, the AntiDiagonal and LabelFlipping constructions of the randomized classifiers are identical on a population level via a re-parameterization. In particular, the following proposition holds:
> > > > > >
> > > > > > **Lemma (Explicit mapping AD $\leftrightarrow$ LF)**
> > > > > > Let $\lambda\in[0,1]$ and $p\in[0,1]$. Define
> > > > > > $p_0 := \lambda p, p_1 := \lambda(1-p).$
> > > > > > Then the _AntiDiagonal_ classifier $f_{\mathrm{AD}}$ with parameters $(\lambda,p)$ has the same conditional distribution given $f^{(0)}(X)$ as the _LabelFlipping classifier_ $f_{\mathrm{LF}}$ with parameters $(p_0,p_1)$.
> > > > > >
> > > > > > Conversely, given any $p_0,p_1\in[0,1]$ with $p_0+p_1\le1$, define
> > > > > > $\lambda := p_0 + p_1,\qquad
> > > > > > p :=
> > > > > > \begin{cases}
> > > > > > \dfrac{p_0}{p_0+p_1}, & p_0+p_1>0; \qquad
> > > > > > \text{arbitrary in }[0,1], & p_0+p_1=0,
> > > > > > \end{cases}$
> > > > > > then LabelFlipping with $(p_0,p_1)$ coincides in distribution with AntiDiagonal with $(\lambda,p)$.
> > > > > >
> > > > > > The proof is immediate from elementary algebra  using conditional probabilities and the definitions of our AntiDiagonal and LabelFlipping randomization procedures.
> > > > > >
> > > > > > In light of this discussion, we have decided to remove the AntiDiagonal method entirely, choosing to keep the LabelFlipping modality since this randomization scheme has been previously studied, though in limited settings, in prior post-processing fairness literature (see, e.g., Hardt et al., 2016; Hsu et al., 2022 and Remark 4.3 in our main text).

---

> > > > > > > ### Author Response · Authors · 2025-12-02
> > > > > > >
> > > > > > > **(Additional) References**
> > > > > > >
> > > > > > > Alekh Agarwal, Alina Beygelzimer, Miroslav Dud´ık, John Langford, and Hanna Wallach. A reductions approach to fair classification. In International conference on machine learning, pp. 60–69. PMLR, 2018
> > > > > > >
> > > > > > > Brian Hsu, Rahul Mazumder, Preetam Nandy, and Kinjal Basu. Pushing the limits of fairness impossibility: Who’s the fairest of them all? Advances in Neural Information Processing Systems, 35:32749–32761, 2022.
> > > > > > >
> > > > > > > Jacob Devlin, Ming-Wei Chang, Kenton Lee, and Kristina Toutanova. Bert: Pre-training of deep bidirectional transformers for language understanding. In Proceedings of NAACL-HLT, 2019
> > > > > > >
> > > > > > > Julia Angwin, Jeff Larson, Surya Mattu, and Lauren Kirchner. Machine Bias. ProPublica, 2016. URL https://www.propublica.org/article/machine-bias-risk-assessments-in-criminal-sentencing.
> > > > > > >
> > > > > > > Lara Groves, Jacob Metcalf, Alayna Kennedy, Briana Vecchione, and Andrew Strait. Auditing
> > > > > > > work: exploring the new york city algorithmic bias audit regime. In Proceedings of the 2024
> > > > > > > ACM Conference on Fairness, Accountability, and Transparency, pp. 1107–1120, 2024.
> > > > > > >
> > > > > > > Maria De-Arteaga, Alexey Romanov, Hanna Wallach, Jennifer Chayes, Christian Borgs, Alexandra Chouldechova, Sahin Geyik, Krishnaram Kenthapadi, and Adam Tauman Kalai. Bias in bios: A case study of semantic representation bias in a high-stakes setting. In proceedings of the Conference on Fairness, Accountability, and Transparency, pp. 120–128, 2019.
> > > > > > >
> > > > > > > Moritz Hardt, Eric Price, and Nati Srebro. Equality of opportunity in supervised learning. Advances in neural information processing systems, 29, 2016.
> > > > > > >
> > > > > > > Piet Groeneboom. Grenander functionals and cauchy’s formula. Scandinavian Journal of Statistics,
> > > > > > > 48(1):275–294, 2021.
> > > > > > >
> > > > > > > Ron Kohavi. Scaling Up the Accuracy of Naive-Bayes Classifiers: A Decision-Tree Hybrid. In Proceedings of the Second International Conference on Knowledge Discovery and Data Mining, 1996.
> > > > > > >
> > > > > > > Ruicheng Xian and Han Zhao. A unified post-processing framework for group fairness in classification. arXiv preprint arXiv:2405.04025, 2024.
> > > > > > >
> > > > > > > Sarah Bird, Miro Dud´ık, Richard Edgar, Brandon Horn, Roman Lutz, Vanessa Milan, Mehrnoosh Sameki, Hanna Wallach, and Kathleen Walker. Fairlearn: A toolkit for assessing and improving fairness in ai. 2020.
> > > > > > >
> > > > > > > Shauli Ravfogel, Yanai Elazar, Hila Gonen, Michael Twiton, and Yoav Goldberg. Null it out: Guard-
> > > > > > > ing protected attributes by iterative nullspace projection. arXiv preprint arXiv:2004.07667, 2020.
> > > > > > >
> > > > > > > Tim Dettmers, Artidoro Pagnoni, Ari Holtzman, and Luke Zettlemoyer. Qlora: Efficient finetuning of quantized llms. Advances in neural information processing systems, 36:10088–10115, 2023.

---

### Official Review · Reviewer_nf54 · 2025-10-28

**Soundness:** 4
**Presentation:** 4
**Contribution:** 2
**Rating:** 4
**Confidence:** 4

**Summary:**

This paper proposes post-processing algorithms for achieving (multiple) group fairness (criteria simultaneously), covering criteria that can be expressed in linear or linear-fractional forms: this covers SP, EO, TPR/FPR parity, and predictive rate parity.  The focus is on binary classification and the attribute-aware setting (sensitive attribute can be used explicitly as input to the classifier/post-processor).

Given the ROC (i.e., feasible TPR-FPR tradeoffs) of the Bayes optimal score function:

1. With a reformulation trick, the authors show that, if the "centroid" of the group statistics of the fair classifier is known, then the fairness criteria can be expressed as linear constraints in terms of the group-conditional TPR and FPR.
2. If the performance of the classifier is a function of the TPR and FPR (e.g., accuracy), then the optimal fair classifier is fully characterized by its TPR-FPR tradeoff.

With these two facts,

1. The authors proposed a bi-level optimization procedure where the outer layer sweeps over the possible centroids, and the inner layer solves the fair classification with the linear constraints as a linear program. Finally, it returns the one with the centroids that leads to the best performance.
2. To return a classifier achieving a specific TPR-FPR tradeoff, two methods are proposed: one is based on randomly interpolating a thresholded classifier (which lies on the boundary of the ROC) and a input-independent randomized classifier (which lies on the diagonal), and another which is based on combining thresholding and label flipping.

**Strengths:**

- To the reviewer's knowledge, this is the first paper proposing a post-processing based method for achieving fair classification on linear (SP, EO, TPR/FPR) and linear-fractional (predictive rate equality) fairness constraints simultaneously.
- Experiment results show good empirical performance.
- The reviewer finds the reformulation trick for "linearizing" the linear-fractional constraints interesting, and could be also be applied to the more general attribute-blind and multi-class setting (for representing the optimal fair classifier).

**Weaknesses:**

- The fact that a bi-level optimization procedure involving a grid search for finding the optimal "centroid" is required is a bit dissatisfying, and suggests that the proposed algorithm may be hard to scale to more general settings.
- The lack of a sample complexity analysis (probably about the same as fitting a linear model with $O(K)$ parameters), and an error propagation analysis: if the upstream score function is non-Bayes-optimal, how much would it hurt performance and fairness?

**Questions:**

N/A

---

> ### Author Response · Authors · 2025-11-22
>
> **C1: Scalability concerns.**
>
> Comment A2 from Reviewer YVAP raised similar concerns about the scalability of our method, and we copy and paste our response here for completeness and convenience.
>
> For convenience and ease of discussion, we have copy-pasted the experimental results of §5.1.2 most relevant to scalability concerns here, numbered as Tables 3, 4 and 5 in our author reply.
>
> _Inner subroutine_: The inner LP subroutine of the grid search does depend on the number of groups, but
> only requires solving a modestly sized LP;
> as discussed in Remark A.2., each iteration of the outer grid search solves an LP with
> $\sum_{a\in\mathcal{A}} |S_a| + |\mathcal{K}\_{\text{L}}|$ decision variables,
> and
> $|\mathcal{A}|+1$ linear constraints
> (for the simplex and post-processing per group),
> $2|\mathcal{A}||\mathcal{K}\_{\text{L}}|$ linear constraints (for the linear centroid bands),
> and $2|\mathcal{A}||\mathcal{K}\_{\text{LF}}|$ linear constraints
> (for the LF bands).
> Here, $|S_a|$ is the number of points in the support of the convex hull of the ROC curve for group $a\in\mathcal{A}$ and is typically small
> both theoretically (Groeneboom et al., 2021),
> and empirically in our experiments, where it usually ranges between values such as 20 to 30.
> Moreover,
>  $|\mathcal{K}\_{\text{LF}}|$ and $|\mathcal{K}\_{\text{L}}|$ are the number of linear-fractional (up to 2 in our experiments)
>  and linear group fairness constraints (up to 3 in our experiments), respectively.
>
>  Thus, for instance, if we have five protected attributes (like in the largest example that we consider), the total number of decision variables is roughly 100-150, and the total number of constraints is roughly 20.
>  This leads to modest-sized linear programs that current computational infrastructure can easily handle.
>
> _Outer search:_ The outer grid search over the centroids of the linear-fractional constraints does not depend on the number of groups $|\mathcal{A}|$,
> since we represent all linear-fractional and linear $\binom{|\mathcal{A}|}{2}$ pairwise disparity constraints as $|\mathcal{A}|$ linear disparity constraints via centroid-based linearization (see, e.g., §4.2).
>
> From a worst-case perspective,
> the outer grid search
> can depend
> exponentially in the number of linear-fractional constraints;
>  however, this is not a practical limitation.
>  Indeed, first of all, the number of linear fractional constraints is always small, at most two, in our experiments.
>  Moreover,
> in practice we found  that using a relatively coarse grid of $25, 50$ or $100$ points works well when controlling for two linear-fractional constraints.
>  This means that we need to search over at most 10,000 grid points,
>  which
>  is very feasible and
>  takes a total of at most  4-5 minutes in our experiments and computational infrastructure.
> As suggested by comment B3 of Reviewer v8n8, and which is presently relevant here, we have added a new section with a
> sensitivity analysis for the granularity of the grid search in §A.6.3,
>  where we show that in our experiments, even a smaller grid size of 50 or 25 can work quite well, and which does further reduces the total number of required grid point evaluations to up to 1250.
>
> _Running times_: Though our methods exhibit considerably larger running times for the setting of multiple linear-fractional constraints and multiple protected attributes (Table 5)
> we would like to note that this is due in part to the adaptive nature of our feasibility guard --
> for this experiment, the guard is triggered approximately $p=2/3$ of the time (Table 4, in blue), which results in a bisection search on top of the outer-grid search.
>
> The benefit is that our procedure finds a fair classifier with a minimal relaxation of the user-specified disparity tolerances, when the nominal disparity levels are ill-specified.
>
> When the feasibility guard is not activated, we see that the outer grid search is modestly timed, coming in at around 4-6 minutes per seed (Tables 4 and 5).
>
> _Summary:_ Ultimately, our methods are able to scale in performance and speed to
> large datasets
> such as ACSIncome
> (which have hundreds of thousands of data points and
> almost a dozen features)
> for the challenging setting of controlling multiple linear-fractional and linear group fairness constraints (Table 3) -- a setting on which existing post-processing methods (see, e.g., Celis et al., 2019; Hsu et al., 2022) have been unable to demonstrate similarly strong empirical performance.

---

> > ### Author Response · Authors · 2025-11-22
> >
> > **C1: Scalability Concerns (ctd.).**
> >
> > **Table 3**: Performance on the test set for ACSIncome ($|\mathcal{A}|{=}5$).
> > The disparities $\delta_{\textrm{EOpp}}, \delta_{\textrm{PP}}, \delta_{\textrm{FOR}}$ are controlled at level $0.10$ whenever they are active. Cells in green indicate that the fairness constraint is satisfied within two standard deviations at level $0.10$, whereas cells in red indicate violation.
> > | Method      | Acc         | DP          | EOpp           | PEq         | PP             | FOR            | Interv.     |
> > | ----------- | ----------- | ----------- | -------------- | ----------- | -------------- | -------------- | ----------- |
> > | Baseline    | 0.78 ± 0.01 | 0.24 ± 0.01 | 🟥 0.25 ± 0.03 | 0.09 ± 0.02 | 🟥 0.23 ± 0.03 | 🟩 0.08 ± 0.01 | 0.00 ± 0.00 |
> > | Oracle      | 0.57 ± 0.04 | 0.48 ± 0.05 | 🟩 0.10 ± 0.01 | 0.57 ± 0.08 | 🟩 0.10 ± 0.01 | 🟩 0.10 ± 0.01 | N/A         |
> > | **ROCF-AD** (ours) | 0.57 ± 0.02 | 0.48 ± 0.03 | 🟩 0.11 ± 0.01 | 0.57 ± 0.05 | 🟩 0.11 ± 0.01 | 🟩 0.11 ± 0.03 | 0.06 ± 0.01 |
> > | **ROCF-LF** (ours) | 0.56 ± 0.02 | 0.49 ± 0.02 | 🟩 0.11 ± 0.01 | 0.59 ± 0.04 | 🟩 0.11 ± 0.01 | 🟩 0.11 ± 0.03 | 0.05 ± 0.01 |
> > | LPP-EOpp    | 0.79 ± 0.00 | 0.15 ± 0.01 | 🟩 0.11 ± 0.01 | 0.06 ± 0.01 | 🟥 0.31 ± 0.01 | 🟩 0.12 ± 0.01 | 0.00 ± 0.00 |
> >
> > **Table 4**: Summary statistics for feasibility guard triggers for the nominal disparity levels; $p$ is the proportion of runs and $\mu$ is the average expansion factor with listed s.d.
> > | Setting    | $ \delta $ = (DP, EOpp, PEq, PP, FOR) | COMPAS               | ACSIncome ($\| \mathcal{A} \| =2$)      | ACSIncome ($\| \mathcal{A} \|=5 $)         |
> > | ---------- | -------------------------------- | -------------------- | -------------------- | ----------------------- |
> > | ROCF-AD/LF | (0.05, 0.05, 0.05, 0.05, None)   | p=0.00 ; μ=1.00±0.00 | p=0.00 ; μ=1.00±0.00 | p=0.00 ; μ=1.00±0.00    |
> > | ROCF-AD/LF | (None, 0.1, None, 0.1, 0.1)      | p=0.10 ; μ=1.01±0.02 | p=0.00 ; μ=1.00±0.00 | 🟦 p=0.66 ; μ=1.04±0.05 |
> > | Oracle     | (0.05, 0.05, 0.05, 0.05, None)   | p=0.00 ; μ=1.00±0.00 | p=0.00 ; μ=1.00±0.00 | p=0.00 ; μ=1.00±0.00    |
> > | Oracle     | (None, 0.1, None, 0.1, 0.1)      | p=0.08 ; μ=1.00±0.02 | p=0.00 ; μ=1.00±0.00 | 🟦 p=0.64 ; μ=1.05±0.08 |
> >
> > **Table 5**: CPU runtime (mean $\pm$ s.d., seconds) for COMPAS, ACSIncome ($|\mathcal{A}|{=}2$), and ACSIncome ($|\mathcal{A}|{=}5$). The disparities $\delta_{\textrm{EOpp}},\delta_{\textrm{PP}}$ and $\delta_{\textrm{FOR-parity}}$ are controlled at level $0.10$ for our methods ROCF-AD/LF, while LPP-EOpp only controls for equality of opportunity.
> > | Method   | COMPAS (sec.)      | ACSIncome ($\| \mathcal{A} \| =2$) (sec.) | ACSIncome ($\| \mathcal{A} \| =5$) (sec.) |
> > | -------- | --------------- | --------------------- | --------------------- |
> > | ROCF-AD  | 377.81 ± 312.37 | 272.50 ± 3.52         | 3903.40 ± 1550.97     |
> > | ROCF-LF  | 385.55 ± 314.28 | 273.22 ± 3.96         | 3675.52 ± 1591.78     |
> > | LPP-EOpp | 0.06 ± 0.02     | 17.85 ± 1.16          | 18.52 ± 0.58          |

---

> > > ### Author Response · Authors · 2025-11-22
> > >
> > > **C2: Sample complexity and error propagation analysis.**
> > > Thank you for this suggestion. We have added a new theoretical result, Theorem 4.2 in §4.2,
> > > which establishes sample-complexity guarantees for both risk generalization and fairness attainment of our procedure’s operating-characteristic solution, conditional on a _fixed_ score function --
> > > in the special case where this score function is Bayes-optimal, the theorem yields a parametric $\tilde O(1/\sqrt{n})$ convergence rate,
> > > showing that our method can attain statistically optimal performance and fairness in the idealized limit.
> > >
> > > For convenience, we reproduce the theorem statement below; the complete proof appears in §A.4.2 of the revised manuscript.
> > >
> > > **Theorem:**
> > >     Let $s:\mathcal{X}\times\mathcal{A}\to[0,1]$ be a fixed probabilistic predictor
> > >     and let $\mathcal{D}\_{\textnormal{post}}$ be an i.i.d. post-processing sample with
> > >     group and group–label specific sample sizes $n_a$ and $n_{a,y}$, respectively,
> > >     with $n_{\min} := \min\_{a,y} n\_{a,y}$ for $a\in\mathcal{A}$ and $y\in\{0,1\}$.
> > >     Also, let
> > >     $\widehat\varrho = \\{\widehat{\vec \rho_a}\\}\_{a\in\mathcal{A}}$ be the lifted operating characteristics returned by our empirical region search procedure
> > >     (i.e., Algorithm 2 in §A.2.1) with grid steps $h_k$ for each linear-fractional centroid and with
> > >     $\varepsilon_k$ as the denominator guards.
> > > For $\delta>0$, define for $j \in \{0,1\}$,
> > > \begin{equation}
> > >         \eta_{a,j}=\sqrt{\frac{1}{2n_{a,j}}\log\frac{4m}{\delta}} + C\cdot
> > >         \sqrt{\frac{1}{2n_{a}}\log\frac{4mK}{\delta}},
> > > \end{equation}
> > > where $C>0$ is a universal constant that depends only on $\{u\_{k,a},v\_{k,a}\}\_{a\in\mathcal{A}, k\in\mathcal{K}}$,
> > > but not on $n_{a,j},m,K$, or $\delta$.
> > >
> > > Then, with probability at least $1-\delta$,
> > >
> > > $\Big|J(\widehat\varrho)-J(\varrho^\star)\Big|
> > >     \lesssim
> > >     \sum\_{a\in\mathcal A}\big(\eta_{a,1}+\eta_{a,0}\big)
> > >     + \sum\_{k\in K_{\mathrm{LF}}} h_k
> > >     + \max\_{k\in K_{\mathrm{LF}}}\varepsilon_k,$
> > >
> > > and for every fairness metric $k\in K\_{\mathrm{LF}}\cup K\_{\mathrm L}$,
> > >
> > > $ \max\_{a,a'}
> > >     \big|
> > >     G\_{k,a}(\widehat\rho_a)-G\_{k,a'}(\widehat\rho_{a'})
> > >     \big|
> > >     \lesssim
> > >     \delta_k +
> > >     \max_{a}(\eta_{a,1}+\eta_{a,0}).$
> > >
> > > Developing error propagation bounds with the Bayes-optimal regression function
> > > is non-trivial for our approach,
> > > since we work directly on the empirical ROC curve:
> > > establishing such results would require characterizing this ROC curve's behavior both along its trajectory as we sweep the threshold $t$ in $1(s(x,a)>t)$ and within the model space (e.g., $s(x,a)$ are 3-layer hidden neural networks).
> > > This control of the ROC curve geometry is interesting, though it would seem to require novel proof ideas and tools not standard to common empirical process theory, so we leave these results to future work.
> > >
> > > More generally speaking,
> > > sample complexity and error propagation results exist in the broader algorithmic fairness literature (see e.g., Kleinberg, 2018; Celis et al., 2019; Xian & Zhao, 2024)
> > >  with the work of Celis et al., 2019 being the most relevant to our approach.
> > > However, we would like to argue that such results are ultimately unable to
> > > bridge the gap between theory and practice, as such methods exhibit subpar empirical results unlike our approach (see empirical analyses in §5).
> > > With this newly added theoretical result, our approach is both empirically and theoretically well motivated, and
> > > we leave it to future work to develop similarly rigorous results for error propagation bounds.
> > >
> > > **References:**
> > >
> > > Celis, L. E., Huang, L., Keswani, V., & Vishnoi, N. K. (2019).
> > > Classification with fairness constraints: A meta-algorithm with provable guarantees.
> > > FAT* Conference, 319–328.
> > >
> > > Hsu, B., Mazumder, R., Nandy, P., & Basu, K. (2022).
> > > Pushing the limits of fairness impossibility: Who’s the fairest of them all?
> > > Advances in Neural Information Processing Systems, 35, 32749–32761.
> > >
> > > Kleinberg, J. (2018).
> > > Inherent trade-offs in algorithmic fairness.
> > > Proceedings of the 2018 ACM SIGMETRICS Conference, 40.
> > >
> > > Xian, R., & Zhao, H. (2024).
> > > A unified post-processing framework for group fairness in classification.
> > > arXiv:2405.04025.

---

> ### Author Response · Authors · 2025-11-28
>
> Dear Reviewer,
>
> We hope you are well. We would like to follow up on this thread. We have posted a detailed response, aiming to answer each of your concerns directly in a point-by-point manner. We hope that this has addressed your concerns. If you have any additional questions or concerns, please let us know. Otherwise, if all of your concerns have been addressed, and therefore you believe that the paper can be published, we would be grateful if you could reconsider your assessment and reflect this in your score.
>
> Thank you again for your careful reading of the paper and valuable suggestions.
> Sincerely,
> The authors.

---

### Official Review · Reviewer_v8n8 · 2025-11-01

**Soundness:** 4
**Presentation:** 3
**Contribution:** 4
**Rating:** 6
**Confidence:** 2

**Summary:**

This paper introduces a method for jointly optimizing multiple fairness constraints under approximation thresholds. The method leverages the ROC to satisfy linear fractional fairness constraints, and provides extensive analytical results on the framework.

The authors empirically evaluate on standard fairness datasets, and demonstrate near-oracle performance over multiple fairness measures, with a principled and flexible delta-approximate fairness.

**Strengths:**

1. The model is well-motivated and the authors situate their work well in the introduction. The application of the ROC is intuitive to a novice reader at a conceptual level.

2. The method evaluates quite well. The method consistently achieves near-oracle performance over multiple fairness measures. The authors evaluate over standard fairness-related datasets which is fine

3. The method allows an intuitive approximate fairness threshold delta, which allows adaptation to the application. Furthermore, the adaptive feasibility search allows this parameter to be over-constrained while still providing a solution in a principled manner .

**Weaknesses:**

1. Overall, the paper is challenging for a general AI researcher reader, and likely challenging for a fair AI researcher. I don't have a great recommendation for improving this, as the analytical results are necessarily technical.

2. I have low-confidence in verifying the proofs in the paper, particularly Thm 4.1 in A.4.1. Therefore I defer to other reviewers on correctness and evaluate as if true.
- I believe the proof requires a Q_k construction, which is deferred to A.5.1. This should be moved up in the proof context

3. The method is sensitive to the cardinality of the region search grid. Finding the appropriate scale on this approximation seems to be challenging. The authors address this in terms of runtime, but does an overly-granular grid produces issues of over-sparsity?

**Questions:**

1. Does this method suffer from sparse values in the grid search? i.e. is the only trade-off runtime, or does the result fail on overly-granular grids?

---

> ### Author Response · Authors · 2025-11-20
>
> **B1: Dense exposition of methodology and results.**
> We have added a fully self-contained algorithm detailing our end-to-end pipeline for the simpler setting of
> controlling for a single linear-fractional and linear group fairness constraint in §A.3.6.
> We hope this new algorithm
> helps clarify our method for both the interested practitioner and the general AI researcher.
>
> If the reviewer has any particular recommendations that can help clarify the paper's exposition, we would be happy to consider them.
>
> **B2: Misplacement of centroid construction.**
> We have moved the general construction of the admissible centroids from its previous location in §A.5.1 to just before the proof of Thm 4.1 in §A.4.
> As before, we defer the theoretical bands and practical implementation details for common linear-fractional constraints to §A.5.1.
>
> **B3: Sensitivity analysis with granularity of grid search.**
> We have added a new section with a
> sensitivity analysis for the granularity of the grid search over the centroid of the linear-fractional constraints
> in §A.6.3 of the updated manuscript. For convenience, we present the same results and a similar discussion here.
>
> We perform a sensitivity analysis of the effect of the grid's granularity on our method's
> accuracy performance \& fairness attainment (Table 1),
> computational runtimes (Table 2),
> and feasibility guard triggers (Table 3)
> for the ACSIncome experiment with multiple protected attributes and linear-fractional constraints.
>
> The grid size of either $10, 25, 50$, or $100$, refers to the number of the equidistantly spaced centroids in both the predictive parity and false omission rate parity bands
> (e.g., for two linear-fractional constraints, a grid size of $q$ would result in $q\times q$ pairs in total for the grid search).
> In the original experiment with multiple linear-fractional constraints (cf. §5.1.2), we use $q=100$ points.
>
> _Discussion_. We observe that our method's performance is quite poor when we have an overly sparse grid of 10 points: this requires considerable relaxation of the nominal disparity levels and, consequently,
> does not attain the desired fairness levels even within 2 s.d.s. (see Table 1(A) and first row of Table 3).
>
> In contrast, having a moderately coarse grid works well -- even with 25 and 50 grid points,
> our methods achieve similar accuracy performance and fairness attainment for 100 grid points, while exhibiting considerably shorter computational runtimes (see Tables 1(B)-(D) and Table 2).
>
> This demonstrates that, though our methods are susceptible to overly granular grids (e.g., $q=10$), a moderately spaced grid performs well in practice and is comparable with the denser grids considered in §5.1.2.
>
> **Table 1(A):** Performance on the test set for ACSIncome ($|\mathcal{A}|=5$) with $q=10$ grid points.
> The disparities $\delta_{\textrm{EOpp}}, \delta_{\textrm{PP}}$ and $\delta_{\textrm{FOR}}$ are controlled at level $0.10$ whenever they are active.
> Cells in green indicate that the fairness constraint is satisfied within two standard deviations at level $0.10$, whereas cells in red indicate violation.
>
> | Method      | Acc         | DP          | EOpp           | PEq         | PP             | FOR            | Interv.     |
> | ----------- | ----------- | ----------- | -------------- | ----------- | -------------- | -------------- | ----------- |
> | Baseline    | 0.79 ± 0.00 | 0.25 ± 0.01 | 🟥 0.24 ± 0.02 | 0.09 ± 0.02 | 🟥 0.23 ± 0.02 | 🟩 0.08 ± 0.01 | N/A         |
> | Oracle      | 0.66 ± 0.04 | 0.41 ± 0.04 | 🟥 0.12 ± 0.01 | 0.41 ± 0.07 | 🟥 0.12 ± 0.01 | 🟩 0.07 ± 0.03 | N/A         |
> | **ROCF-AD** (ours) | 0.65 ± 0.02 | 0.42 ± 0.03 | 🟥 0.14 ± 0.01 | 0.44 ± 0.05 | 🟥 0.13 ± 0.01 | 🟩 0.07 ± 0.02 | 0.02 ± 0.02 |
> | **ROCF-LF** (ours) | 0.65 ± 0.02 | 0.42 ± 0.03 | 🟥 0.14 ± 0.01 | 0.44 ± 0.05 | 🟥 0.13 ± 0.01 | 🟩 0.07 ± 0.02 | 0.02 ± 0.02 |
>
> **Table 1(B):** Performance on the test set for ACSIncome ($|\mathcal{A}|=5$) with $q=25$ grid points.
> | Method      | Acc         | DP          | EOpp           | PEq         | PP             | FOR            | Interv.     |
> | ----------- | ----------- | ----------- | -------------- | ----------- | -------------- | -------------- | ----------- |
> | Baseline    | 0.79 ± 0.00 | 0.25 ± 0.01 | 🟥 0.24 ± 0.02 | 0.09 ± 0.02 | 🟥 0.23 ± 0.02 | 🟩 0.08 ± 0.01 | N/A         |
> | Oracle      | 0.58 ± 0.03 | 0.47 ± 0.03 | 🟥 0.11 ± 0.00 | 0.55 ± 0.06 | 🟩 0.11 ± 0.01 | 🟩 0.09 ± 0.01 | N/A         |
> | **ROCF-AD** (ours) | 0.58 ± 0.03 | 0.47 ± 0.03 | 🟥 0.12 ± 0.01 | 0.55 ± 0.05 | 🟩 0.11 ± 0.01 | 🟩 0.10 ± 0.02 | 0.05 ± 0.01 |
> | **ROCF-LF** (ours) | 0.58 ± 0.03 | 0.47 ± 0.03 | 🟥 0.12 ± 0.01 | 0.55 ± 0.05 | 🟩 0.11 ± 0.01 | 🟩 0.10 ± 0.02 | 0.05 ± 0.01 |

---

> > ### Author Response · Authors · 2025-11-20
> >
> > **B3: Sensitivity analysis with granularity of grid search (ctd.).**
> >
> > **Table 1(C):** Performance on the test set for ACSIncome ($|\mathcal{A}|=5$) with $q=50$ grid points.
> >
> > | Method      | Acc         | DP          | EOpp           | PEq         | PP             | FOR            | Interv.     |
> > | ----------- | ----------- | ----------- | -------------- | ----------- | -------------- | -------------- | ----------- |
> > | Baseline    | 0.79 ± 0.00 | 0.25 ± 0.01 | 🟥 0.24 ± 0.02 | 0.09 ± 0.02 | 🟥 0.23 ± 0.02 | 🟩 0.08 ± 0.01 | N/A         |
> > | Oracle      | 0.57 ± 0.03 | 0.48 ± 0.03 | 🟩 0.10 ± 0.00 | 0.57 ± 0.05 | 🟩 0.10 ± 0.01 | 🟩 0.10 ± 0.01 | N/A         |
> > | **ROCF-AD** (ours) | 0.56 ± 0.02 | 0.48 ± 0.03 | 🟩 0.11 ± 0.01 | 0.57 ± 0.04 | 🟩 0.11 ± 0.01 | 🟩 0.10 ± 0.01 | 0.05 ± 0.01 |
> > | **ROCF-LF** (ours) | 0.56 ± 0.02 | 0.48 ± 0.03 | 🟩 0.11 ± 0.01 | 0.57 ± 0.04 | 🟩 0.11 ± 0.01 | 🟩 0.12 ± 0.09 | 0.05 ± 0.01 |
> >
> > **Table 1(D):** Performance on the test set for ACSIncome ($|\mathcal{A}|=5$) with $q=100$ grid points.
> >
> > | Method      | Acc         | DP          | EOpp           | PEq         | PP             | FOR            | Interv.     |
> > | ----------- | ----------- | ----------- | -------------- | ----------- | -------------- | -------------- | ----------- |
> > | Baseline    | 0.79 ± 0.00 | 0.25 ± 0.01 | 🟥 0.24 ± 0.02 | 0.09 ± 0.02 | 🟥 0.23 ± 0.02 | 🟩 0.08 ± 0.01 | N/A         |
> > | Oracle      | 0.57 ± 0.04 | 0.48 ± 0.05 | 🟩 0.11 ± 0.01 | 0.56 ± 0.08 | 🟩 0.10 ± 0.01 | 🟩 0.10 ± 0.01 | N/A         |
> > | **ROCF-AD** (ours) | 0.57 ± 0.02 | 0.48 ± 0.03 | 🟩 0.11 ± 0.01 | 0.57 ± 0.05 | 🟩 0.11 ± 0.01 | 🟩 0.11 ± 0.03 | 0.06 ± 0.01 |
> > | **ROCF-LF** (ours) | 0.56 ± 0.02 | 0.49 ± 0.02 | 🟩 0.11 ± 0.01 | 0.59 ± 0.04 | 🟩 0.11 ± 0.01 | 🟩 0.11 ± 0.03 | 0.05 ± 0.01 |
> >
> > **Table 2:** CPU runtime (mean $\pm$ s.d., seconds) for ACSIncome ($|\mathcal{A}|{=}5$) across various grid sizes. The grid size refers to the number of equidistantly spaced points searched over in the outer grid search for linear-fractional constraints.
> > | Grid Size | AD Runtime (s)    | LF Runtime (s)    |
> > | --------- | ----------------- | ----------------- |
> > | 10        | 48.26 ± 1.83      | 52.48 ± 2.75      |
> > | 25        | 255.41 ± 49.35    | 238.28 ± 45.50    |
> > | 50        | 789.46 ± 294.07   | 764.11 ± 281.46   |
> > | 100       | 3903.40 ± 1550.97 | 3675.52 ± 1591.78 |
> >
> > **Table 3:** Summary statistics for feasibility guard triggers for the nominal disparity levels considered across various grid sizes;
> > $p$ is the proportion of runs and $\mu$ is the average expansion factor with listed s.d.
> >
> > | Grid Size | ROCF-AD p | ROCF-AD μ   | ROCF-LF p | ROCF-LF μ   |
> > | --------- | --------- | ----------- | --------- | ----------- |
> > | 10        | 1.00      | 1.23 ± 0.01 | 1.00      | 1.23 ± 0.01 |
> > | 25        | 0.88      | 1.08 ± 0.01 | 0.88      | 1.08 ± 0.01 |
> > | 50        | 0.72      | 1.04 ± 0.01 | 0.72      | 1.04 ± 0.01 |
> > | 100       | 0.64      | 1.05 ± 0.01 | 0.64      | 1.04 ± 0.01 |

---

### Official Review · Reviewer_YVAP · 2025-11-03

**Soundness:** 3
**Presentation:** 3
**Contribution:** 3
**Rating:** 4
**Confidence:** 3

**Summary:**

This paper presents ROCF (Fair Classification via Operating Characteristic Feasibility Regions), a post-processing framework for achieving approximate group fairness under multiple constraints such as demographic parity (DP), equalized odds (EO), and predictive parity (PP).
Instead of retraining models, the method intervenes directly on the operating characteristics (TPR/FPR) of a pre-trained classifier. It identifies feasible fairness-adjusted points using each group’s convex hull and post-processes predictions to reach those points. The approach extends to multiple protected attributes and supports linear-fractional fairness metrics.
Experiments on COMPAS and ACSIncome show that ROCF achieves approximate fairness across several criteria with minimal accuracy loss and few label interventions, outperforming prior post-processing baselines (Celis et al., 2019; Hsu et al., 2022).

**Strengths:**

Novel geometric perspective: Optimizing fairness directly in ROC space via convex hulls is elegant and intuitive.
Handles multiple fairness criteria: Supports several linear and fractional constraints simultaneously, a limitation of most existing post-processing methods.
Practical deployment path: Works on top of pre-trained models, making it applicable when retraining is infeasible.

**Weaknesses:**

Limited dataset diversity: Evaluation on only two benchmark datasets limits generalization; larger and more complex domains should be tested.
Scalability concerns: The grid search may be expensive when handling many groups or fairness constraints.
Sensitivity to data size: Since ROCF relies on convex hull estimation, small or noisy samples may affect feasibility, but this is not quantified.
Post-processing limitations: As with all post-processing methods, bias in the learned representation remains unaddressed.

**Questions:**

Complexity and scalability: How does the computational cost of the grid search scale with the number of fairness constraints and protected groups?
Generalization beyond binary tasks: Could the ROC-based formulation extend to multi-class or regression settings, or is it inherently tied to binary classification?

---

> ### Author Response · Authors · 2025-11-20
>
> **A1: Limited dataset diversity.** We have added new experimental results for another dataset in §5 of the revised manuscript.
> This school-admissions dataset, Lawschool (Wightman 1998; Fabris et al. 2022),
> seeks to predict whether an individual will be admitted to law school, and is a moderately sized dataset containing approximately $96{,}500$ individuals and six covariates including LSAT score, GPA, gender, etc.
>
> Similar to our existing experimental setup,
> we are able to simultaneously control (i) demographic parity, equalized odds, and predictive parity at the $\delta=0.02$ level, and (ii) equality of opportunity, predictive parity, and false-omission rate parity at the $\delta=0.03$ level.
> We have chosen these levels to showcase that our method can have advantages compared to existing methods.
> If we choose higher levels of $\delta$, then all methods can satisfy the constraints,
> whereas if we choose smaller values, then the problem becomes too difficult for any method.
> We have displayed the new results of the various methods' performance on Lawschool
> here in Tables 1and 2 below for convenience.
>
> **Table 1**: Performance on the test set for Lawschool  ($|\mathcal{A}|{=}2$).
> The disparities $\delta_{\textrm{DP}}, \delta_{\textrm{EOpp}}, \delta_{\textrm{PEq}}, \delta_{\textrm{PP}}$ are controlled at level $0.02$ whenever they are active. Cells in green indicate that the fairness constraint is satisfied within two standard deviations at the nominal level, whereas cells in red indicate violation.
>
> | Method      | Acc         | DP             | EOpp           | PEq            | PP             | FOR         | Interv.     |
> | ----------- | ----------- | -------------- | -------------- | -------------- | -------------- | ----------- | ----------- |
> | Baseline    | 0.79 ± 0.00 | 🟥 0.07 ± 0.01 | 🟥 0.10 ± 0.02 | 🟩 0.03 ± 0.01 | 🟩 0.04 ± 0.02 | 0.04 ± 0.00 | 0.00 ± 0.00 |
> | Oracle      | 0.78 ± 0.00 | 🟩 0.02 ± 0.00 | 🟩 0.01 ± 0.01 | 🟩 0.01 ± 0.00 | 🟩 0.02 ± 0.00 | 0.06 ± 0.00 | N/A         |
> | **ROCF-AD** (ours) | 0.77 ± 0.00 | 🟩 0.02 ± 0.01 | 🟩 0.01 ± 0.01 | 🟩 0.01 ± 0.00 | 🟩 0.03 ± 0.03 | 0.06 ± 0.00 | 0.01 ± 0.00 |
> | **ROCF-LF** (ours) | 0.77 ± 0.00 | 🟩 0.02 ± 0.01 | 🟩 0.01 ± 0.01 | 🟩 0.01 ± 0.00 | 🟩 0.03 ± 0.02 | 0.06 ± 0.00 | 0.01 ± 0.00 |
> | MFOpt       | 0.67 ± 0.00 | 🟥 0.05 ± 0.01 | 🟩 0.03 ± 0.02 | 🟥 0.05 ± 0.01 | 🟩 0.04 ± 0.01 | 0.07 ± 0.01 | 0.16 ± 0.01 |
> | META        | 0.75 ± 0.01 | 🟩 0.02 ± 0.01 | 🟩 0.04 ± 0.03 | 🟩 0.03 ± 0.02 | 🟥 0.11 ± 0.01 | 0.05 ± 0.01 | 0.00 ± 0.00 |
> | LPP-DP      | 0.79 ± 0.00 | 🟩 0.02 ± 0.01 | 🟩 0.01 ± 0.01 | 🟩 0.01 ± 0.01 | 🟥 0.12 ± 0.01 | 0.05 ± 0.00 | 0.00 ± 0.00 |
> | LPP-EO      | 0.79 ± 0.00 | 🟩 0.03 ± 0.01 | 🟩 0.03 ± 0.01 | 🟩 0.01 ± 0.00 | 🟥 0.10 ± 0.01 | 0.05 ± 0.00 | 0.00 ± 0.00 |
>
> **Table 2**: Performance on the test set for Lawschool ($|\mathcal{A}|{=}2$). The same protocol as in Table 1 is used, except $\delta_{\textrm{EOpp}}, \delta_{\textrm{PP}}$ and $\delta_{\textrm{FOR}}$ are controlled at level $0.03$.
>
> | Method      | Acc         | DP          | EOpp           | PEq         | PP             | FOR            | Interv.     |
> | ----------- | ----------- | ----------- | -------------- | ----------- | -------------- | -------------- | ----------- |
> | Baseline    | 0.79 ± 0.00 | 0.07 ± 0.01 | 🟥 0.11 ± 0.02 | 0.03 ± 0.01 | 🟥 0.04 ± 0.02 | 🟥 0.04 ± 0.00 | 0.00 ± 0.00 |
> | Oracle      | 0.57 ± 0.03 | 0.16 ± 0.02 | 🟩 0.03 ± 0.00 | 0.16 ± 0.03 | 🟩 0.02 ± 0.01 | 🟩 0.03 ± 0.00 | N/A         |
> | **ROCF-AD** (ours) | 0.58 ± 0.04 | 0.15 ± 0.02 | 🟩 0.03 ± 0.01 | 0.16 ± 0.03 | 🟩 0.02 ± 0.01 | 🟩 0.03 ± 0.01 | 0.01 ± 0.01 |
> | **ROCF-LF** (ours) | 0.58 ± 0.04 | 0.15 ± 0.02 | 🟩 0.03 ± 0.01 | 0.16 ± 0.03 | 🟩 0.02 ± 0.01 | 🟩 0.03 ± 0.01 | 0.01 ± 0.01 |
> | LPP-EOpp    | 0.79 ± 0.00 | 0.04 ± 0.01 | 🟥 0.04 ± 0.02 | 0.01 ± 0.00 | 🟥 0.09 ± 0.01 | 🟥 0.05 ± 0.00 | 0.00 ± 0.00 |

---

> > ### Author Response · Authors · 2025-11-20
> >
> > **A1: Limited dataset diversity (ctd.).**
> >
> > We would like to note that the ACSIncome dataset is already a considerably large dataset with approximately
> > 1.6 million data points,
> > and has been proposed and used as a novel benchmark for the task of fair classification in algorithmic fairness
> > (Ding et al., 2021; Ezzeldin et al., 2023; Xian et al., 2024).
> >
> > Choosing datasets based on complexity is an interesting idea. The datasets that we have selected already demonstrate a significant level of complexity and depth, addressing important and complex social phenomena, such as social science problems (like income prediction) and educational questions (such as law school admissions). The data sets have a lot of depth to them with many features, which exhibit non-trivial relations and complexity, and a large number of observations. Moreover, these datasets are also widely used and recognized in the literature. This motivates our choice for these datasets.
> >
> > Moreover, on these complex datasets, we have demonstrated strong empirical results that to the best of our knowledge are the first of their kind: as illustrated in the empirical analysis of §5, our proposed methods _AntiDiagonal_ and _LabelFlipping_ achieve near-oracle level performance across all considered datasets, and,
> > as far as we know,
> > is the first post-processing method that can achieve simultaneous control of linear-fractional and linear group fairness constraints.
> >
> > With the added dataset during our rebuttal phase, we have now raised the number of datasets analyzed to three, which meets the standards for diversity, depth, and completeness of data analysis examples in papers in the area of algorithmic fairness published in
> > recent ML conferences.
> > However, if the reviewer has some specific suggestions for an additional data set to consider, we are happy to consider it within
> > reasonable limits.

---

> > > ### Author Response · Authors · 2025-11-20
> > >
> > > **A2: Scalability concerns.**
> > > For convenience and ease of discussion, we have copy-pasted the experimental results of §5.1.2 most relevant to scalability concerns here, numbered as Tables 3, 4 and 5 in our author reply.
> > >
> > > _Inner subroutine_: The inner LP subroutine of the grid search does depend on the number of groups, but
> > > only requires solving a modestly sized LP;
> > > as discussed in Remark A.2., each iteration of the outer grid search solves an LP with
> > > $\sum_{a\in\mathcal{A}} |S_a| + |\mathcal{K}\_{\text{L}}|$ decision variables,
> > > and
> > > $|\mathcal{A}|+1$ linear constraints
> > > (for the simplex and post-processing per group),
> > > $2|\mathcal{A}||\mathcal{K}\_{\text{L}}|$ linear constraints (for the linear centroid bands),
> > > and $2|\mathcal{A}||\mathcal{K}\_{\text{LF}}|$ linear constraints
> > > (for the LF bands).
> > > Here, $|S_a|$ is the number of points in the support of the convex hull of the ROC curve for group $a\in\mathcal{A}$ and is typically small
> > > both theoretically (Groeneboom et al., 2021),
> > > and empirically in our experiments, where it usually ranges between values such as 20 to 30.
> > > Moreover,
> > >  $|\mathcal{K}\_{\text{LF}}|$ and $|\mathcal{K}\_{\text{L}}|$ are the number of linear-fractional (up to 2 in our experiments)
> > >  and linear group fairness constraints (up to 3 in our experiments), respectively.
> > >
> > >  Thus, for instance, if we have five protected attributes (like in the largest example that we consider), the total number of decision variables is roughly 100-150, and the total number of constraints is roughly 20.
> > >  This leads to modest-sized linear programs that current computational infrastructure can easily handle.
> > >
> > > _Outer search:_ The outer grid search over the centroids of the linear-fractional constraints does not depend on the number of groups $|\mathcal{A}|$,
> > > since we represent all linear-fractional and linear $\binom{|\mathcal{A}|}{2}$ pairwise disparity constraints as $|\mathcal{A}|$ linear disparity constraints via centroid-based linearization (see, e.g., §4.2).
> > >
> > > From a worst-case perspective,
> > > the outer grid search
> > > can depend
> > > exponentially in the number of linear-fractional constraints;
> > >  however, this is not a practical limitation.
> > >  Indeed, first of all, the number of linear fractional constraints is always small, at most two, in our experiments.
> > >  Moreover,
> > > in practice we found  that using a relatively coarse grid of $25, 50$ or $100$ points works well when controlling for two linear-fractional constraints.
> > >  This means that we need to search over at most 10,000 grid points,
> > >  which
> > >  is very feasible and
> > >  takes a total of at most  4-5 minutes in our experiments and computational infrastructure.
> > > As suggested by comment B3 of Reviewer v8n8, and which is presently relevant here, we have added a new section with a
> > > sensitivity analysis for the granularity of the grid search in §A.6.3,
> > >  where we show that in our experiments, even a smaller grid size of 50 or 25 can work quite well, and which does further reduces the total number of required grid point evaluations to up to 1250.
> > >
> > > _Running times_: Though our methods exhibit considerably larger running times for the setting of multiple linear-fractional constraints and multiple protected attributes (Table 5)
> > > we would like to note that this is due in part to the adaptive nature of our feasibility guard --
> > > for this experiment, the guard is triggered approximately $p=2/3$ of the time (Table 4, in blue), which results in a bisection search on top of the outer-grid search.
> > >
> > > The benefit is that our procedure finds a fair classifier with a minimal relaxation of the user-specified disparity tolerances, when the nominal disparity levels are ill-specified.
> > >
> > > When the feasibility guard is not activated, we see that the outer grid search is modestly timed, coming in at around 4-6 minutes per seed (Tables 4 and 5).
> > >
> > > _Summary:_ Ultimately, our methods are able to scale in performance and speed to
> > > large datasets
> > > such as ACSIncome
> > > (which have hundreds of thousands of data points and
> > > almost a dozen features)
> > > for the challenging setting of controlling multiple linear-fractional and linear group fairness constraints (Table 3) -- a setting on which existing post-processing methods (see, e.g., Celis et al., 2019; Hsu et al., 2022) have been unable to demonstrate similarly strong empirical performance.

---

> ### Author Response · Authors · 2025-11-20
>
> **A2: Scalability concerns (ctd.).**
>
> **Table 3**: Performance on the test set for ACSIncome ($|\mathcal{A}|{=}5$).
> The disparities $\delta_{\textrm{EOpp}}, \delta_{\textrm{PP}}, \delta_{\textrm{FOR}}$ are controlled at level $0.10$ whenever they are active. Cells in green indicate that the fairness constraint is satisfied within two standard deviations at level $0.10$, whereas cells in red indicate violation.
> | Method      | Acc         | DP          | EOpp           | PEq         | PP             | FOR            | Interv.     |
> | ----------- | ----------- | ----------- | -------------- | ----------- | -------------- | -------------- | ----------- |
> | Baseline    | 0.78 ± 0.01 | 0.24 ± 0.01 | 🟥 0.25 ± 0.03 | 0.09 ± 0.02 | 🟥 0.23 ± 0.03 | 🟩 0.08 ± 0.01 | 0.00 ± 0.00 |
> | Oracle      | 0.57 ± 0.04 | 0.48 ± 0.05 | 🟩 0.10 ± 0.01 | 0.57 ± 0.08 | 🟩 0.10 ± 0.01 | 🟩 0.10 ± 0.01 | N/A         |
> | **ROCF-AD** (ours) | 0.57 ± 0.02 | 0.48 ± 0.03 | 🟩 0.11 ± 0.01 | 0.57 ± 0.05 | 🟩 0.11 ± 0.01 | 🟩 0.11 ± 0.03 | 0.06 ± 0.01 |
> | **ROCF-LF** (ours) | 0.56 ± 0.02 | 0.49 ± 0.02 | 🟩 0.11 ± 0.01 | 0.59 ± 0.04 | 🟩 0.11 ± 0.01 | 🟩 0.11 ± 0.03 | 0.05 ± 0.01 |
> | LPP-EOpp    | 0.79 ± 0.00 | 0.15 ± 0.01 | 🟩 0.11 ± 0.01 | 0.06 ± 0.01 | 🟥 0.31 ± 0.01 | 🟩 0.12 ± 0.01 | 0.00 ± 0.00 |
>
> **Table 4**: Summary statistics for feasibility guard triggers for the nominal disparity levels; $p$ is the proportion of runs and $\mu$ is the average expansion factor with listed s.d.
> | Setting    | $ \delta $ = (DP, EOpp, PEq, PP, FOR) | COMPAS               | ACSIncome ($\| \mathcal{A} \| =2$)      | ACSIncome ($\| \mathcal{A} \|=5 $)         |
> | ---------- | -------------------------------- | -------------------- | -------------------- | ----------------------- |
> | ROCF-AD/LF | (0.05, 0.05, 0.05, 0.05, None)   | p=0.00 ; μ=1.00±0.00 | p=0.00 ; μ=1.00±0.00 | p=0.00 ; μ=1.00±0.00    |
> | ROCF-AD/LF | (None, 0.1, None, 0.1, 0.1)      | p=0.10 ; μ=1.01±0.02 | p=0.00 ; μ=1.00±0.00 | 🟦 p=0.66 ; μ=1.04±0.05 |
> | Oracle     | (0.05, 0.05, 0.05, 0.05, None)   | p=0.00 ; μ=1.00±0.00 | p=0.00 ; μ=1.00±0.00 | p=0.00 ; μ=1.00±0.00    |
> | Oracle     | (None, 0.1, None, 0.1, 0.1)      | p=0.08 ; μ=1.00±0.02 | p=0.00 ; μ=1.00±0.00 | 🟦 p=0.64 ; μ=1.05±0.08 |
>
> **Table 5**: CPU runtime (mean $\pm$ s.d., seconds) for COMPAS, ACSIncome ($|\mathcal{A}|{=}2$), and ACSIncome ($|\mathcal{A}|{=}5$). The disparities $\delta_{\textrm{EOpp}},\delta_{\textrm{PP}}$ and $\delta_{\textrm{FOR-parity}}$ are controlled at level $0.10$ for our methods ROCF-AD/LF, while LPP-EOpp only controls for equality of opportunity.
> | Method   | COMPAS (sec.)      | ACSIncome ($\| \mathcal{A} \| =2$) (sec.) | ACSIncome ($\| \mathcal{A} \| =5$) (sec.) |
> | -------- | --------------- | --------------------- | --------------------- |
> | ROCF-AD  | 377.81 ± 312.37 | 272.50 ± 3.52         | 3903.40 ± 1550.97     |
> | ROCF-LF  | 385.55 ± 314.28 | 273.22 ± 3.96         | 3675.52 ± 1591.78     |
> | LPP-EOpp | 0.06 ± 0.02     | 17.85 ± 1.16          | 18.52 ± 0.58          |
>
> **A3: Sensitivity to data size.**
> Any algorithm for working with small data sets is potentially susceptible to the sampling error inherent due to a small sample size.
> Prior methods
> for fair classification are not an exception, and they are often
> significantly
> affected by limited dataset sizes:
> under standard datasets and similar split ratios for training, post-processing, and evaluation, our empirical analyses in §5 illustrate
> that
> existing post-processing methods like Celis et al., 2019; Hsu et al., 2022
> fail to achieve non-trivial accuracy while maintaining approximate fairness.
> In contrast,
> our method demonstrates strong empirical performance on relatively small, standard datasets like COMPAS with $\approx$5000 samples.
> This demonstrates that our method is relatively stable to small dataset sizes, especially when compared to prior methods.

---

> ### Author Response · Authors · 2025-11-20
>
> **A4: Post-processing limitations.**
> Post-processing
> approaches start with a fixed classifier and update it to aim at ensuring various forms of algorithmic fairness.
> This paradigm is one of the
> most popular and successful approaches towards reducing disparity because of a number of reasons including:
> - Its lightweight nature (it does not require retraining a classifier from scratch)
> - Broad compatibility (it can work with any pre-trained classifier).
> For this reason,
> post-processing algorithms have found successful application in a variety of domains (see e.g., Nandy et al., 2022; Weerts et al., 2023; Mackin et al., 2025, etc.)
> and are particularly useful when access to the training data or re-training the base model is infeasible (see discussion in §A.1 of our work).
>
>  Nonetheless,
>  as the reviewer points out,
> they come with the potential limitation that they do not update the learned representations of the underlying model.
> When this property is required, it is necessary to use alternative training paradigms, such as preprocessing or in-processing methods, for achieving algorithmic fairness.
> In future work, it will be an important question to study how our approach could be adapted to these alternative training strategies.
> For now,
> we would like to argue that
> the post-processing paradigm is both practically useful and popular,
> and that our methods demonstrate significant improvement compared to prior state-of-the-art approaches in this area.
>
> **A5: Generalization beyond binary tasks.**
> In the revised manuscript, we now provide an extension of our framework (namely, a description of the algorithms) for the multi-class case,
> following the formalizations of multi-class group fairness constraints in Alghamdi et al., 2022; Xian et al., 2024.
>
> At a high level, we show that our method naturally extends to settings with $C>2$ classes by replacing the ROC curve with a polytope of confusion-rate matrices,
> whose vertices correspond to deterministic multi-class post-processors.
> We describe how to construct this empirical polytope,
> how to impose multi-class analogues of demographic parity, equalized odds, and sufficiency constraints,
> and how to recover a realizable randomized classifier via structured mechanisms (e.g., label-flipping or anti-diagonal mixing).
>
> This extension requires no new assumptions; it relies only on convexity and the same polytope-based optimization as in the binary case. Full details are provided in §A.3.5. of the updated manuscript.
> This analysis suggests that our approach is indeed not fundamentally tied to the binary classification setting, but that similar ideas may extend to the setting of a multi-class classification problem.
>
> **References**
>
> Alghamdi, W., Hsu, H., Jeong, H., Wang, H., Michalak, P., Asoodeh, S., & Calmon, F. (2022).
> Beyond adult and compas: Fair multi-class prediction via information projection.
> Advances in Neural Information Processing Systems, 35, 38747–38760.
>
> Celis, L. E., Huang, L., Keswani, V., & Vishnoi, N. K. (2019).
> Classification with fairness constraints: A meta-algorithm with provable guarantees.
> FAT* Conference, 319–328.
>
> Ding, F., Hardt, M., Miller, J., & Schmidt, L. (2021).
> Retiring adult: New datasets for fair machine learning.
> Advances in Neural Information Processing Systems, 34, 6478–6490.
>
> Ezzeldin, Y. H., Yan, S., He, C., Ferrara, E., & Avestimehr, A. S. (2023).
> FairFed: Enabling group fairness in federated learning.
> AAAI Conference on Artificial Intelligence, 37(6), 7494–7502.
>
> Fabris, A., Messina, S., Silvello, G., & Susto, G. A. (2022).
> Algorithmic fairness datasets: The story so far.
> Data Mining and Knowledge Discovery, 36(6), 2074–2152.
>
> Groeneboom, P. (2021).
> Grenander functionals and Cauchy’s formula.
> Scandinavian Journal of Statistics, 48(1), 275–294.
>
> Hsu, B., Mazumder, R., Nandy, P., & Basu, K. (2022).
> Pushing the limits of fairness impossibility: Who’s the fairest of them all?
> Advances in Neural Information Processing Systems, 35, 32749–32761.
>
> Mackin, S., Major, V. J., Chunara, R., & Newton-Dame, R. (2025).
> Post-processing methods for mitigating algorithmic bias in healthcare classification models: An extended umbrella review.
> BMC Digital Health, 3(1), Article 26.
>
> Nandy, P., DiCiccio, C., Venugopalan, D., Logan, H., Basu, K., & El Karoui, N. (2022).
> Achieving fairness via post-processing in web-scale recommender systems.
> FAccT Conference 2022, 715–725.
>
> Weerts, H., Dudík, M., Edgar, R., Jalali, A., Lutz, R., & Madaio, M. (2023).
> Fairlearn: Assessing and improving fairness of AI systems.
> Journal of Machine Learning Research, 24(257), 1–8.
>
> Wightman, L. F. (1998).
> LSAC National Longitudinal Bar Passage Study.
> LSAC Research Report Series.
>
> Xian, R., & Zhao, H. (2024).
> A unified post-processing framework for group fairness in classification.
> arXiv:2405.04025.

---

> > ### Author Response · Authors · 2025-11-28
> >
> > Dear Reviewer,
> >
> > We hope you are well. We would like to follow up on this thread. We have posted a detailed response, aiming to answer each of your concerns directly in a point-by-point manner. We hope that this has addressed your concerns. If you have any additional questions or concerns, please let us know. Otherwise, if all of your concerns have been addressed, and therefore you believe that the paper can be published, we would be grateful if you could reconsider your assessment and reflect this in your score.
> >
> > Thank you again for your careful reading of the paper and valuable suggestions.
> > Sincerely,
> > The authors.

---

### Author Response · Authors · 2025-11-20

We thank all of the reviewers for their insightful questions and comments. In the revised manuscript, we have made the following changes:

- **New theoretical result.** A new theorem with finite-sample guarantees for the optimal fairness-constrained operating characteristic rates, conditional on a fixed score function; please see Theorem 4.2 in §4.2 of the revised manuscript.

- **New empirical analyses.** Added new empirical analyses with an additional dataset and a sensitivity analysis of the granularity of the outer grid-search; please see §5 and §A.6.3.

- **New extension.** Added an extension of our approach from the binary class setting to the multi-class setting (§A.3.5.).

- **Clarifications \& Restructuring.** Added clarifications and performed a slight restructuring of the paper for easier digestibility; please see replies to the comments A4, B1, B2, D3, D5, D8, and D9 below.

In our responses below, we have done our best to address all of the reviewers' points and would be happy to continue the discussion if there are any further or remaining questions.

---

> ### Author Response · Authors · 2025-12-02
>
> After the reviewer's replies, we have made the following additional changes to the manuscript:
>
> **New empirical analyses.** Added new empirical analyses with an additional NLP-based dataset (BiasBios, in §5) and a non-uniform relaxation policy for the feasibility guard (§A.2.1 for methodology and §A.6.4 for experimental results).
>
> **Clarifications & Restructuring.** Added further clarifications, in particular for the sample and computational complexity of the multi-class extension, and performed a slight restructuring of the paper for easier digestibility (see §A.3.4 and the author's 2nd reply to Reviewer keRQ below).
>
> Altogether, we believe we addressed all issues and concerns brought up by the reviewers, but unfortunately the reviewers were not able to raise their score due to external circumstances outside of their control. We thank the reviewers again for all of their time and effort.

---

### Meta-Review · Area_Chair_r8D9 · 2026-01-07

**Summary:**

This paper proposes a post processing method for fair binary classification by intervening on the classifier ROC. The authors optimize over group-wise ROC convex hulls using a centroid-based linearization of linear-fractional constraints, leading to different notions of fairness to be satisfied with near-optimal accuracy loss. The approach is extended to multiple protected attributes, multiple constraints, and (theoretically) to the multi-class setting.

There's a consensus among reviewers, particularly after the authors' rebuttal, that the presented approach is an elegant contribution to post-processing methods for fair classification. Many of the reviewers comments were common (limitations to binary settings, lack of sensitivity and computational analysis to level of discretization, lack of further datasets), and all of these were address to a very high level and quality by the authors in their rebuttal. Thus, I'm recommending acceptance.

**Reviewer Concerns:**

### Rev YVAP
- Questioned the small number of datasets employed for validation (just COMPAS and ACSIncome). In the rebuttal, the authors included the Lawschool and BiasBios datasets.

- Raised concerns on the computational cost of the grid search and scalability to many groups or constraints. The authors provided a detailed complexity analysis and runtime measurements, as well as sensitivity to the level of discretization, addressing this concern.

- Questioned whether the estimation of the ROC could be of concern in groups or settings with small samples. The authors argued that, even in cases with small number of samples (as in COMPAS), the proposed method still works well.

- The reviewer questioned the fact that this is only a post-processing approach, which is acknowledged and clearly articulated by the authors in their rebuttal.

- The reviewer noted on the limitation of only addressing the binary case, but the authors further extended the study to (theoretically) to confusion-rate polytopes in multi-class settings.

### Rev v8n8

- The reviewer had suggestions on the organization of the technical theoretical results, and asked mostly about the sensitivity to the grid resolution. The authors provided a self-contained algorithm description and re-organized the proofs for clarify. They further added the sensitivity analysis to the resolution of the grid.

### Rev nf54

- Questions the level of the contribution as too incremental. To this, the authors responded by extending their theoretical results with finite-sample guarantees for the optimal fairness-constrained operating characteristic rates, conditional on a fixed score function.

- Had concerns on the stability of the approach based on bi-level optimization with grid search. The authors argued that the grid search is still feasible in practice, and provided a runtime analysis.

- The reviewer further asked about sample complexity and error propagation analysis when the base score is not bayes optimal. The authors did not provide these, arguing that it is a non-trivial extension.

### Rev keRQ
- The reviewer had some initial concerns on the limited number of datasets, the limitation to only binary settings, narrow baseline settings, and grid sensitivity. These were addressed significantly as per the authors rebuttal (see above).
- They noted that intervention optimality is only proven conditionally and not globally. This was acknowledged and clarified by the authors.
- The reviewer further had questions on practical deployment (e.g. oracle definitions) which were easily clarified.

**Reviewer Scores:**

- Rev YVAP (originally 4): did not engage with the authors after the rebuttal. However, I believe the level of detail in the authors' responses should have increased their score to 5 or 6.
- Rev v8n8 (originally 6). All of the questions raised by this reviewer were addressed, but the review was not confident (2), so I'm weighing it very little.
- Rev nf54 (originally 4); the reviewer did not engage further, but I believe that they would have maintained or increased their score, given the provided extended theoretical results (as well as the empirical ones).
- rev keRQ (originally 8): while most of their concerns were addressed and the paper improved, the reviewer stated that some of these improvements were not substantial, and thus maintained their score.

---

### Decision · Program_Chairs · 2026-01-26

Accept (Poster)